# PISM-LakeCC: Implementing an adaptive proglacial lake boundary into an ice sheet model

Sebastian Hinck[1], Evan J. Gowan[1,2], Xu Zhang[1,3], and Gerrit Lohmann[1,2]

[1]Alfred Wegener Institute Helmholtz Centre for Polar and Marine Research, Am Handelshafen 12, 27570 Bremerhaven, Germany.
[2]MARUM - Center for Marine Environmental Sciences, University Bremen, Leobener Strasse 8, 28359 Bremen, Germany.
[3]College of Earth and Environmental Science, Center for Pan-Third Pole Environment, Lanzhou University, 222 Tianshui South Road, Lanzhou City, China

**Correspondence:** Sebastian Hinck (sebastian.hinck@awi.de)

**Abstract.** During the Late Pleistocene and Holocene retreat of palaeo ice sheets in North America and Europe, vast proglacial lakes existed along the land terminating margins. These proglacial lakes impacted ice sheet dynamics by imposing boundary conditions analogous to a marine terminating margin. These lacustrine boundary conditions cause changes in the ice sheet's geometry, stress balance and frontal ablation and therefore affect the entire ice sheet's mass balance. Despite this, dynamically evolving proglacial lakes have rarely been considered in detail in ice sheet modelling endeavors. In this study, we describe the implementation of an adaptive lake boundary into the Parallel Ice Sheet Model (PISM), which we call the PISM-LakeCC model. We test our model with a simplified glacial retreat setup of the Laurentide Ice Sheet (LIS). By comparing the experiments with lakes with control runs with no lakes, we show that the presence of proglacial lakes locally enhances the ice flow, which leads to a lowering of the ice sheet surface. In some cases, this also results in an advance of the ice margin and the emergence of ice lobes. In the warming climate, increased melting on the lowered ice surface drives the glacial retreat. For the LIS, the presence of lakes triggers a process similar to the marine ice sheet instability, which causes the collapse of the ice saddle over Hudson Bay. In the control experiments without lakes, Hudson Bay is still glaciated when the climate reaches present day conditions. The results of our study demonstrate that glacio-lacustrine interactions play a significant role in the retreat of land terminating ice sheet margins.

## 1 Introduction

During the Last Glacial Maximum (LGM), ice sheets covered parts of the Eurasian and North American continents. As they retreated, the topography was left deeply depressed due to delayed glacial isostatic adjustment (GIA). Vast lakes formed along the ice margins due to a combination of this depression, and the blocking of drainage routes by the ice sheets. Examples of these palaeo lakes are Lake Agassiz in North America (Teller and Leverington, 2004) and the Baltic Ice Lake in Eurasia (Björck, 1995). Lacustrine sediments and shorelines provide geological evidence of their presence and extent. Reconstructions of Lake Agassiz, which bordered the Laurentide Ice Sheet (LIS) for several hundred km, for example, suggest areal extents of several

hundred thousand $\text{km}^2$ and water depths of several hundred meters (Teller et al., 2002; Leverington et al., 2002; Teller and Leverington, 2004).

The basins of proglacial lakes constantly evolved, because the ice sheet margin and topography were not static. Reorgani-
zation of the lakes' drainage networks and sudden drainage events due to the opening of lower spillways may have impacted the global climate by perturbing the thermohaline circulation system of the oceans (Broecker et al., 1989; Teller et al., 2002; Peltier et al., 2006; Condron and Winsor, 2012). Furthermore, the presence of a lake at the ice margin impacts the ice dynamics by adding a marine-like boundary condition to the ice sheet. This changes the boundary conditions of terrestrial ice margins: modification of the thermal regime at the submerged ice base, formation of ice shelves, increased ice loss due to melting and
calving, and enhanced basal sliding near the grounding line due to decreased effective pressure at the ice base (e.g. Carrick and Tweed, 2013). These processes can lead to the formation of ice streams (Stokes and Clark, 2003; Margold et al., 2015), which impact the mass balance of large parts of the ice sheet. Interactions between ice and lake resemble processes at the marine boundary, but their magnitude might differ due to various differences at lacustrine and marine boundaries (Benn et al., 2007).

In periods with large amounts of meltwater that caused the formation of proglacial lakes the lake-ice interactions might have been a key factor to explain rapid glacial retreat. As an example, one hypothesis for the cause of the $8.2\text{ka}$ event, a period characterized by a sudden drop in Northern Hemispheric mean temperatures, is the rapid demise of the central LIS (Carlson et al., 2008; Gregoire et al., 2012; Matero et al., 2017; Lochte et al., 2019) leading to a large freshwater input into the Labrador Sea. Although the retreat is assumed to be governed by the negative surface mass balance due to enhanced melting in a warming
climate (Carlson et al., 2009; Gregoire et al., 2012), other dynamical effects, such as marine and lacustrine interactions, might have further amplified this effect (Matero et al., 2017, 2020). Matero et al. (2020) state the lack of proper representation of Lake Agassiz/Ojibway along the southern ice margin in their modeling study as a source of uncertainty. Lake reconstructions of this time suggest water depths up to several hundred meters (Teller et al., 2002; Leverington et al., 2002). By adding a simple parameterization for lacustrine calving to their energy balance type box model, Fowler et al. (2013) could show that
this feedback can potentially explain the $100\,\text{kyr}$ climate oscillations since the mid-Pleistocene.

In most previous numerical studies, glacio-lacustrine interactions have not been considered or are applied in an unrealistic way. Often ice dynamical models apply marine boundary conditions at places with surface elevation below global mean sea level (e.g. the PISM authors, 2015). This can lead to inner-continental ocean basins which can be considered as 'fake' lakes, with a water level that can greatly differ from the true level of a ponded proglacial lake (Matero et al., 2020). Cutler et al. (2001)
and Tsutaki et al. (2019) investigated the impact of a prescribed lake level on ice dynamics using two-dimensional flow-line models. Tarasov et al. (2012) included the effects of proglacial lakes in an ice sheet model through a calving parameterization and thermodynamic refreezing scheme. Recent work of Sutherland et al. (2020) presents a novel approach analyzing the impact of lake boundary conditions on ice dynamics using the three-dimensional thermo-mechanically coupled ice sheet model BISICLES (Cornford et al., 2013). Their regional survey treats the post-LGM retreat of a mountain glacier in the Southern Alps
in New Zealand, terminating in glacial Lake Pukaki. They subtracted the reconstructed water level from model topography and adapted parameters controlling marine boundary interactions for lakes. This approach, however, only allows one fixed water

level and is thus not applicable for more complex ice sheet scenarios featuring multiple, time variable lakes. In another recent study Quiquet et al. (2021) applied the GRISLI model (Quiquet et al., 2018) to simulate the glacial retreat of the North American ice sheets during the last glacial cycle. At the southern ice margin they simulated the effect of a proglacial lake by locally elevating the sea level. As in the aforementioned study, this water level is fixed. The model results show an accelerated demise of the LIS caused by the occurrence of a proglacial lake ice sheet instability (PLISI). The need for including glacio-lacustrine interactions into numerical ice sheet modeling attempts was recently highlighted in review articles by Margold et al. (2018) and Carrivick et al. (2020). The latter referenced article discusses challenges for the implementation of such a lake-ice boundary condition for ice sheet models.

In this study, we describe the implementation of an adaptive lake boundary into a 3D thermo-mechanically coupled ice sheet model. Implementation into the Parallel Ice Sheet Model (PISM[1]; Bueler and Brown, 2009; Winkelmann et al., 2011) is done using a generalization of the model's marine boundary condition. The fundamental algorithm that determines the lake basins is based on the standalone model LakeCC (Hinck et al., 2020), so the model is called PISM-LakeCC. 'CC' stands for connected components, the algorithm the model is based on.

We demonstrate the model's impact on the ice dynamics through the application of a simple post-LGM deglaciation scenario of the North American ice complex. A comparison with control runs shows that lakes induce strong dynamical effects on the ice sheet. Increased mass loss and reduced basal strength lead to an acceleration of ice flow upstream of the lake boundary, which effectively drains mass from the ice sheet interior. During the deglaciation, lakes form with water depths up to several hundreds meters. The differences from the control experiments become apparent with the demise of the remainder of LIS in the Hudson Bay Area. In regions where there is ice-inward sloping topography, the grounding line retreats in a self-amplified manner, which corresponds to a rapid expansion of the lake. The PLISI, in combination with the increased runoff at the lowered ice surface in the warming climate, results in an accelerated retreat of the ice sheet. In our study, this mechanism leads to the demise of the LIS and finally to the drainage of the lake through Hudson Strait. In the control experiments, Hudson Bay is still glaciated when the present day conditions are achieved.

## 2 Methodology

### 2.1 Ice sheet model

All implementations described in this work are based on the stable release v.1.2.1 of PISM, which is a 3-dimensional thermo-mechanically coupled ice sheet model. PISM's modular implementation grants the user freedom to choose between different realizations of various sub-models. In the following we give a short overview about the configuration used for our experiments. All details that diverge from the PISM defaults are listed in Tables A1 and A2 in the Appendix.

PISM's computational grid is horizontally equally spaced. We use a resolution of $20\ \mathrm{km}$. In the vertical direction, the computational box has a height of $5750\ \mathrm{m}$ and is divided into 101 layers, with height that increases quadratically from the ice

---

[1] http://www.pism.io/, accessed 2021/10/31

base. The thermal layer within the ground has a thickness of 2000 m and is divided into 11 levels. An adaptive time-stepping mechanism determines the shortest time step needed by any of PISM's sub-models.

The stress balance is modeled using a hybrid scheme based on the Shallow Ice (SIA) and Shallow Shelf Approximations (SSA) of the full Stokes equations (Bueler and Brown, 2009). We use an energy conserving flow law, the Glen-Paterson-Budd-Lliboutry-Duval law (Lliboutry and Duval, 1985; Aschwanden et al., 2012). This models the ice softness as a function of temperature and liquid water fraction.

The basal resistance is determined using a model that assumes that the base of the ice sheet is underlain by deformable till. It only allows sliding when the driving stresses exceed the yield stress of the till. This threshold value is determined by a Mohr-Coulomb formulation (Cuffey and Paterson, 2010) using a constant till-friction angle and the effective pressure in the till. The latter is estimated from the amount of water in the till, which comes from the non-water conserving hydrology model of PISM (Tulaczyk et al., 2000), and the ice thickness. The till below the water level next to the grounding line is assumed to be saturated, which reduces the basal strength. Bed deformation due to ice load is modeled in PISM using the Lingle–Clark model (Lingle and Clark, 1985; Bueler et al., 2007). It uses an idealized two-layered Earth model, approximating the Earth as a viscous upper mantle overlain by an elastic plate lithosphere.

The surface mass balance (SMB) is estimated from monthly means of precipitation and surface air temperature fields using the positive degree-day (PDD) approach (Calov and Greve, 2005). To prescribe the atmospheric forcing for the transient experiment conducted here, two additional models were implemented. Precipitation and temperature fields are interpolated between two distinct climatic states, which are weighted according to a glacial index (see Appendix D1). To prevent the ice sheet from expanding into regions and high elevations where a more advanced approach would limit precipitation, the second model sets precipitation to zero above a threshold height or accordingly to a given mask (see Appendix D2).

Marine regions of the ice sheet are defined in PISM via a flotation criterion, which describes whether the ice in areas below sea level is grounded or floating. The sea level is defined via a 2D map, which allows it to be spatially variable. In general, however, it is set to a global mean value prescribed by a scalar time series. The marine boundary treatment is described in Winkelmann et al. (2011) and Martin et al. (2011). It includes a sub-shelf-melting parameterization (Beckmann and Goosse, 2003), and sub-grid parameterizations of the ice shelf advance (Albrecht et al., 2011) and the grounding line (Feldmann et al., 2014). To parameterize the effect of calving at the ice shelf front, a combination of the Eigen-calving[2] (Levermann et al., 2012) and thickness calving mechanisms, which removes ice thinner than a given threshold thickness, is applied.

## 2.2 LakeCC

The environment along the ice sheet edge is particularly dynamic. Ice margin migration and ongoing GIA significantly impact shape and size of proglacial lakes that form within this environment. For the dynamical coupling of ice sheets and proglacial lakes, continuous updating of lake basin geometry is necessary. Depending on the complexity of the lake model, computational overhead can drastically increase. For application on continental-sized ice sheets, the use of lightweight algorithms is necessary. To reduce the complexity, trade-offs have to be made. This section describes how the LakeCC model computes lake

---

[2]Although the Eigen-calving mechnism was activated for all experiments, the model was found to not have impacted the results.

reconstructions and prepares them for use in PISM. Details on how this lake boundary condition affects the ice dynamics are provided in Sect. 2.3. Limitations of the current implementation are discussed in Sect. 2.4.

For the model described here, we assume that lake basins tend to be entirely filled. This assumption might be valid for proglacial lakes during times of glacial retreat, when meltwater is entering the lake. However, rapid changes in the boundary conditions resulting from this approach often cause numerical instabilities that cause the model to crash (see Sec. 2.4.3). To overcome this, changes in the lake geometry need to be monitored, and the water level needs to be adjusted gradually. Our implementation uses two different fields to realize this: the target level and the lake level (see Figure 1).

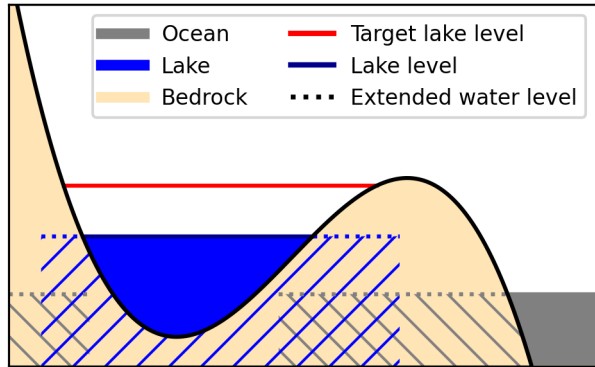

**Figure 1.** Sketch of the PISM LakeCC and SL2DCC models. The target water level indicates the maximum water height of a lake basin before it overflows. The actual lake level, as it is seen from the rest of the ice model, is slowly raised (or lowered) towards the target level. The dotted lines and hatched areas indicate where the respective water levels extend into the ground. To prevent the formation of inland ocean basins, the SL2DCC model sets the sea level in those basins to invalid. Note that lake level and the gap in the sea level field are extended by one cell, respectively. For details on this the reader is referred to the main text.

The target level field holds information about the maximum water level of all lake basins of the domain, while in the other field the actual water level is stored. These fields resemble a spatial map whose cells hold the respective information of the associated lake basin; cells outside such basin are set to be invalid. To determine the target level, a simple lake filling algorithm is used that is adapted from Hinck et al. (2020). Changes in this field, i.e. the appearance or disappearance of lake basins, are monitored by comparing the target level against the lake levels from the previous time step. This is necessary because the gradual filling algorithm relies on new lake cells being properly initialized in the lake level field. In general, new lakes are initialized with zero water depth. However, there are special cases, such as adding a lake basin to an existing lake or adding a basin that has previously been connected to the ocean, that need more advanced treatment. For more details on this, see Appendix B.

After initialization, the lake level is gradually adapted towards the target level. Determining a rate $\gamma$ at which the water levels change is not trivial. For simplicity the fill-rates in our model are assumed to be constant. See Sect. 2.4 for more details on this.

Merging of lakes or adding a new basin to an existing lake might result in an unbalanced water level. To quickly overcome this unrealistic scenario and balance the lake level, water levels for each lake are adapted from a common level. When water level is rising, this common level is chosen to be the lowest water level of that lake, $h_{\mathrm{min}}$, while the highest level, $h_{\mathrm{max}}$, is selected for the falling water level. The potential new water level, $h$, is determined by adding or subtracting the respective change for a given time-step, $\mathrm{dt}$:

$$h = h_{\mathrm{min/max}} \pm \gamma \cdot \mathrm{dt} \tag{1}$$

Only when $h$ has exceeded the lake level, does its value get updated. This makes sure that the lake level is slowly raised or lowered towards the target level, while aligning the water levels. When a lake disappears, *i.e.* the target level is set to be invalid, the lake is not immediately removed, as this could also lead to numerical instabilities. The water level is gradually lowered until it is below the bed elevation, or the ice sheet is grounded. If a basin disappears because it merged with the ocean, the lake level is gradually changed until sea level is reached, and then removed.

In the final step, the lake basins are all extended by one grid cell in each direction. This treatment serves two purposes: (i) providing information about the presence of a neighboring lake to the ice sheet model to possibly adapt its boundary conditions, and (ii) helping to close gaps in between ice margin and adjacent lake when the ice is retreating in the following time-step. It should be noted that this does not artificially enlarge the lake, as the ice sheet model independently computes the lake geometry based on the information of the lake model and the water level in these cells is below the flotation threshold. The dotted lines in Fig. 1 illustrates this.

Another issue that we resolve here is the way how PISM and other ice sheet models (*e.g.* BISICLES, as noted in Matero et al. (2020) and Sutherland et al. (2020)) handle sea level and how it affects the LakeCC model. The sea level elevation is usually assumed to be globally constant and hence is set by a scalar (time series). Differentiation between ocean and land is only done by checking if the bed elevation is above or below the sea level. Consequently, isolated basins can occur within the continent, and if they are below sea level, are falsely regarded as ocean. As described in Hinck et al. (2020), the LakeCC algorithm relies on a proper land-sea mask to properly determine lake basins. This mask can be computed by the SL2DCC model (for more details see Appendix B4), which is also based on Hinck et al. (2020) and uses the connected components (CC) algorithm. It checks potential ocean cells for connectivity with the domain boundary and marks isolated basins as invalid. The corresponding cells are recognized by the ice sheet and the lake model as land cells, which are potentially available for lake formation.

## 2.3 Lacustrine impact on ice dynamics

In this implementation, lakes are treated as a generalization of the existing marine boundary condition in PISM that dynamically adapts to changing environments. Most parts of the ice sheet model that access the sea level elevation do this to obtain information about the current geometry, e.g. to determine water depth. The code is adapted in a way that, if a lake is present, the lake level is used in the calculation instead of sea level and the water density is adjusted accordingly. Generally, the parameterizations provided by PISM for a marine boundary are applied analogously at the lake interface. This, however, may not

be the optimal treatment in every case. The model might benefit from future implementations of advanced or more specialized lake boundary treatment. Such limitations are discussed in Sect. 2.4.

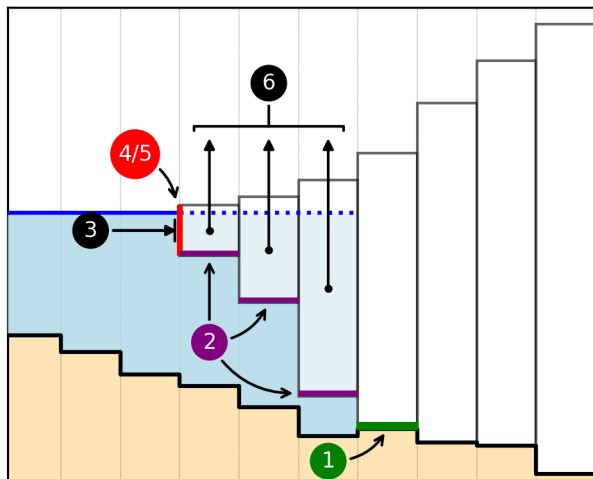

**Figure 2.** Different ways how the ice dynamics are impacted by a marine or lacustrine environment in PISM. 1 – Till in cells at the grounding line is assumed to be water-saturated, which reduces basal friction. 2 – Sub-shelf melting/refreezing. 3 – Pressure of the water column against the submerged part of the grounded or floating part of the ice margin. It can be further increased by modeling the ice mélange. Parameterization of mass-loss due to – 4 – calving at the shelf edge and due to – 5 – frontal melt at the submerged ice margin. 6 – Where ice fulfills the flotation criterion it becomes buoyant, hence lowering ice surface slope, eliminating basal friction and exposing the ice to shelf processes. Sub-grid treatment of shelf edge and grounding line position are not explicitly marked here.

Figure 2 shows the various locations where an aquatic environment impacts the ice dynamics in PISM. In the following we
will describe the parameterizations used in this setup to describe the lacustrine impact on ice dynamics. The numbers refer to their respective label in Fig. 2.

1 – Lubrication of ice base at grounding line: Cells with grounded ice below water level next to a lake are assumed to have saturated till. This reduces the effective pressure at the base of the ice sheet and reduces basal resistance at this location.

2 – Sub-shelf melting/ refreezing: In the current configuration, the ice sheet model does not differentiate between ice shelves
in marine and lacustrine environments, i.e. the same parameterization for ice base temperature and sub-shelf mass-flux are used. Ice temperature at the base is set to the pressure-melting point, which is a function of pressure, hence ice thickness. Sub-shelf mass-flux is calculated based on the parameterization from Beckmann and Goosse (2003). This formulation explicitly assumes a marine environment. It models the mass-flux proportional to the difference between the pressure-dependent freezing point of saline water and the temperature of the ambient ocean. Salinity and ambient ocean temperature are prescribed in this
model implementation and thus can not be adjusted for freshwater environments. The model's choice of parameters is such that

the temperature of the ambient water is always above the calculated freezing temperature and consequently the model does not allow for refreezing at the shelf base.

3 – Ice-marginal pressure difference and mélange back-pressure: The pressure difference against the submerged ice margin is done analogously to the marine boundary, but evaluated for freshwater density. Ice mélange, which is not modeled here, can have a buttressing effect on the ice sheet.

4 & 5 – Calving and frontal melt: Frontal retreat is a complex but also very sensitive factor in modeling both, marine and lacustrine terminating ice sheets. In this study we only parameterize calving, with no frontal melt schemes applied. At the lake boundary, the same calving mechanisms are applied as for the ocean (compare with Sect. 2.1). The implementation of the thickness threshold calving was adapted to accept a distinct threshold value that is used at lacustrine boundaries ($\Delta h_{\mathrm{L}}$).

6 – Formation of ice shelves: Where ice thickness is below a certain threshold, determined by depth and density of the adjacent water body, the ice floats. Formation of an ice shelf not only gives rise to the appearance of the previously mentioned effects (1-5), but also directly impacts ice dynamics. Due to geometric changes of the ice sheet, the flow regime is altered. Furthermore, friction at the shelf base is negligible.

By impacting the ice sheet geometry and mass balance further, secondary mechanisms are triggered that feed back onto the ice dynamics. These include changes in GIA and affecting the local climate due to temperature elevation feedback.

## 2.4 Limitations

For the PISM-LakeCC model, several assumptions were made to reduce the complexity of the problem and adaptively updating the lake boundary condition within the ice sheet model, without adding too much computational overhead and destabilizing the numerical system. These include both aspects of the model, the lake reconstruction and the coupling to the ice sheet. In the following, known limitations are listed and discussed.

### 2.4.1 Lake reconstruction

The PISM-LakeCC model uses the same numerical grid as the ice sheet model. For hydrological applications, such as lake basin reconstructions, the resolution an ice sheet model usually operates on is too coarse to resolve spillways through the terrain. At low resolutions, lake volumes are potentially overestimated (Berends and van de Wal, 2016). Since we are not so much interested in lake volume, the ice margin position and bed deformation due to GIA are, in our case, expected to be more important than resolution (Hinck et al., 2020). Considering the uncertainties of these fields retrieved from an ice sheet model, resolution is regarded as a secondary issue. A method, proposed by Hinck et al. (2020), is implemented into the LakeCC model to adapt to the low resolution. In that study, satisfactory results were obtained by applying the LakeCC algorithm to a low resolution PD topography map of North America after applying a correction. This correction field was obtained from a high resolution dataset in a preprocessing step by applying a minimum filter that retains the lowest surface elevation within a certain distance of each grid cell, and calculating the difference with the lower resolution topography. The PISM-LakeCC model reads this field from an input file and applies it to the low resolution topography prior to the model update. If higher resolved input

fields are available, e.g. from an external GIA model, the LakeCC model could be modified to do the calculations on that field instead and interpolate the output back onto the ice sheet model grid.

We further assume that all lakes tend to fill to the brim, i.e. there is no accounting for conservation of water mass. Doing this properly would require updating and accounting of the hydrological network and lake basins, accumulating all water fluxes and finally iteratively redistributing overflow down-gradient. This requires highly resolved hydrology, including water fluxes of climatological and glacial origin. The required data is not available in our framework and furthermore, such algorithm would dramatically increase computational cost. During glacial retreat, when meltwater is pervasive, the assumption of filled lakes is

likely valid.

 One process that can limit the maximum fill height of a lake is sub-glacial drainage. In the LakeCC model it is only crudely included via the flotation criterion: when an ice dam becomes buoyant and opens a new drainage route. In reality, however, sub-glacial drainage can also happen on much smaller scales through channels underneath the ice. This process could lead to repeated lake drainage and refilling events. Even though sub-glacial drainage through channels might be an important aspect

of glacio-lacustrine interactions, its parameterization is not trivial and is not included in our model.

 To avoid numerical instabilities due to sudden jumps in water level, the water level is gradually adapted, which can lead to unphysical situations. When lakes merge, for example, this can lead to an unbalanced water level. Furthermore, small, shallow lakes can be overrun by the forward advancing ice margin, which creates an artificial sub-glacial lake, until the water level dropped below the floating threshold. These issues can not be avoided using such a simple model. However, the algorithm

automatically resolves these problems within a few time steps.

### 2.4.2 Glacio-lacustrine interactions

The lake-ice boundary is treated, in general, the same as the marine ice boundary (see Sect. 2.3). Some parameterizations are explicitly formulated for the marine ice margin and other processes are reported to substantially differ between lacustrine and oceanic environments (Carrivick et al., 2020). Implementation of specific lacustrine processes could yield a more appropriate

treatment of lake boundaries. In the following we discuss the lake-ice interactions with respect to the validity at the lacustrine ice margin. The numbers refer to the processes in Fig. 2.

 For our experiments, the so-called slippery grounding line parameterization (1) has been used, which reduces the basal friction in the cells upstream the grounding line. This simple model therefore adds a direct dependency to the grid resolution. Furthermore, an increased sensitivity of the grounding line is reported when using this parameterization (Golledge et al., 2015).

For this reason, sensitivity tests were run that confirm these findings; these results are presented in the supplementary material and discussed in Sect. 4.2. Due to lack of a more advanced implementation of lubrication at the grounding line, we apply the slippery grounding line treatment.

 At the shelf base (2), mass flux is parameterized using the model proposed by Beckmann and Goosse (2003). The model relates the mass flux to the difference between the pressure dependent freezing point of saline water and the temperature of

the ambient ocean. Freshwater, however, behaves differently from saline water, as its temperature is always above the melting point of ice, and due to the properties of freshwater, the densest waters at the lake bottom are around $4°C$. Due to the difference

in density between fresh and ocean water, the layer of melt water underneath the ice shelf experiences a $\sim 200$ times higher buoyancy in seawater (Funk and Röthlisberger, 1989). In saline water this increases the flux and mixing of ambient warmer water along the ice base and increases melting. Based of these considerations, the melt flux was scaled accordingly in the *MR* sensitivity run, which is documented in the supplementary material. Because of these differences it will be important to implement a sub-shelf melting model for lacustrine environments in future studies.

A lacustrine model is also needed for frontal ablation (calving (4) and frontal melt (5)). Observations at contemporary proglacial lakes show calving rates an order of magnitude below the rates of tidewater glaciers (Funk and Röthlisberger, 1989; Warren et al., 1995; Skvarca et al., 2002; Warren and Kirkbride, 2003; Haresign, 2004; Benn et al., 2007). However, it should be noted that alpine proglacial lakes differ fundamentally in size and depth from the major palaeo proglacial lakes. In order to have some control on that, the current implementation accepts a unique parameter for the threshold thickness calving mechanism at the lacustrine boundary. Implementation of a more physically based calving model capable of accurately parameterization of mass losses in both lacustrine and marine environments, will be needed (Benn et al., 2007).

Another important issue that is ignored in our model is the effect of ice mélange (3), especially in smaller lakes. This is expected to have a buttressing effect on the ice sheet and reduce mass loss. Seasonal ice cover might even increase this effect (Mallalieu et al., 2020).

### 2.4.3  Gradual filling & numerical instability

When running the model, we observed that PISM occasional crashes when there is rapidly rising or dropping the water levels between time steps. Unfortunately, we can not tell the exact reason why the numerical solver failed, but we found that these crashes only appear shortly after the new lake or ocean basins, which are in contact with the ice, appear or vanish. From a numerical point of view, however, this is not surprising, as commonly the numerical representation of a physical system requires the underlying equations to be sufficiently smooth. These crashes, due to local unsteadiness in the fields, are known by the developers of PISM to appear[3].

With all the simplifications described above, the *LakeCC* model is not designed to exactly reconstruct the water level of proglacial lakes, nor to estimate freshwater fluxes. It rather gives a crude approximation of where potential lake basins are located and what their maximum water level is. Therefore, we assume the gradual filling mechanism using a (rather arbitrarily chosen) constant fill rate does not result in lake levels that are less valid.

If an almost instantaneous filling or draining of lake basins is desired, this could be achieved by setting a high fill rate $\gamma$. This would most likely require a reduction of the duration of the time steps, though, which increases the computational cost. In sensitivity tests, which are found in the supplementary material, several higher values were chosen. For these tests, no model crashes were observed. This might either be because we restricted the time step to $0.25\mathrm{yr}$ for all experiments, or simply because no critical situations were triggered. The results of these experiments show no fundamental difference to the *LAKE* experiment. Future implementations could target a more realistic treatment of basin filling.

---

[3]https://pism-docs.org/wiki/doku.php?id=kspdiverged, accessed: 2021/08/09

## 2.5 Experiments

To test the impact of the LakeCC model on the ice dynamics, we choose to simulate the glacial retreat of the North American ice sheets after the LGM at 21 kaBP. The computational domain covers the North American continent north of $\sim 35°$ N and Greenland and spans a rectangle of $7800$ km $\times 6600$ km (see Fig. 3). Further details on the used parameterizations are given

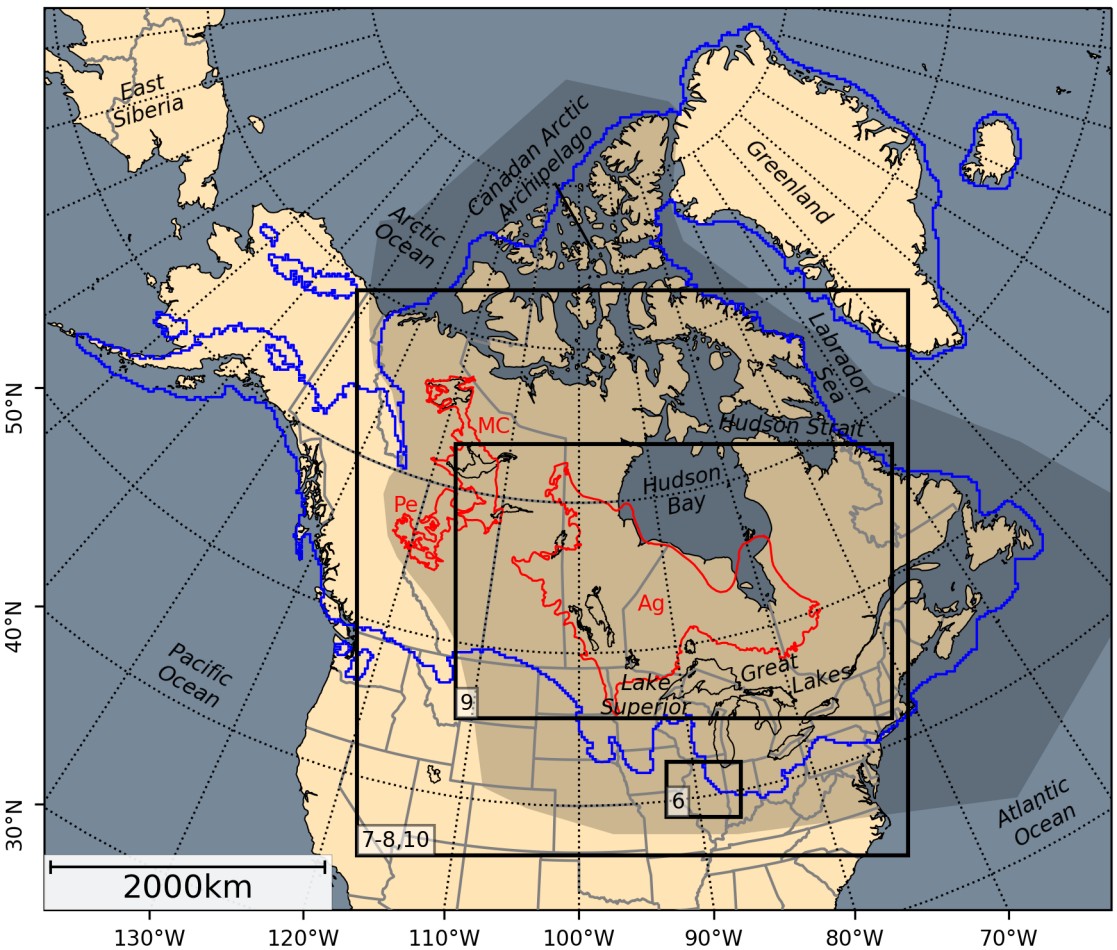

**Figure 3.** Map giving an overview of the experiment domain of the current study. Present day coastline and the major North American lakes are shown in black, political boundaries are gray. The outlines of selected palaeo proglacial lakes are shown in red: Ag – Lake Agassiz/Ojibway (Teller and Leverington, 2004), MC – Lake McConnell (Smith, 1994) and Pe – Lake Peace (Mathews, 1980; Hickin et al., 2015). Initial ice extent at LGM (Gowan et al., 2016) is shown in blue. Black rectangles visualize the map sections discussed in the text (the numbers refer to the respective figures). The shaded area shows the regions attributed to the (continental) Laurentide Ice Sheet (LIS) in this study.

in Sect. A.

The experiments for this study are all based on the same simplified model setup. Known issues with this setup are listed and shortly discussed in Sect. A2. To properly simulate a realistic glacial retreat, more advanced models and setups would be needed. However, these shortcomings are the same for all experiments. The results can therefore not be expected to match with observations, but give insights into the model impact by inter-comparison.

In total there are three main experiment setups: with lakes (*LAKE*), the control run without lakes but using the SL2DCC model for calculating a proper land sea mask (*CTRL*), and the PISM default run (*DEF*). An overview of all experiments is given in Table 1. For the *LAKE* experiment, the lacustrine calving threshold thickness $\Delta h_{\mathrm{L}}$ is set to a quarter of its marine counterpart ($\Delta h_{\mathrm{O}} = 200\mathrm{m}$) in order to reflect the fact that calving rates for freshwater terminating glaciers are reported to be an order of magnitude lower than rates observed for tidewater glaciers (Funk and Röthlisberger, 1989; Warren et al., 1995; Skvarca et al., 2002; Warren and Kirkbride, 2003). Hereafter, the experiments referenced, if not explicitly stated otherwise, are *LAKE*, including lakes, and *CTRL*, without lakes. The PISM default run *DEF* produces ocean basins within the continent, which will behave like lakes, but with water level restricted to sea level.

Additionally, a number of sensitivity experiments were run, exploring the impact of different parameters on the results. More information on these can be found in the supplementary material.

## 3   Results

Table 1 shows a list of all experiments run for this study. Apart from a brief description, it also provides a short summary of the main outcomes. The results of the *Lake* experiment are presented by focusing on selected aspects of the modeled glacial retreat. Comparison with the no-lake control experiments emphasizes the impact of proglacial lakes on the ice dynamics and ice sheet reconstruction. The highlighted area in Fig. 3 marks the region of interest for this study. In the following, when referring to the LIS, we will consider only the parts of the ice sheet that are within this region. Overview maps of the entire domain are provided in the supplementary materials to show the temporal evolution of all experiments. These plots show the topographic configuration for time slices every $500$ yr.

In the following, all expressions of time are given in model years, relative to the start of the experiments, and ice sheet volume is quantified as sea level equivalent (SLE). SLE denotes the rise of the water level, if the ice volume $V_{\mathrm{IS}}$ would melt and the equivalent freshwater volume $V_{\mathrm{fw}}$ is added into a basin of the mean ocean area $A_{\mathrm{O}} = 3.625 \cdot 10^{8}\mathrm{km}^{2}$:

$$\Delta_{\mathrm{SLE}} = V_{\mathrm{fw}}/A_{\mathrm{O}} = V_{\mathrm{IS}}\frac{\rho_{\mathrm{i}}}{\rho_{\mathrm{fw}}}/A_{\mathrm{O}}, \tag{2}$$

where $\rho_{\mathrm{i,fw}}$ are the respective densities of ice and freshwater. Note, this formulation neglects any changes in the ocean surface area and the geoid due to mass redistribution. Furthermore, the ice sheet volume accounts for both grounded and floating ice.

Figure 4 shows the temporal evolution of the SLE ice volume of the LIS for all experiments. Starting from the LGM initial state after the spin-up, the ice sheets laterally expand and gain mass for about $6$ kyr. Within this time, the LIS almost doubles its volume, before it retreats during the rest of the simulation. As the ice margin laterally expands, some shallow lakes that formed immediately after the simulation started are quickly overrun by the ice and are removed.

**Table 1.** Overview of all experiments done for this study. The first three experiments are discussed in the text, details about the other experiments can be found in the supplementary material.

| Name | Description | Results |
|------|-------------|---------|
| *LAKE*[*] | standard lake experiment, as described in the text | accelerated glacial retreat; occurrence of PLISI; Hudson Bay fully ice free at the end of the experiment |
| *CTRL*[*] | standard no-lake experiment, land-sea mask corrected by the SL2DCC model | Hudson Bay remains mostly glaciated at the end of the run |
| *DEF*[*] | PISM default no-lake setup, occurrence of inner-continental ocean basins | slightly faster glacial retreat than *CTRL*; Hudson Bay glaciated north of $\sim 58°$ N at the end of the run |
| *IncCalv* | increased calving; lacustrine thickness calving threshold set to 500m | almost immediate removal of shelf ice, which leads to a more rapid glacial retreat than in *LAKE* |
| *RedCalv* | reduced calving; lacustrine thickness calving threshold set to 20m | apart from slightly larger ice shelves, similar to *LAKE* |
| *MR* | tuning parameter for sub-shelf melting adapted to account for differences between marine and lacustrine environment | as above |
| *nSG* | slippery grounding line model disabled | strongly reduced grounding line flux leads to smaller ice shelves; no PLISI; Hudson Bay still glaciated north of $\sim 59°$ N |
| *TWO* | use of grounding line treatment proposed in Albrecht et al. (2020) (tillwater ocean) instead of slippery grounding line model | no qualitative difference to *LAKE* |
| *GIA* | adapted Earth model parameters for the Lingle-Clark bed deformation model | no qualitative difference to *LAKE*; only the timing is slightly different |
| *FR5* | lake fill rate set to 5m year$^{-1}$ | no qualitative difference to *LAKE* |
| *FR10* | lake fill rate set to 10m year$^{-1}$ | as above |
| *FR50* | lake fill rate set to 50m year$^{-1}$ | as above |

[*] default experiments

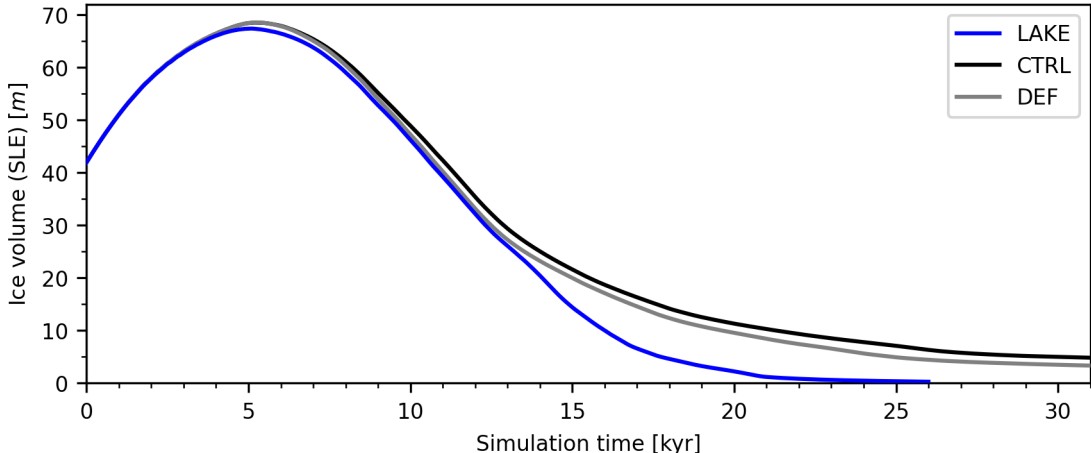

**Figure 4.** Temporal evolution of the ice volume of the continental LIS in the different experiments. The spatial extent that is attributed to the LIS is shaded in Fig. 3. Ice volume is given in sea level equivalent (SLE) units; see main text for definition. Note that both control experiments were run $5$ kyr longer than the *LAKE* experiment.

Figure 5 shows the temporal evolution of the ice margins and grounding lines at $90°$ W to compare the northward retreat of the LIS between the different experiments. The southward-shifted grounded ice margin between about $1$ and $3.5$ kyr is due to formation of ice lobes that advance into small lakes (compare also with Fig. 6). At $\sim 3.5$ kyr the ice margin is the furthest south, after which it rapidly retreats northward. The presence of a lake initiates the retreat about $1$ kyr before the control runs. Topography revealed by the retreating ice is left deeply depressed and is immediately occupied by proglacial lakes following the ice margin. From about $6$ kyr, an inner-continental ocean basin shows up in the PISM default experiment (*DEF*), where the topography is below sea level, which increases the ice retreat compared to the *CTRL* experiment, closely following the ice margin of the *LAKE* experiment.

Figure 7 shows a comparison of the *LAKE* and *CTRL* experiments at $7$ kyr. The lakes' impact on the ice sheet dynamics and geometry is obvious. Where the ice sheet margin terminates even in small proglacial lakes, the ice surface velocity anomaly (Fig. 7b) shows a strong increase. The increased ice flow and mass loss results in a drastically lowered ice profile (Fig. 7a) and accelerated retreat of the ice margin. These effects can be traced back several hundreds of kilometers upstream.

From $7$ to $9$ kyr, the Great Lakes region is covered by a single proglacial lake, which shares an up to $2000$ km long boundary with the ice sheet (compare with Fig. 8a). At around $9$ kyr the water level of this lake rapidly dropped, as a lower outlet to the Atlantic became ice-free. The successor lake occupies the basin of Lake Agassiz/Ojibway (see Fig. 8b and c). Where the water depth is sufficient, the ice floats. Figure 9 shows profiles of the ice margin of the *LAKE* and the two control experiments at $13$ kyr along the $90°$W transect. Note that only a few kilometers behind the grounding line, the bed elevation declines into the Hudson Bay basin. As soon as the grounding line retreats into this deep basin, the lake quickly expands underneath the ice sheet (compare with Fig. 5 and Fig. 8c and d), forming an enormous ice shelf. This dramatically accelerates the glacial

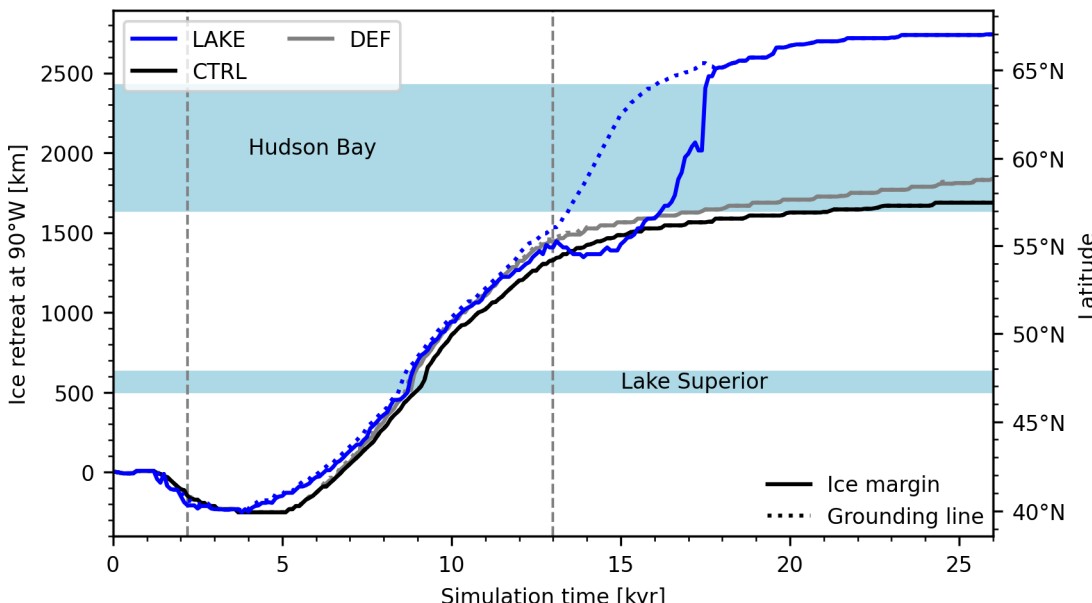

**Figure 5.** Temporal evolution of the ice margins and grounding lines at 90° W for the different experiments. Ice retreat is relative to the initial LGM ice front position. The vertical dashed lines at 2.2 and 13 kyr highlight the timing of the plots in Figs. 6 and 9, respectively. To provide some geographical context, the locations of Lake Superior's and Hudson Bay's contemporary basins are marked by the blueish stripes.

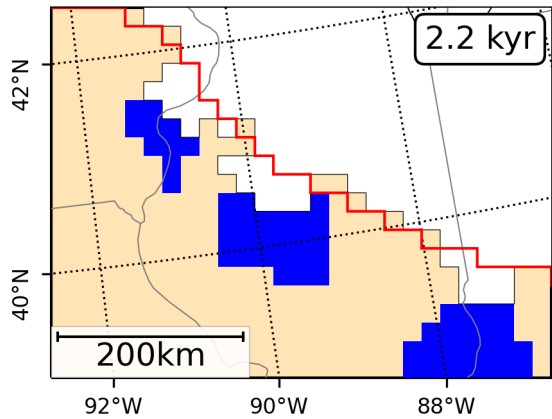

**Figure 6.** Formation of ice lobes at the southern ice margin (at 2.2 kyr) where lake boundaries promote accelerated ice flow. Ice extent of the control experiment is shown in red. The pixelated appearance of this plot comes from the 20 km model resolution.

retreat over Hudson Bay, which can also clearly be seen in the ice volume evolution of the LIS (Fig. 4), as the lake and no-lake experiments rapidly diverge.

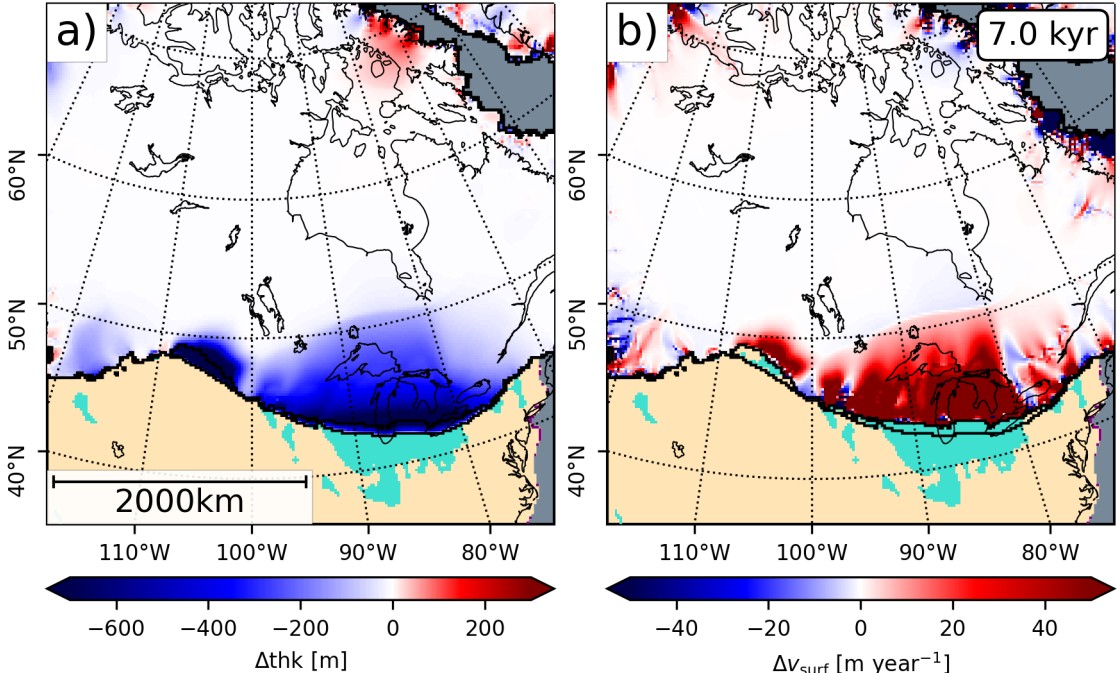

**Figure 7.** Comparison of the *LAKE* and *CTRL* experiments at 7 kyr. Both plots show anomalies (*LAKE - CTRL*) of (a) ice thickness and (b) surface velocity. Acceleration of ice flow at the proglacial lake boundaries and the resulting impact on ice geometry can clearly be seen. Ice margins of both simulations are shown in black.

At the southern ice margin and especially at the ice shelves, where the surface elevation is low, the surface runoff is greatly increased (see Fig. 10a). Figure 10b shows the modeled mass flux due to sub-shelf melting and calving. In this region, surface runoff contributes most to mass loss.

At around 17.9 kyr, the ice saddle over Hudson Strait breaks apart and allows the lake to drain into the Labrador Sea (Fig. 8e). We want to note that the sudden jump of the ice margin ($\sim 500$ km), seen at this time in Fig. 5, stems from the ice margin vanishing laterally out of the 90°W profile. Due to GIA processes, an ice-free Hudson Bay basin eventually emerges above sea level (see Fig. 8e and f). For both control experiments, the Hudson Bay Area is still covered by a thick ice dome.

The drainage route towards the Arctic is blocked until the ice saddle connecting the northern LIS and the Cordilleran Ice Sheet (CIS) collapses. Along the southern margin of this ice saddle, a vast proglacial lake forms (see Fig. 8d and e). At around 21 kyr, the saddle collapses, which drains most of the lake (Fig. 8f). For the control experiments, *DEF* and *CTRL*, this separation occurs at about 24 kyr and 26 kyr, respectively. This event marks the final retreat of the LIS in the *LAKE* experiment (Fig. 4).

## 4   Discussion

### 4.1   Lake reconstructions

The reconstruction of palaeo proglacial lakes depends mainly on GIA and ice margin locations, as these define the extent and depth of lake basins. Resolution issues are assumed to be secondary in this context and are tackled by applying the prepro-cessing method described in Hinck et al. (2020). Because of the simplified model setup, the ice margin retreat modeled in our experiments does not match well with reconstructions based on geological evidence. For this reason, the lake reconstructions are not expected to match well with observations. Drainage towards the Arctic, for example, is blocked until the ice saddle connecting the LIS and the CIS collapses, which results in a large proglacial lake in western Canada between 15 and 21 kyr. This lake is located where glacial lakes McConnell (Smith, 1994) and Peace (Mathews, 1980; Hickin et al., 2015) existed, but its basin does not match the outlines of the palaeo lakes (see Fig. 8d and e).

Another issue, which is also partly related to drainage, is the immense size and depth of some lakes that appear along the southern ice margin. From around 6 to 9 kyr one large lake occupies the entire Great Lakes region. Later, it transitions into the basin of Lake Agassiz/Ojibway (Teller and Leverington, 2004) before expanding into Hudson Bay ($\sim 13 - 18$ kyr) (Fig. 8a-d). Maximum water depths close to the grounding line are up to 1000 m. As a result of the increased accumulation of ice due to the simplified climate forcing, the topography is further depressed.

### 4.2   Lacustrine impact on ice dynamics

In comparison with the *CTRL* experiment, the modeled ice dynamics reveals a high sensitivity to the presence of proglacial lakes. The combination of modified stress regime, changed ice geometry and increased frontal ablation leads to accelerated ice flow upstream of the lake boundary. This acceleration and the associated lowering of the ice surface is shown in the ice thickness and surface velocity anomaly plots of Fig. 7. It should be noted that some features in these plots also arise from the fact that the ice geometries of the experiments diverge. Sensitivity runs (see supplementary material) have shown that the observed response is strongly impacted by the grounding line treatment. If the basal resistance at the grounding line is not reduced, the grounding line flux is strongly decreased. This becomes aparent in reduced mass loss and slower retreat of the ice margins. Nevertheless, when compared to the *CTRL* experiment, the impact on ice flux and margin retreat is clear.

Another feature that can be observed in the model results is the formation of ice lobes (Fig. 6). Due to the enhanced sliding at the lake boundary an advance is locally promoted. However, these lobes only appear at shallow lakes and are not comparable in size with major ice lobes of the LIS (e.g. Hooyer and Iverson, 2002; Dyke, 2004). When water depth of the proglacial lake becomes too deep, the thin advancing ice front is lost due to calving. A calving law adapted for lakes, as mentioned in Sect. 2.4, could help improve modeling this aspect. The application of a more realistic GIA model may also aid in the development of these lobes, as the lake depth at the grounding line margin will be reduced.

The most drastic impact on the glacial retreat can be observed at a later stage, when the grounding line enters the deeply depressed basin of Hudson Bay (Fig. 8c - e). Since there is a reverse sloping bed, no stable grounding line position can be achieved (see Fig. 5), leading to an increase of ice flux into the expanding lake basin. This process is similar to the marine

ice sheet instability (MISI, Weertman, 1974; Thomas and Bentley, 1978; Schoof, 2007). Contrary to MISI, where the ice loss is generally driven by calving and sub-shelf melt processes at the ice shelf, we observe a dominance in ice loss via surface runoff (see Fig. 10), due to the strong surface-elevation feedback in the warm climate. At the ice-shelves, the surface runoff is so large that when comparing with an experiment that has a reduced calving threshold, the shelf geometry hardly differs. The ice that is not calved off is subject to strong melting (see sensitivity run *RedCalv* in the supplementary material). However, we want to stress here that changes in local climate due to the presence of the lake might weaken this process (Krinner et al., 2004; Peyaud et al., 2007). This proglacial lake ice sheet instability (PLISI) described here, was also reported by Quiquet et al. (2021). Finally, the PLISI results in the disintegration of the ice saddle blocking drainage through Hudson Strait (Fig. 8d - e).

We want to note that the vast ice shelves that can be observed at this late phase of the collapse of the LIS (Fig. 5 & 8c-d) might be unrealistic. At least we are not aware of any geological evidence or reconstruction, indicating such ice shelves existed. It is possible that the simulated grounding line flux is too high or the calving rate too low. Two sensitivity runs checking these parameters (*IncCalv* and *nSG* in the supplementary material) do not exhibit such large lacustrine shelves. However, more research is necessary to decide how to best parameterize these issues.

## 4.3 Implications on the demise of the LIS

Although the modeled glacial retreat of the North American ice sheets does not match up with reconstructions based on geological observations, the importance of proglacial lakes on the glacial retreat is evident. Comparing to the two no-lake experiments with the *LAKE* experiment, totally different ice geometries exist at the end of the runs. The presence of lakes triggers the rapid collapse of the LIS over Hudson Bay. Furthermore, it accelerates the disintegration of the ice saddle connecting LIS and CIS. Even after extending the simulations for both control runs by another 5kyr, Hudson Bay still remained ice-covered.

We further want to note that the glacial retreat seen in our results might be strongly accelerated due to our simplified experiment setup. The large ice sheet growth, caused by the simple climate forcing, leads to deeply depressed topography and thus deeper lake basins.

## 5 Conclusions

In this study we have described the implementation of an adaptive lake boundary condition into a 3D thermo-mechanically coupled ice sheet model. To the best of the authors' knowledge this has not been done in any other study. Application of this model to a simple North American deglacial scenario, starting at LGM conditions, shows the importance of proglacial lakes for the proper modeling of the ice dynamics.

The lake model promotes the acceleration of ice flow where the ice margin is bounded by a lake, which leads to a lowering of the upstream ice surface. In a warming climate, the lowered ice surface is exposed to higher temperatures and thus higher melt. Our simple experiments suggest that the mass balance at the southern ice margin of the LIS is governed by surface melt, while lacustrine calving and sub-shelf melting are secondary.

Once the lacustrine grounding line enters a deep basin on a retrograde bed, the ice sheet exhibits an instability similar to the MISI, which is called accordingly PLISI. As the ice thickness over the new grounding line position is higher, this increases

the grounding line flux, which forces the grounding line to retreat even further. This positive feedback loop is ongoing until a stable grounding line position is reached. Due to increased melting at the lowered ice surface, the feedback cycle is further accelerated.

In our experiments, the occurrence of the PLISI results in the final break-up of the LIS. The lake rapidly expands into the still glaciated basin of Hudson Bay, followed by a quick disintegration of the ice barrier that blocks the lake's drainage through

Hudson Strait into the Labrador Sea.

Due to the significant impact of the *LakeCC* model on the ice sheet dynamics, we would recommend the use of an adaptive lake boundary condition when modeling the deglaciation of land terminating ice sheets.

*Code availability.* All implementations described in this study are based on PISM v.1.2.1. The modified code can be found in SH's GitHub repository (`https://github.com/sebhinck/pism-pub`). Implementations of the various independent models are found in separate

branches. The code of the modified PISM version, as it was used in this study, is archived online (Hinck and PISM Authors, 2020).

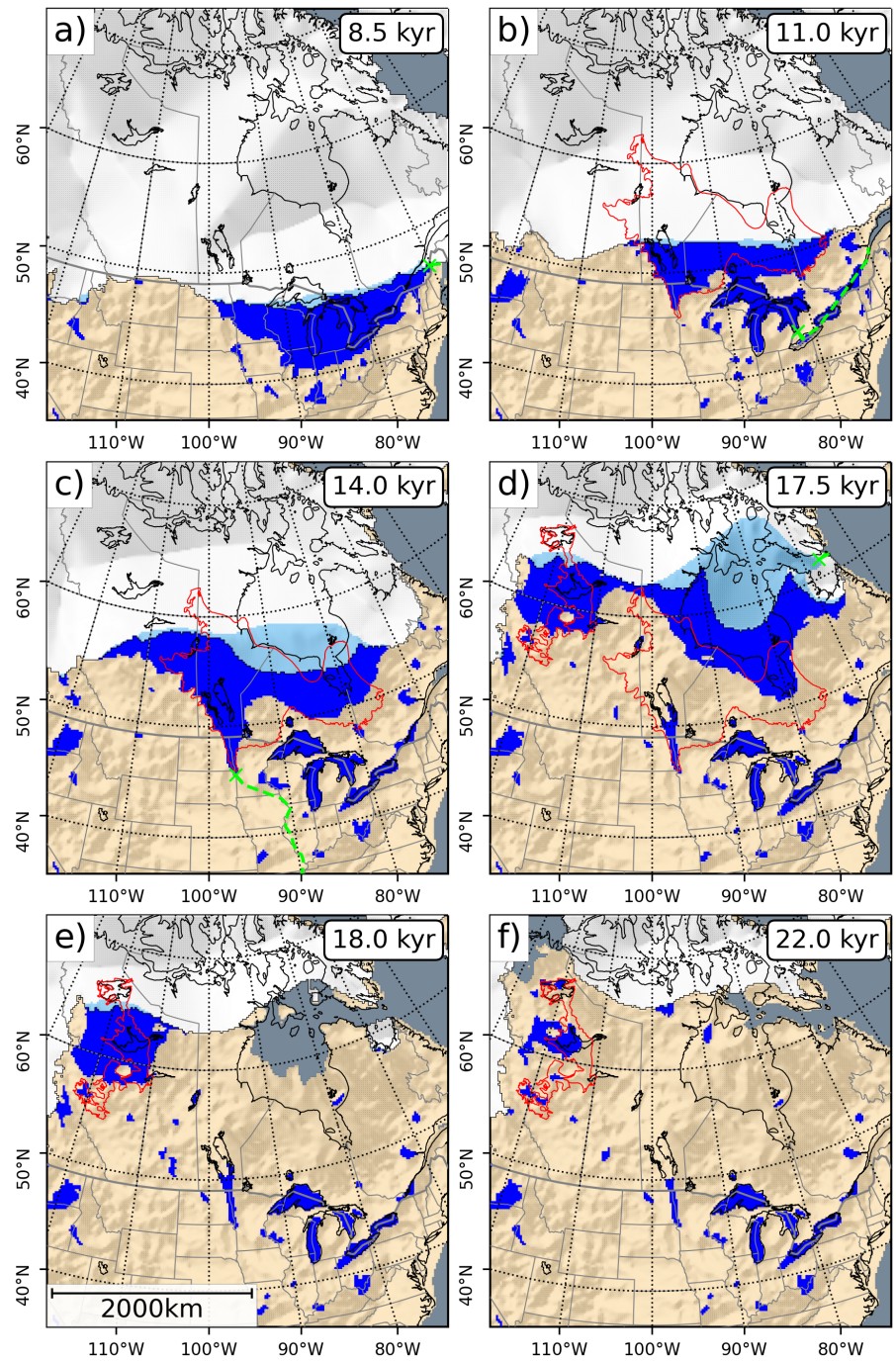

**Figure 8.** Snapshots for different stages of the *LAKE* experiment. Lacustrine ice shelves are shown in light blue. The red outlines show the maximum extent of selected palaeo lakes (Fig. 3). Modeled spill point and drainage routes for Lake Agassiz are shown in green in panels a-d.

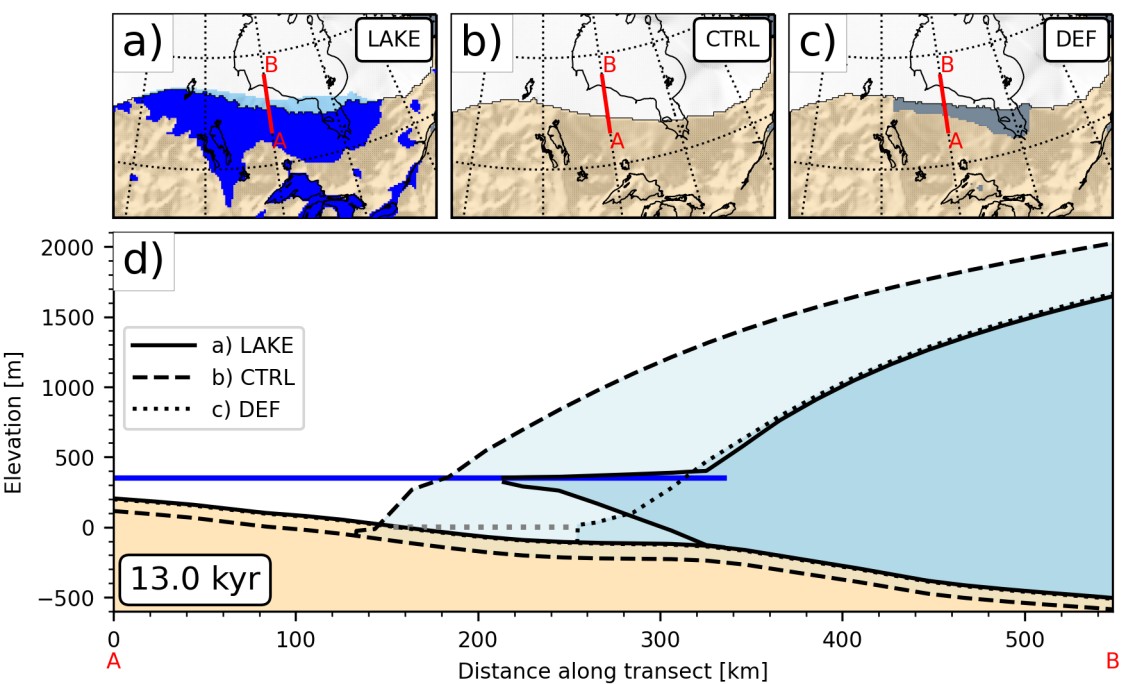

**Figure 9.** Profiles of the ice sheets of the (a) *LAKE*, (b) *CTRL* and (c) *DEF* experiments at 13 kyr. The transects shown in (d) are along 90°W, between 53°N (A) and 58°N (B) and are marked in the upper panels by a red line. In panels (a) - (c) ice shelves are shown in light blue, lakes in blue, and ocean in gray. The blue line in panel (d) marks the lake level elevation in the *LAKE* experiment, while the gray line marks the sea level elevation of the *DEF* experiment.

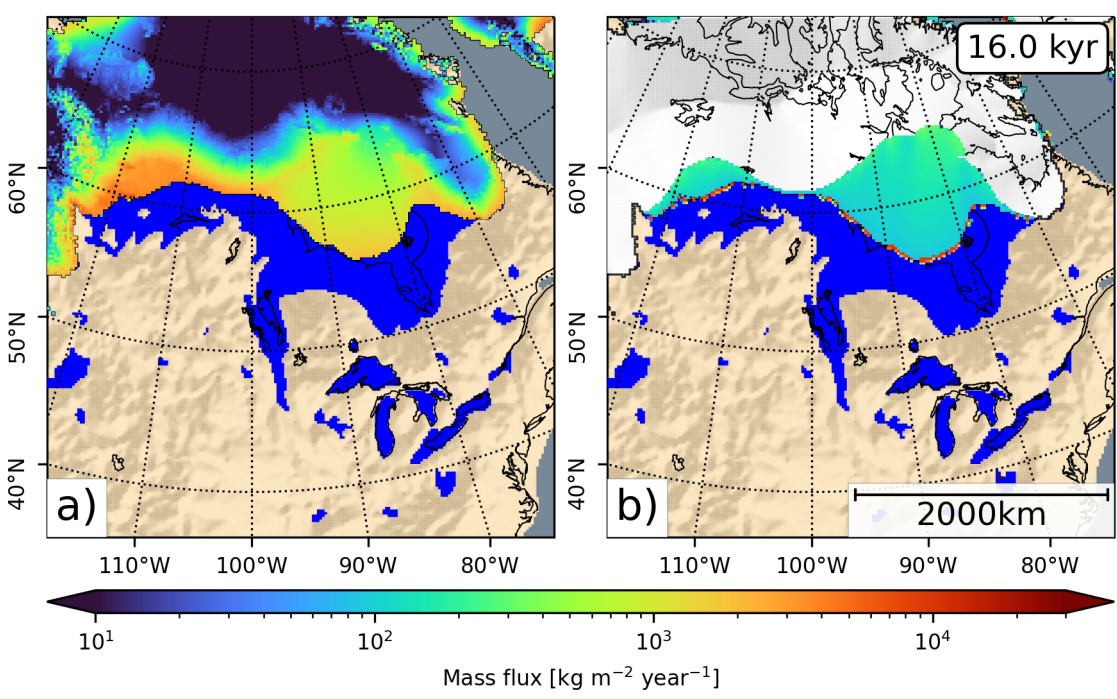

**Figure 10.** Comparison of the modelled mass loss due to a) surface runoff and b) sub-shelf melting and calving, shown here at 16 kyr. At the shelf regions, the surface runoff is about an order of magnitude larger than sub-shelf melting. Locally, at the shelf margin, the mass loss may be greatly increased due to calving events. The fields shown here are averaged over the ice model's reporting interval (here: 5 years).

## Appendix A: Experiments

### A1 Setup

In Section 2 all relevant parts of the ice sheet model, as used in this study, are explained. Here, a general description of the experimental setup is given. Aspects diverging between the different experiments are discussed in Sect. 2.5 and the supplementary
material.

For testing the impact of the proglacial lake boundary condition with PISM, we chose to run simplified experiments of the glacial retreat of the North American ice sheets after the LGM. The computational domain used for this study is shown in Fig. 3. Grid resolution is set to 20 km.

Experiments for this study were done using a modified version of the Parallel Ice Sheet Model (PISM; the PISM authors,
2015; Bueler and Brown, 2009; Winkelmann et al., 2011). The hierarchy of the parameterizations for PISM's different sub-models is listed in Table A1. Configuration parameters that diverge from PISM's default configuration are shown in Table A2.

**Table A1.** PISM sub-models used for the experiments.

| Coupler | Model(s) used |
| --- | --- |
| atmosphere | `index`[*], `precip_cutoff`[*] |
| bed_def | `lc` |
| calving | `eigen_calving`, `thickness_calving` |
| hydrology | `null` |
| lake_level | `(lakecc)`[*] |
| ocean | `pik` |
| sea_level | `constant`, `delta_sl(`, `sl2dcc)`[*] |
| stress_balance | `sia+ssa` |
| surface | `pdd` |

[*] Models added for this study.

Further information about PISM's sub-models and the default configuration can be found in the ice model's documentation. Initial conditions are taken from the NAICE palaeo ice and GIA reconstructions from Gowan et al. (2016). To obtain the palaeo topography, pre-computed bed deformation is added to the present day topography dataset RTopo-2 (Schaffer et al.,
2016). From the temporal evolution of the topography provided by NAICE, we use the uplift rates calculated from NAICE that are used to initialize the Lingle–Clark bed deformation model of PISM. Geothermal heat flux is considered constant in the domain over time and interpolated from the dataset of Davies (2013).

As described in Sect. 2.1, the atmospheric forcing model needs two climatic states between which the transient climate is interpolated. Both climate states are monthly means of the surface air temperature and precipitation fields from steady state
experiments with the climate model COSMOS. Details on these runs can be found in Zhang et al. (2013). The PD climate

**Table A2.** Configuration parameters used for the PISM experiments that diverge from the defaults.

| Option | Default value | Used value | Units |
|---|---|---|---|
| -lakecc_dz[*] | 1 | 5 | m |
| -lakecc_zmin[*] | 0 | −300 | m |
| -Lbz | 1000 | 2000 | m |
| -Lz | 4000 | 5750 | m |
| -max_dt | 60 | 0.25 | yr |
| -Mbz | 1 | 11 | - |
| -meltfactor_pik | 0.005 | 0.01 | - |
| -Mz | 31 | 101 | - |
| -pik | false | true | - |
| -precip_cutoff_height[*] | 3000 | 3500 | m |
| -pseudo_plastic | false | true | - |
| -sia_e | 1 | 5 | - |
| -stress_balance.sia.max_diffusivity | 100 | 200 | $\mathrm{m^2\,s^{-1}}$ |
| -tauc_slippery_grounding_lines | false | true | - |
| -temp_lapse_rate | 0 | 7.9 | $\mathrm{K\,km^{-1}}$ |
| -thickness_calving_threshold | 50 | 200 | m |
| -till_effective_fraction_overburden | 0.02 | 0.01 | - |
| -use_precip_cutoff_height[*] | false | true | - |

[*] Commandline options added by one of the models described in this study.

state is the control run from that reference, while the LGM experiment is described in Hossain et al. (2018). It is determined using the same protocol, but uses the LGM reconstruction from Gowan et al. (2016). When using a glacial index derived from the NGRIP $\delta^{18}$O measurements (North Greenland Ice Core Project Members, 2007), the modeled deglaciation is too rapid to identify contributions from the lake model. Instead, the deglacial climate signal was crudely approximated by a linear model

(see Fig. A1a). The transition from LGM to PD climate takes place over 12 kyr (e.g. from 21 to 9 kaBP). To prevent ice sheet growth in eastern Siberia and above 3500 m elevation (relative to PD sea level), the precipitation is set to zero in these regions. The maximum elevation threshold was chosen in an ad-hoc fashion. However, current LGM reconstructions (e.g. NAICE (Gowan et al., 2016), and ICE-6G (Peltier et al., 2015)) exhibit higher surface elevations only in few mountainous regions.

Transient sea level forcing is applied accordingly to the glacial index. It is linearly increased from −120 m at the LGM to

the contemporary level 0 m (see Fig. A1b).

Before running the experiments, the model needs to be spun-up. This is done by keeping all boundary conditions fixed at LGM conditions and running PISM in a fixed-geometry mode for several thousand years until the temperature field within the ice has equilibrated. The final output is then used to start the experiments.

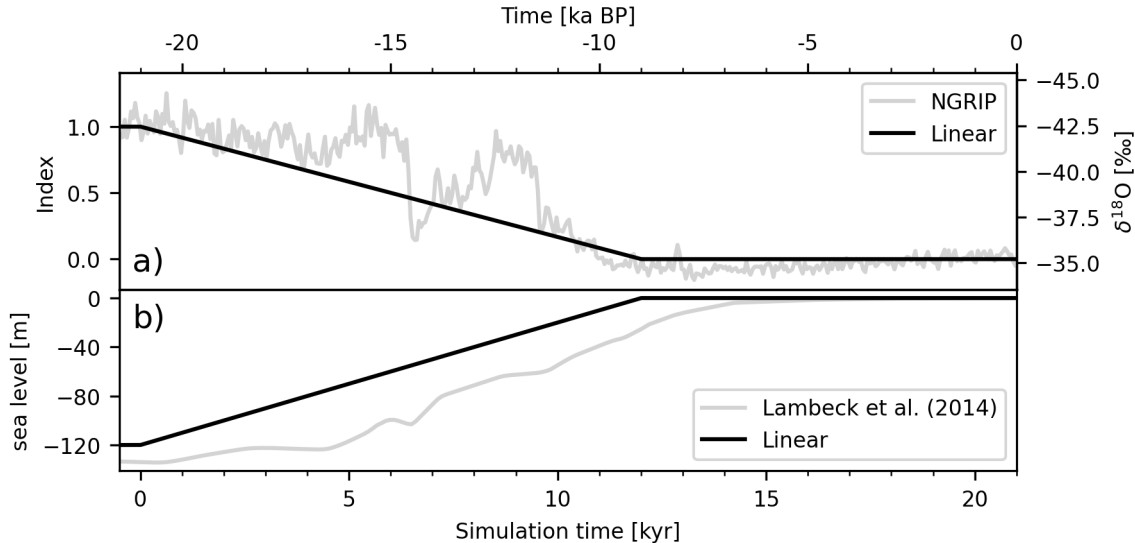

**Figure A1.** The black lines show the temporal evolution of the (a) glacial index and (b) the mean global sea level used in the experiments. Both time series are simplified by a linear evolution from their LGM to PD values between 21 and 9 kaBP and represent a crude approximation of proxy records. For visualization, such proxy records are shown in gray. These are (a) the index derived from the NGRIP $\delta^{18}$O measurements (North Greenland Ice Core Project Members, 2007), and (b) the global mean sea level reconstruction from Lambeck et al. (2014).

For all experiments, the maximum time step of the ice sheet model was limited to $0.25$ yr. We want to stress that this setting is not requisite by the *LakeCC* model, but we had several reasons to do so. Estimating the computational overhead of an algorithm (as done in Sec. C) requires the time steps of the compared runs to be similar. Another benefit from limiting the time step is that the occurrence of numerical instabilities due excessively large jumps in water level (see Sec. 2.4.3) is potentially reduced.

## A2   Limitations

In the following, we will mention and shortly discuss some limitations of the experimental setup.

### A2.1   Climate model

The climate forcing is relatively simple in our setup. Using a glacial index leads to increased mass accumulation in cold regions and on top of the ice sheet. The experiments therefore suffer from excessive ice sheet growth after model initialization.

Furthermore, the presence of vast proglacial lakes would impact the local climate by reducing temperatures and increasing precipitation patterns (Krinner et al., 2004; Peyaud et al., 2007). This could locally increase the ice sheet's SMB and counteract the accelerated mass loss observed in this study. Also, the potential impact on ocean circulation and global climate due to redistribution of freshwater (Broecker et al., 1989; Teller et al., 2002; Condron and Winsor, 2012) is ignored here.

## A2.2 GIA model

PISM's default GIA model (Lingle–Clark; Bueler et al., 2007) is based on a simple two-layered Earth model and therefore lacks viscosity variations between the upper and lower mantle. This variation significantly contributes to the GIA signal in central Canada (Wu, 2006). In combination with the excessive mass accumulation due to the simple climate forcing, our results show a strongly depressed topography, with deep lake basins that only slowly relax. For a realistic simulation of the LIS deglaciation, with a proper representation of proglacial lakes, a more advanced model to calculate GIA signal would be needed.

Another feature missing in the GIA model is self-gravitational effects of the ice sheet. The ice sheet's mass impacts the geoid, along which the free water surfaces align. As a result, the lake water is attracted towards the ice sheet and the water depth at the grounding line would potentially increase. Due to lack of an appropriate model of gravitational change, we can not estimate the potential magnitude of this effect. However, according to James et al. (2000) the effect is secondary compared to the crustal deformation.

Furthermore, in the calculation of the GIA signal we do not include the mass held by the lakes. We would expect the water mass to have a significant impact on the GIA and thus also on the lake basins.

## A2.3 Model initialization

Initialization of the Lingle–Clark model from a glaciated state is problematic here. For the model to calculate a relief topography, to which bed deformation is applied, it should ideally be initialized from an interglacial state when the residual GIA from previous glaciations is limited. Test runs, comparable to Niu et al. (2019), however, suffered from the fact that the bed deformation along the southern ice margin was so deep, that the basin was connected to the Atlantic Ocean, which consequently inhibited the formation of lakes. We therefore chose to initiate the experiments from NAICE LGM reconstructions. The mismatch in calculated relief topography, results in ice free regions to over-relax. Hudson Bay, for example, is elevated above sea level at PD (see Fig. 8f).

## Appendix B: PISM-LakeCC

This Appendix adds a more technical overview of the model description of the main text. At first, a short description of the general `lake_level` interface within PISM is given. As the LakeCC model, and also the SL2DCC model, make extensive use of functions based on the connected components (CC) algorithm (see also Hinck et al., 2020), its implementation is delineated in the following section. The last two sections address details of the implementation of the LakeCC and SL2DCC models. The names of model components in PISM and of configuration parameters are shown in monospaced font.

### B1   PISM's lake interface

PISM is designed in a modular way, that prescribes the interface of all sub-models. These modules inherit certain properties and functionality from *C++* base classes and extent upon these. This way, different combinations of sub-models can be easily realized via configuration parameters. Furthermore, this facilitates the implementation of new sub-models. For the sub-systems acting as boundary conditions on the ice-sheet, such as atmosphere, surface and ocean, PISM distinguishes between models

and modifiers. A model parameterizes all aspects needed for the treatment of the respective boundary, and can be combined with combination of different modifiers, which can apply changes to the model's parameterization.

A lake, as seen by an ice sheet model, is very similar to a locally elevated sea level. It would be possible to implement the lake interface as a sea level modifier in PISM. However, physical differences (e.g. density, temperature distributions) require different treatment of marine and lacustrine environments. To distinguish between both cases, at least two spatial maps are

required. This can either be done by providing the lake and sea level combined in one field and additionally providing a mask, or providing lake and sea level elevations as separate maps. For the implementation of the PISM-LakeCC model, we chose the latter case.

The `lake_level` interface is based on PISM's `sea_level` interface. It provides a 2D field `lake_level_elevation` to the ice sheet model, which contains each cell's lake level elevation in meters above the reference geoid. Where no lake level

is set, the cell is set to invalid, and a fill value is inserted, which is a large negative number (it is set by the configuration parameter `output.fill_value`, which has a default value of $-2 \cdot 10^9$). If no lake model is chosen, the entire returned field is set to invalid.

As the ice sheet and lake geometry is dynamic, it turned out that it is beneficial to spread information about the presence of lakes to neighboring cells. Lake models can therefore ask the lake interface to expand lakes one cell beyond their reconstructed

basin, which can be regarded as kind of ghost cells. This does not change the actual lake geometry within the ice sheet model, because cells are considered dry, where the water level is below bed elevation or the ice sheet is grounded. The lake level elevation field can therefore rather be regarded as a virtual lake level.

Where the lake level is valid and above bed elevation, a grid cell is treated as a lake. The lacustrine boundary is essentially treated the same as the ocean, except of the different water level and adapted water density. A more advanced boundary treat-

ment would need to be implemented for ocean (or rather wet boundary) and calving parameterizations. So far, only the thickness calving mechanism accepts a distinct lacustrine parameter for the thickness threshold (`calving.thickness_calving.`

`threshold_lakes`). Note, contrarily to marine ice shelves, lacustrine ice shelves are not excluded from the calculation of the potential sea level rise potential.

For restarting, the variable `effective_lake_level_elevation` is added to each restart file. Optionally, the two 2D diagnostic variables `lake_depth` and `lake_level_real` can be added to PISM's output (see Table B1). Outside where the lake level is actually above bed elevation, both fields are set invalid.

**Table B1.** Newly added diagnostic variables. Some diagnostic fields are generally available, independent of the sea-level or lake model chosen.

| Name | Model | Description |
| --- | --- | --- |
| `lake_depth` | general | Water depth of lakes. |
| `lake_level_real` | general | Water level of lakes, where above bed elevation. |
| `lakecc_gradual_target` | LakeCC | Field `target_level`, as used internally by the LakeCC model. |
| `ocean_depth` | general | Water depth of the ocean, where sea-level above bedrock. |
| `sl2dcc_gradual_target` | SL2DCC | Field `target_level`, as used internally by the SL2DCC model. |

## B2 Connected Components

PISM-LakeCC and PISM-SL2DCC make extensive use of the connected components algorithm. The algorithm and its use in the LakeCC standalone model is described in Hinck et al. (2020). Parts of the models are implemented analogously, i.e. the algorithm is used to determine the target lake level (using the LakeCC method) and the ocean mask (using the SL2DCC method). However, implementation into a dynamic ice sheet model needs a more advanced treatment than the standalone tool.

In several parts of the algorithm, non-local information of the entire lake is needed. The connected component algorithm, when adapted accordingly, is capable of gathering different information and attributing it to the respective lake basins. In the following we shortly introduce the *C++* classes based on this algorithm, which are used in the implementation of the PISM-LakeCC and PISM-SL2DCC models.

The `LakeLevelCC` class determines the maximum fill height of lake basins by iteratively checking the entire domain for a set of increasing water levels, as described in Hinck et al. (2020). The `SeaLevelCC` class is also implemented as described in that reference. It identifies basins in the topography, which are below the global sea level but are not connected to either of the domain margins (compare Fig. B1a). The `IsolationCC` class (Fig. B1b) also marks cells as invalid that are either ice-covered or not connected by an ice-free corridor with the domain margin. It is used to restrict the formation of lakes in the ice sheet interior and of subglacial lakes where thin ice covers a deep basin. To remove narrow lakes, which are often related to under-resolved topography, the `FilterLakesCC` class checks the lakes' geometry. Only lakes, which contain one (or more) cells that have at least a certain amount of neighbors that also are part of that lake, are retained (Fig. B1c). `LakePropertiesCC` collects the minimum and maximum current water level of each lake basin (compare Fig. B1d). The

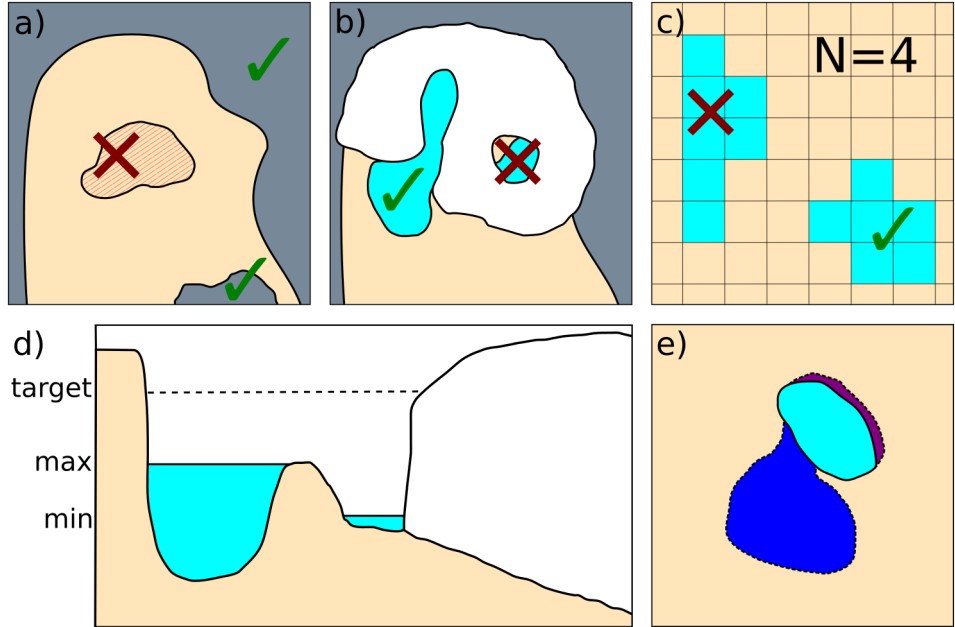

**Figure B1.** Sketches depicting the different algorithms implemented. a) `SeaLevelCC` – only basins below sea level and connected to either of the domain margins is considered ocean. b) `IsolationCC` – Lake basins fully enclosed by the ice are considered invalid. c) `FilterLakesCC` – Lakes are only retained if they contain at least one cell with a certain amount of lake neighbors. In the example $N = 4$ was chosen. d) `LakePropertiesCC` – This algorithm collects the minimum and maximum water levels for each lake basin. e) `FilterExpansionCC` – This algorithm compares the current lake mask with the previous mask and labels newly appeared and vanished lake cells. It distinguishes between narrow strips (purple) and wider basins (dark blue).

class `FilterExpansionCC` (Fig. B1e) is used to compare the new lake basins with the state of the previous time step. This method returns a mask that marks cells that were newly added or have vanished, but it also distinguishes the basin shape, similarly to `FilterLakesCC`. Furthermore, for each new lake basin the minimum bed elevation and, if it was an ocean basin in the previous time step, sea level are returned. This information is needed to catch different scenarios when initializing the lake level and treat them accordingly.

The difference of the implementation of the underlying connected components algorithm to Hinck et al. (2020) is the use of PISM's parallelized data types. 2D fields are partitioned and distributed over several processors using the library *PETSc* (Balay et al., 1997, 2019). This implementation requires some extra steps, as processors need to exchange information with the adjacent processor domain.

## B3 LakeCC

The PISM-LakeCC model utilizes the LakeCC algorithm (Hinck et al., 2020) to determine the maximum water level of lake basins. The topographic and glacial state, as needed for the lake reconstruction, is provided by PISM. Options used by the model are set by configuration parameters (see Table B2) and are explained in the following. Cells with ice thinner than $\delta_{if}$ are considered ice-free.

Table B2. Configuration parameters read by the LakeCC model. Prepend `lake_level.lakecc.` to the parameter to get the full name.

| Parameter | Type | Default | Symbol | Description |
| --- | --- | --- | --- | --- |
| dz | Number | 1m | $\Delta z$ | Spacing between successive water levels. |
| filter_size | Integer | 4 | $N_{\text{filter}}$ | Number of neighboring cells used by filter algorithm. See text for details. |
| ice_free_thickness | Number | 10m | $\delta_{\text{if}}$ | Threshold ice thickness below which a cell is considered ice-free. |
| init_filled | Boolean | false | - | Bootstrap lakes as filled. |
| keep_existing_lakes | Boolean | true | - | Keep existing lakes, even though they are surrounded by ice. |
| max_fill_rate | Number | $1\,\text{m}\,\text{yr}^{-1}$ | $\gamma$ | Fill rate used by LakeCC gradual fill algorithm. |
| topg_overlay_file | File | - | - | File containing field `topg_overlay`. |
| zmax | Number | 1000m | $z_{\text{max}}$ | Maximum water level to check. |
| zmin | Number | 0m | $z_{\text{min}}$ | Minimum water level to check. |

At initialization of the model, the lake level is read from the PISM input file (`effective_lake_level_elevation`). This is necessary to guarantee a smooth continuation of the simulation after model restart. When this field is not available, lakes are either initialized empty, or filled to the brim (configuration flag `init_filled`).

In every time step of the ice sheet model, the ice and topography configuration changes, thus the lake model needs to be updated. Fig. B2 shows a sketch of the update sequence of the PISM-LakeCC model. The basic principle of how the model works was given in Sect. 2. Both fields, target level and lake level are successively refreshed.

The process to update the target level (Fig. B2b) resembles that described in Hinck et al. (2020). If a filename is provided using the `topg_overlay_file` option, the field `topg_overlay` is read and applied to PISM's topography before proceeding. Using this processed topography field and PISM's sea level elevation data (which should have been computed using the SL2DCC model) a mask is created that marks all the ocean cells. Another mask is prepared using the `IsolationCC` method. It is called and marks cells as valid if they are ice-free and are connected by an ice free corridor to the domain margin. The LakeCC method only keeps lakes that contain at least one valid cell. This prevents isolated lakes from forming in the ice sheet interior or entirely beneath the ice. It might, however, happen that lakes suddenly marked as invalid when the ice sheet advances and cuts them off from the exterior. If this is not desired, the option `keep_existing_lakes` labels all exiting lakes as valid. With everything prepared, the lake basins are computed using the LakeCC algorithm and their maximum water level is kept as the target level. The water levels that the algorithm iteratively checks are equally distributed between $z_{\text{min}}$

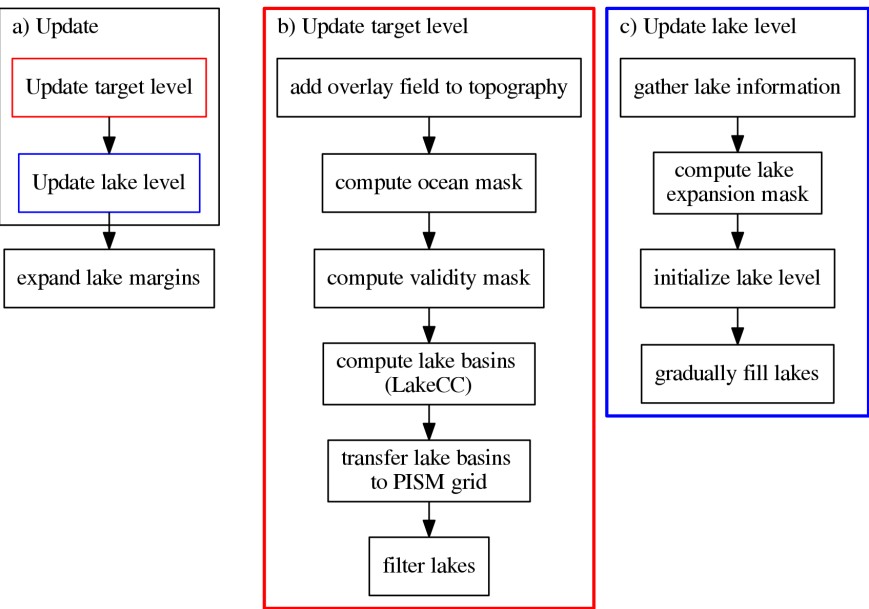

**Figure B2.** (a) Flowchart diagram showing the steps of the Pism-LakeCC update process. The red and blue boxes expand to the diagrams on the right, (b) and (c), with correspondingly colored outline.

and $z_{\mathrm{max}}$, with a spacing of $\Delta z$. As the target levels are computed using the modified topography, the lake basins have to be transferred back onto PISM's topography. Finally, to get rid of narrow lakes, which are often caused by the under-resolved topography, the target level is filtered by applying the `FilterLakesCC` method using a filter size of $N_{\mathrm{filter}}$. The target level can be added to the output for diagnostic purposes (see Tab. B1).

     In the next step, the lake level, which is the field that is accessed by other parts of the ice sheet model, is updated (Fig. B2c).
Before gradually filling each basin to its maximum fill level, which is set by the target level, newly added lake cells need to be initialized. Therefore, information about existing lakes within the lake basins and about patches of newly added lake cells is collected using the `LakePropertiesCC` and `FilterExpansionCC` methods. In the initialization process (sketched in Fig. B3), these data are used to ensure that all cells of a lake start at the same water level. However, this is not always possible. New lake basins should ideally be initialized empty, with a water level at its deepest point. Another case is when an existing
lake is enlarged by just a few cells (e.g. because the ice margin retreats). This gap should be initialized at the actual lake level. To identify this case, the `FilterExpansionCC` method considers the shape of the newly added patch similarly to the `FilterLakesCC` method: if at least one cell of the patch is entirely surrounded by other cells, also belonging to that patch, it is marked as 'wide', otherwise as 'narrow'. To omit instabilities due to rapid change of the water level, 'wide' basins are treated as new lakes and initialized empty. In case the lake basin was previously part of the ocean, the lake level is initialized at
the sea level, to omit a rapid jump in the boundary conditions. Once the lake level is initialized, the water level can be gradually adapted towards the target level using the constant fill rate $\gamma$, as described in the main text.

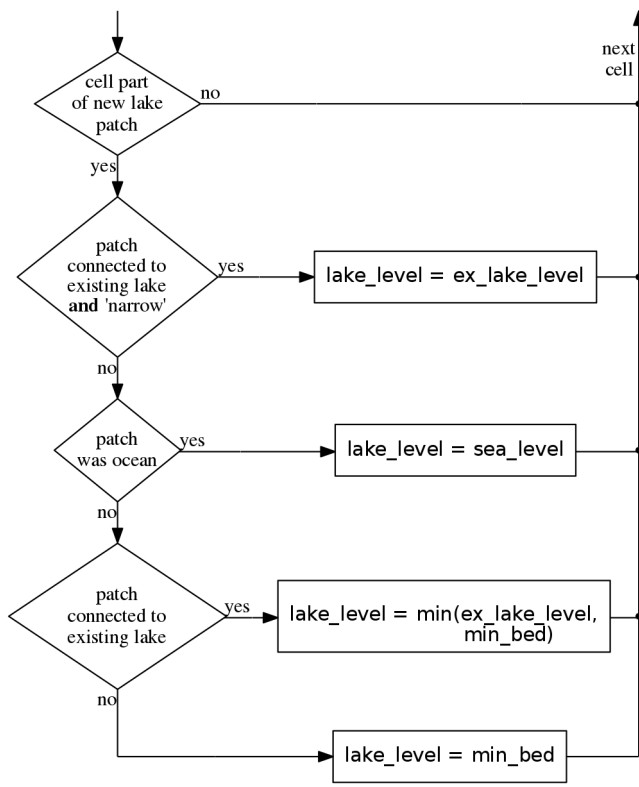

**Figure B3.** Flowchart diagram of the initialization process, which is executed for each cell of the grid. Cells that have just been assigned a lake level (i.e. it was set invalid in the previous time step) are initialized. Interconnected clusters of these cells are grouped as patches by the `FilterExpansionCC` method and are labeled accordingly to their shape ('narrow' or 'wide', see main text). Lake level is initialized either to the minimal bed elevation of the new basin (`min_bed`), to the lake level of a connected existing lake (`ex_lake_level`) or to sea level (`sea_level`).

## B4  SL2DCC

**Table B3.** Configuration parameters read by the SL2DCC model. Prepend `sea_level.sl2dcc.` to the parameter to get the full name.

| Parameter | Type | Default | Symbol | Description |
|---|---|---|---|---|
| `max_fill_rate` | Number | $1\,\mathrm{m\,yr^{-1}}$ | $\gamma^{\mathrm{SL}}$ | Fill rate used by SL2DCC gradual fill algorithm. |
| `sl_offset` | Number | 10m | $\delta_{\mathrm{SL}}$ | Sea level offset. |
| `topg_overlay_file` | File | - | - | File containing field `topg_overlay`. |

In PISM the sea level field is provided as a 2D field, which usually is set to the global mean sea level. Here, we present the implementation of a sea level modifier, which takes advantage of the possibilities of a spatially variable sea level field and removes isolated ocean basins. The model is based on the connected components (CC) algorithm as it was described in Hinck et al. (2020) and Sect. B2. Using the `SeaLevelCC` class a mask is determined with all isolated ocean basins marked. These basins are removed from the sea level field by setting the respective cells to a fill value, analogously to the lake interface described in Sect. B1. The configuration parameters read by the model are listed in Table B3.

Similarly to the `LakeCC` model (Sect. B3), the low-resolution topography field from the ice sheet model can be overlain by an input field read from `topg_overlay_file`. These fields are then used by the `SL2DCC` model to identify isolated ocean basins. Potential ocean cells (cells that fulfill the flotation criterion) are grouped into inter-connected patches using the `SeaLevelCC` method. Only patches that are connected to the margin of the computational domain are considered to be part of the ocean. All other patches are treated as isolated inland ocean basins. When categorizing all ocean basins, the `SeaLevelCC` method internally raises the water level by an offset $\delta_{SL}$. It is only used internally to raise the water level when checking for connected ocean basins. This treatment effectively extents the ocean mask slightly beyond the coastline. Basins near the coast, which would otherwise be labeled as isolated, are identified as regular ocean basins. This is done to account for the relatively low resolution topography.

Defining sea level spatially constant over the entire domain has the advantage that no sudden jumps in water level occur, that cause numerical problems. Using the SL2DCC model, however, ocean basins are dynamically added and removed, depending on geometric considerations using the evolving glacial topography. For this reason, the implementation of the SL2DCC model needs to take care of smoothly applying changes to the water level. When using the LakeCC model (Sect. B3), the lake model takes care of this. If the LakeCC is not used, the SL2DCC model gradually adjusts the water level using a constant fill rate $\gamma^{SL}$. This algorithm is very similar to the gradual filling algorithm used in the LakeCC model.

## Appendix C: Efficiency of the PISM-LakeCC model

Figure C1 shows the efficiency for all four experiments. Usage of the LakeCC model reduces the efficiency by $\sim 40\%$ compared to the PISM default run. However, as the numbers shown in the figure depend strongly on the adaptive time steps chosen by PISM, the plot does not show the pure computational burden of the LakeCC model. At times when ice shelves are present, smaller time steps are chosen, which can directly be seen in the model's efficiency. This is the reason why the *Ctrl* run is generally more efficient than the *Def* run, although an additional model needs to be evaluated: as no inland ocean basins are

present, no ice-shelves appear and ice velocities in general are slower.

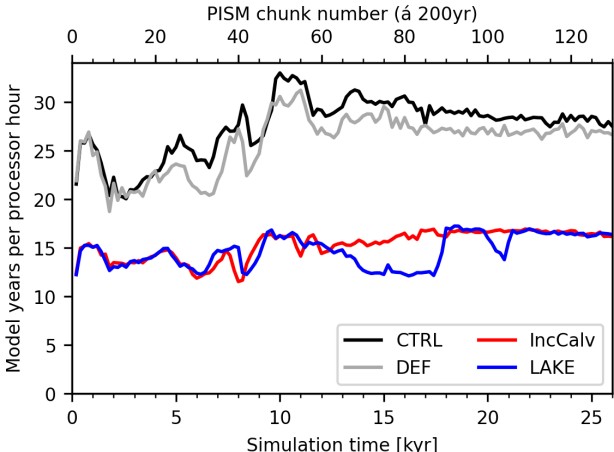

**Figure C1.** The plot shows the efficiency of PISM in model years per processor hour averaged over 200 yr chunks for the different experiments: *CTRL* – no lakes but with the SL2DCC model to prevent inland ocean basins; *DEF* – PISM default, no lakes, no advanced ocean treatment; *LAKE* – standard lake experiment; *IncCalv* – lakes, but increased calving threshold such that ice shelves are effectively removed (for more details see supplementary material). Variability is mainly due to the adaptive time stepping mechanism of PISM which hits especially hard when big ice shelves are present. A maximum time step of $0.25$ yr is used for all experiments.

## Appendix D:  Climate forcing

### D1    Glacial index

**Table D1.** Command-line options for use with the PISM atmosphere model `index`.

| Command-line option | Comment |
| --- | --- |
| `-atmosphere_index_file` | File containing climate index time series (`glac_index`). |
| `-atmosphere_index_climate_file` | File containing climate forcing fields for both climate states (`i` needs to be replaced with the corresponding index $(0, 1)$): air temperature (`airtemp_i`), precipitation (`precip_i`), surface elevation (`usurf_i`). |
| `-temp_lapse_rate` | Temperature lapse rate $\gamma_T$ used in Eq. (D1). Default value: $0.0\ \mathrm{K\,km^{-1}}$ |
| `-atmosphere.precip_exponential_` `factor_for_temperature` | Exponential factor $C$ used in Eq. (D2). Default value: $7.04167 \cdot 10^{-2}\ \mathrm{K^{-1}}$ |

The climate forcing applied for the test run carried out for this study uses a custom implementation that is based on the concept of a glacial index. It approximates the temperature $T_i$ and precipitation fields $P_i$ between two climate states, for example between LGM and PI. For simplicity, it is assumed that the spatially varying fields can be linearly interpolated between these states, which are each assigned an index of $0$ and $1$, respectively. Interpolation between these states is done by weighting them according to a time-dependent climatic index $i(t)$. It is usually derived from measurements of, for example, $\delta 18O$ (e.g. NGRIP (North Greenland Ice Core Project members, 2004), EDML (Augustin et al., 2004)), which is assumed to give a rough approximation of the global mean surface temperature. However, this index can also be artificially designed, for example as a simple linear transition from one state to the other (see Fig. A1a).

The basic idea has already been used in a variety of studies (e.g. Niu et al., 2019). The difference is only how the final interpolation method for $T$ and $P$ is implemented.

For both climate states a reference surface elavation $h_{0,1}^{\mathrm{ref}}$ must be provided. This is then used to account for elevation changes, similarly to the `elevation_change` atmosphere model of PISM, before combining both climate states. Temperature is assumed to change linearly with elevation change, $\Delta T = -\gamma_T \cdot \Delta h$, where $\gamma_T$ is the temperature lapse rate.

$$T^* = T + \Delta T = T - \gamma_T \cdot \Delta h \tag{D1}$$

Precipitation is scaled using an exponential factor $C$ for temperature:

$$P^* = P \cdot \exp\left(C \cdot \Delta T\left(\Delta h\right)\right) = P \cdot \exp\left(-C \cdot \gamma_T \cdot \Delta h\right). \tag{D2}$$

Using these equations, the temperature and precipitation fields are scaled to a common reference surface (i.e. sea level), before the actual interpolation is applied. As it applies to both fields, $T$ and $P$, we write $V$ instead:

$$V^{\mathrm{SL}}(t) = \left(V_1^{\mathrm{SL}} - V_0^{\mathrm{SL}}\right) \cdot i(t) + V_0^{\mathrm{SL}}. \tag{D3}$$

After this step, it needs to be checked that the result for precipitation is positive. In a final step, Eqs. (D1) and (D2) are applied again to transfer the results back onto the current surface elevation $h(t)$.

It should be noted that the `index` atmosphere model, as it was described above, also accepts seasonal (i.e. time varying, one year periodic) input fields. The climate states can thereby be described by monthly mean values as it is used by the Positive Degree Day (PDD) `surface` model to calculate the surface mass balance of the ice sheet. All command-line options used by this model are listed in Table D1.

## D2 Precipitation Cut-off

The PISM `precip_cutoff` atmosphere modifier sets precipitation to zero where one of the following conditions apply:

– the surface elevation exceeds a threshold height $h_{max}$ (and this method is not disabled),

– a user-defined mask has a value of 1.

This method is a simple approach to limit ice sheet growth in regions where other climate forcing methods fail to restrict precipitation. Table D2 shows all command-line options that are used by this PISM modifier.

**Table D2.** Command-line options for use with the PISM atmosphere modifier `precip_cutoff`.

| Command-line option | Comment |
| --- | --- |
| -use_precip_cutoff_height | Boolean variable to determine if the threshold height should be used. Default value: *false*. |
| -precip_cutoff_height | Defines the threshold height $h_{max}$ above which precipitation is set to zero. Default value: $3000\,\text{m}$ |
| -precip_cutoff_file | File containing the mask `precip_cutoff_mask`, which is 1 where precipitation should be eliminated, and 0 elsewhere. If not set, the mask is initialized with zeroes. |

*Author contributions.* The concept of this study was developed by all authors. SH developed the tool, conducted the experiments and did the analysis. EJG provided LGM palaeo-geography reconstructions used for model initialization. XZ provided LGM and PD climate reconstructions used for climate forcing of the model. The manuscript was written by SH with contributions from all co-authors.

*Competing interests.* The authors have no competing interests to declare.

*Acknowledgements.* SH received funding from the Federal Ministry of Education and Science (BMBF) through the project PalMod. EJG was funded by Helmholtz Exzellenznetzwerk "The Polar System and its Effects on the Ocean Floor (POSY)" and Helmholtz Climate

Initiative REKLIM (Regional Climate Change), a joint research project at the Helmholtz Association of German research centres (HGF). XZ was funded by the National Science Foundation of China (Nr. 42075047) and the Helmholtz Postdoc Program (PD-301). GL received funding from Helmholtz through the program PACES and Changing Earth — Sustaining our Future. Development of PISM is supported by NSF grants PLR-1603799 and PLR-1644277 and NASA grant NNX17AG65G. All computations were done at the AWI computing center. Furthermore, we want to thank Constantine Khroulev from University of Alaska Fairbanks for the helpful advises concerning the

implementation into PISM. Torsten Albrecht, Tijn Berends and an anonymous reviewer helped improve this manuscript with their thoughtful comments.

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
