# Peer review of "PISM-LakeCC: Implementing an adaptive proglacial lake boundary into an ice sheet model"

_The Cryosphere, 2020_

## Referee Comment (RC1) · Torsten Albrecht (Referee) · 28 Jan 2021

**Review comments for the manuscript No. tc-2020-353,**
**"PISM-LakeCC: Implementing an adaptive proglacial lake boundary into an ice sheet model"**
**submitted to The Cryosphere by Sebastian Hinck et al.,**
**reviewed by Torsten Albrecht (PIK)**

**General comments:**

Hinck and colleagues investigate the role of ponded proglacial lakes along the margin of the Laurentide Ice Sheet during glacial retreat. Assuming that, similar to marine-terminating glaciers (e.g. in Antarctica), the adaptive boundary conditions influence the (land-lake-terminating) ice sheet's stress balance in various ways. They can alter the ice flow (ice streams) and hence the overall ice sheet's geometry and stability. The changing ice load, in turn, results in the isostatic adjustment of the underlying lithosphere, which hence affects the formation (and demise) of lakes with up to several hundred meters depth in the vicinity of the ice sheet. By considering this feedback in a coupled model system applied to the North American ice complex the authors find in some lake regions self-amplified deglacial retreat, similar to what is commonly discussed for the Antarctic Ice Sheet in terms of the marine ice sheet instability (MISI). The significance of this "lake"-effect, comparing the four conducted experiments, is in deed surprising. Hence, the scientific insights of this study would be certainly a valuable contribution to the paleo ice sheet modelers community within the scope of "The Cryosphere".

The authors created a method based on a simple and efficient 4-neighbor "connected components (CC)" labelling algorithm that determines ocean and multiple lake basins for a given (or processed) bed topography and estimate the corresponding water levels by iterating over a set of increasing water levels, without including computational-expensive flow routing techniques. The standalone models "LakeCC" and "SL2dCC", adapted from Hinck et al., 2020, were implemented in the open-source Parallel Ice Sheet Model (PISM), which already comes with an solid-Earth deformation module. As the title of this study suggests, the ice model's marine boundary conditions have been generalized for this lake-coupling procedure. This does not mean that water density and calving rates have been simply adjusted, but that a whole PISM sub-module has been created with many functions and special considerations. For numerical stability reasons, the authors consider prescribed (ad hoc) lake filling rates that permit gradually evolving lake water levels for changing bed topography and ice margins.

The focus of this study is the description of the model implementation into PISM with a very detailed technical Appendix (which would also fit well to the Geoscientific Model Development) and to run simple deglacial simulations to test for its relevance as compared to the default case without the LakeCC method. From a modeler's point of view, this study could benefit from a few more sensitivity tests, that could help disentangling the individual contributions of the relevant processes acting at the lake-ice-bed boundary ("modification of the thermal regime at the submerged ice base, formation of ice shelves, increased ice loss due to melting and calving, and enhanced basal sliding near the grounding-line due to decreased effective pressure at the ice base... "). Hence, the reader would not only learn "that lakes matter" but also "why lakes matter". Also, the authors state, that the simulated ice sheet margins do not match well with reconstructions based on geological evidence, which is not the focus of the study. However, I encourage the authors to follow some of my suggestions in the specific comments below (mainly regarding the initialization of the LC bed deformation model and the

non-linear precipitation dependence on temperature index), which likely can improve the simulation outcome.

Overall, the study is well structured and the manuscript clearly-arranged. The draft consists of 36 pages including 9 main figures, 12 Appendix pages and 57 references.

**Specific comments:**

drainage events
*l. 22 "Reorganization of the lakes' drainage networks and sudden drainage events due to the opening of lower spillways may have impacted the global climate by perturbing the thermo-haline circulation system of the oceans (Broecker et al., 1989; Teller et al., 2002)." and l. 131: "For simplicity the fill-rates in our model are assumed to be constant." and l. 142: "If a basin disappears because it merged with the ocean, the lake level is gradually changed until sea level is reached, and then removed. "*
→ This is super exciting, but is it correct, that due to numerical stability requiring gradual lake filling, such events would be prohibited (or at least smoothed over long times) in this implementation?

lake merging
*l. 127: "However, there are special cases, such as adding a lake basin to an existing lake or adding a basin that has previously been connected to the ocean, that need more advanced treatment. For more details on this, see Appendix B."*
*l. 134: When water level is rising, this common level is chosen to be the lowest water level of that lake, $h_{min}$, while the highest level, $h_{max}$, is selected for the falling water level.*
*l. 139: "Only when h has exceeded the current (local) lake level, does its value get updated."*
→ What exactly is the current (local) lake level? A simple sketch could help here (maybe added to Fig. 1?). An explanation as in Hinck et al., 2020 may help: *"Until a patch merges with another one that is a sink, i.e. the lake overflows, the current level h is stored for all associated cells."*

2d sea level
*l. 157: " to determine the two-dimensional sea level field"*
→ I recommend to mention here early in the manuscript that, although the sea-level in PISM is treated as a 2D variable, in this application the value in each cell in every time step equals either a global constant or NaN.

*l. 522: "Here, we present the implementation of a sea level modifier, which takes advantage of possibility of a spatially variable sea level..."*
→ The manuscript could benefit from some motivation, why it is not sufficient to just update the already available 2D sea-level field with the various target lake levels (as general water level) and simply add a 2D field of corresponding water densities? I guess the numerical instabilities requiring a gradual lake filling (and hence more fields as the lake level) are one important argument for it, what else?

grounding line treatment

*l. 92: "The till below the water level next to the ice margin and grounding-line are assumed to be saturated. " and l. 172 "Cells with grounded ice below water level next to a lake are assumed to have saturated till. This reduces the effective pressure at the base of the ice sheet and reduces basal resistance in this location."*

→ I fully agree that this is a valid model choice. However, the underlying assumption of having saturated till within one grid cell length upstream of the grounding line (here 20km) has often been criticized within the community. Apparently, it highly increases the ice sheet's sensitivity (e.g., Golledge et al., 2015), in particular in combination with the basal melt interpolation (has this also been used)? For better comparison with other model studies it would be helpful to state the expected consequence of this model choice ("higher sensitivity"). Generally, this study could very much benefit from a few sensitivity runs in order to attribute the relative effects of some of the different processes named in Sect. 2.3 (and Fig. 2). Another (maybe more physical) way would be initializing all water-covered cells as saturated (https://github.com/pism/pism/pull/425), such that an advancing grounding line would not get temporarily stuck on initially unsaturated till (until it reaches saturation), which surprisingly seems to have a similar effect on paleo time scales, also for grounding line retreat (Albrecht et al., 2020a; Fig. A2).

marine vs. lake boundary conditions

*l. 165: "Generally, the parameterizations provided by PISM for a marine boundary are applied analogously at the lake interface. This, however, may not be the optimal treatment in every case. The model might benefit from future implementations of advanced or more specialized lake boundary treatment. Such limitations are discussed in Sect. 2.4." and l. 175: " the same parameterization for ice base temperature and sub-shelf mass-flux are used."*

→ As this marine assumption may overestimate sub-shelf melt rates in lacustrine environments, it would be helpful to state about what average melt rates we talk here, and what its relative contribution to the ice sheet mass balance is (aggregated rates) and on the enhanced deglacial retreat.

*l. 234: "At the shelf base (2), mass flux is parameterized using the model proposed by Beckmann and Goosse (2003)."*

→ In this paragraph the authors mention that the melt pump is expected to be weaker in fresh water due to lower buoyancy, while it could be also stronger as the temperature at the grounding line is likely higher than in marine environments. What estimate provides the Beckmann-Goosse model when the default values (35 g/Kg and -1.7°C?) were changed (0 g/Kg and +4.0°C) accordingly? Would the effective melting be higher or lower than for the marine default values and could this still be realistic (even though it was designed for marine environments)?

calving

*l. 247: "Implementation of a more physically based calving model capable of accurately parameterization of mass losses in both lacustrine and marine environments, will be needed (Benn et al., 2007)."*

→ The authors are also using the Eigencalving parameterization, which applies well for ice shelves in rather confined embayments, as fund around present-day Antarctica. Can you roughly estimate the relative contributions of thickness calving or Eigencalving in this study (maybe by comparing the two experiments with different thickness calving thresholds)?

l. 285: "...large ice shelves like those seen in the LCC experiment are unlikely to have existed."
→ Is there some reference? Does "large" mean covering entire lakes? Or would this imply that really thin ($\Delta h_L$ = 50m, l. 281) ice shelves are in fact unrealistic.

numerical instability and time stepping
l. 219: "Sudden jumps in water level can trigger numerical instabilities in the ice sheet model"
→ Can you say some more words on the possible reasons for numerical instabilities? Is this due to large areas of grounded ice becoming afloat at once affecting the non-local KSP iterative solution?

l. 83: "In this study we set an upper bound of 0.25 yr for the time step." and l.222: "Future implementations could possibly adapt a volumetric rate instead of fixing the rate of change of water level. By limiting the time step, sudden changes in water level could be performed quicker.
→ Please better motivate this particular time step. Does this choice help with keeping numerical stability with regard to the constant lake filling rate (l. 131)? Is this a best-practice value or is there some relationship between temporal and spatial resolution, as for instance the famous CFL criterion? Is the adaptive time step of PISM's sub-modules for the used resolution usually larger?

solid-Earth feedback
l. 93: "Bed deformation due to ice load is modeled in PISM using the Lingle-Clark model ..." and
l. 210: " If higher resolved input fields are available, e.g. from an external GIA model, the LakeCC model could be modified to do the calculations on that field instead and interpolate the output back onto the ice sheet model grid."
→ PISM has been recently coupled to a global solid-Earth model (VILMA, not published yet). Therefore, PISM (https://github.com/pism/pism/pull/463) can read the history of bed topography change on the ice model grid relative to a (high resolution) reference topography. The real benefit of this external GIA over the internal LC model, however, is that it self-consistently solves for the sea-level equation, i.e. it accounts for self-gravitational effects, which can be very relevant in (deglacial) grounding line migration in marine (or lake) environments. My guess is that lakes could be quite easily included, but this may require a closed water budget between ice sheet, lakes and ocean (l. 213). In any case, this would be rather an option for a follow-on study.

l. 259: "The temporal evolution of the topography provided by NAICE can be used to calculate the uplift rates that are used to initialize the Lingle-Clark bed deformation model of PISM."
→ The NAICE model makes use of a GIA model (which also solves the sea level equation) constrained with many different paleo data types, but it makes use of very simple assumptions on the steady ice state and boundary conditions. Please, provide some more information on how and when the NAICE uplift rates were used. It reads as you would take the LGM state and uplift rates from NAICE and run the Lingle-Clark model from there up to the present day? If this is true, I would expect for a different GIA (bed deformation) model with different mantle viscosities used, that this would imply a (almost

equilibrated) bed topography at present, which may differ from what we observe today, even if the ice thickness would be perfectly reproduced. Can you quantify the misfit in bed topography in your study?

l. 365 *"After deglaciation, the Hudson Bay region is over-relaxed and above PD sea level (see Fig. 8f)."* and l. 342: *" Our deglacial scenario fails at simulating realistic ice margin positions for the western LIS."*
A probably better way to avoid such a large misfit would be to initialize the Lingle-Clark model from present-day geometry and uplift rates and run it into the LGM state, from which you then start the experiments. Or if you want to make use of the constrained NAICE results at LGM you could make use of the simulated misfit at present an rerun each experiment with the initial bed topography adjusted according to the misfit, such that you end up with a better match at present in the second iteration.

l. 361: *"Maximum water depths close to the grounding-line are up to 1000 m. The main reason for this, we assume, is the GIA response modeled by PISM's bed deformation model."*
→What if the surface mass balance is simply overestimated (see my comments about index method), which "tends to accumulate too much ice" (l. 363), such that the Lingle-Clark model simply responds (in the correct way) to the higher load?

l. 362: *"This simple, two layered Earth model is not capable of handling the extreme deglaciation scenario of an entire continent."*
→ I think this statement is a bit harsh, as for the assumptions made, the Lingle-Clark model in fact can handle glacial cycles over an entire continent comparably well. I would agree that it is simple as it uses only one spatially constant mantle viscosity and does not account for self-gravitational effects, which may play a large role here. But my guess would be that the proper initialization of the Lingle-Clark model may bring much improvements here (see comment above).

l. 364: *"the rebound was not quick enough."*
The PISM default value for the upper mantle viscosity (likely used in this study) is $10^{21}$ Pas. NAICE, for instance, used a lower value of $4 \times 10^{20}$ Pas for the upper mantle, which implies a faster rebound. According to the relevance of GIA in this study, one or two sensitivity tests with varied mantle viscosity could bring some more interesting insights.

climate forcing
l. 269: *"To prevent ice sheet growths ... above 3500 m elevation, ice accumulation is prevented by setting precipitation to zero in these regions."* And also l. 338: *"Only by limiting precipitation above 3500 m elevation ... does further expansion of the ice sheets stop."*
What is the motivation for this constraint? Is this related to the findings (of maximum surface elevation) in the NAICE model? Or is it just gained experience with the model setup?

l.96: *"The surface mass balance (SMB) is estimated from monthly means of precipitation and surface air temperature fields... Precipitation and temperature fields are interpolated between two distinct climatic states, which are weighted according to a glacial index (see Appendix D1). "*
→ Does this mean that the interpolation is between each month of the two climate states?

*l. 295: "Starting from the LGM initial state after the spin-up, the ice sheets laterally expand and gain mass for about 6 kyr. Within this time, the LIS almost doubles its volume, before it retreats during the rest of the simulation." and also l. 332: "...dynamical equilibrium, as the initial ice sheet was reconstructed using geological evidence of ice margin history and GIA observations, and no ice dynamics were included (Gowan et al., 2016)."*
→ A model drift after initialization is typical. A dynamic equilibrium simulation (with constant dry LGM climate conditions) prior to actual forcing experiments could help here identifying relevant parameter settings that counteract the lateral expansion.

*l.335: "The reason for the ice accumulation is assumed to be due to the climate forcing. By linearly interpolating between the warm, humid PD and the cold, dry LGM climate states, unrealistically high accumulation is produced, especially in cold regions and on top of the ice sheet."*
→ This seems to be quite some considerable imbalance for applied linearly changing climate forcing. I agree that the interpolation is the most likely candidate here, as precipitation is rather non-linearly related to temperature change (Frieler et al., 2015). This means that precipitation after 6kyr would be much overestimated by the arithmetic mean (index 0.5) between LGM and PD state.

SLE unit
*l. 295: " sea level equivalent (SLE) ice volume"* and Fig. 4 caption: *"...and also includes ice shelves"*
→ I assume that the authors use SLE as a converted unit of grounded ice volume. If so, please specify the used conversion factor and also mention, that you do not make use of a 'volume above flotation' definition (for good reasons), to avoid misunderstandings.

ice lobes
*l. 300: "...ice margin between about 1 and 3.5 kyr is due to formation of small ice lobes that advance into small lakes (compare also with Fig. 6). "*
→ Is the ice margin at those ice lobes grounded or floating? Maybe provide some definition here? Is there a difference to an "ice shelf tongue"? There is a reference quite later in the text (l. 375).

ice streams
*l. 360: "The formation of ice streams .."*
→ Does a speed-up of ice flow imply that there wasn't an ice stream before, i.e. in terms of confined ice flow with speeds above (let's say) 100 m/yr?

channel filter
*l. 504: "Finally, to get rid of narrow lakes, which are often caused by the under-resolved topography, the target level is filtered by applying the FilterLakesCC method using a filter size of $N_{filter}$."*
→ Why would this be an issue? Does it mean, that a drainage event can only occur for a channel opening of width larger than $N_{filter}$?

**Technical corrections:**

l. 11 and l. 70 "...at the end of the simulation"
→ "... at present-day"

l. 28: "These processes can lead to the formation of ice streams..."
→ or: "...speed-up of ice streams..."

l. 30 "Due to various differences between the freshwater and ocean water, the interactions might be different than at a marine boundary (Benn et al., 2007)."
→ "Due to various differences between the freshwater and ocean water **regime**, the interactions **at the lake-ice boundary** might be different than at a marine boundary (Benn et al., 2007)."

l. 32 "In times when there was the production of large amounts of meltwater that caused the formation of proglacial lakes..."
→ "In **periods with** large amounts of meltwater that caused the formation of proglacial lakes..."

l. 34: " caused by rapid demise"
→ " caused by **a/the** rapid demise"

l. 36: "governed by the negative surface mass balance due to a warming climate"
→ As there are counteracting effects on the surface mass balance (also more precipitation for higher temperatures), it would be good to be more precise here, i.e. "due to **enhanced melting in** a warming climate"

l. 40: "Lake reconstructions of this time suggest water depths up to several hundred meters (Teller et al., 2002; Leverington et al., 2002)."
→ This has been already stated in l. 20, but that's ok.

l. 46:  global **mean** sea level

l. 56: "The latter reference discusses concerns when implementing such a lake-ice boundary condition for ice sheet models."
→ "The latter referenced article discusses **challenges for the implementation of** such a lake-ice boundary condition for ice sheet models."

l. 68:  "In places where there is ice-inward sloping topography, the lake can rapidly expand, as the water is deep enough that the ice begins to float."
→  "In places where the water is deep enough, the ice begins to float. In regions of ice-inward sloping topography, the grounding line can retreat in a self-amplified manner, which corresponds to a rapid expansion of the lake."

l. 109: "...quite dynamic" → "...particularly dynamic"

l. 110: "Therefore, when dynamically coupling ice sheets and proglacial lakes, steady updating of the geometry is necessary. "

→ "**For the dynamical** coupling **of** ice sheets and proglacial lakes **the continuous update** of **their geometries** is necessary. "

l. 112: "continental‑sized "

l. 113: " trade-offs have to be **made**"

Fig. 1 caption: "before it overflows into the ocean **or a neighboring lake basin**."

l. 183: "The pressure difference against the submerged ice margin is done analogously to the marine boundary"
→ but evaluated for the fresh water density

l. 207: "a after applying a secondary field."
→ As far as I understand this means "applying a correction" or "adjustment" of the low resolution data. Is this just an "anomaly field added"?

l. 259 "that are used  to initialize..."

l. 286: "... and Ctrl, without lakes. The PISM default run Def produces..."
→ Would it be possible to use some different font for the experiment names?

l. 293: "topographic setting" → maybe "configuration" would also fit here

l. 299: "Fig. 5 " → Figure 5

l. 300: "southward‑shifted"

*l. 311: "Where ice streams diverge, the surface velocity anomaly partially shows a slowdown of ice flow."*
→ Please be more precise, where is this and why?

l. 320: "... in **the** ice volume evolution of the LIS"

l. 325: " CIS" probably means Cordilleran Ice Sheet, please define.

l. 350: "**The r**econstruction"

l. 368: " large response" → "high sensitivity"

l. 369: " upstream **of** the lake boundary"

l. 371: " due to the changes in ice dynamics, the different experiments constantly diverge and features visible in these plots might therefore only indirectly be triggered by the lake boundary, but rather be a result of differences in ice sheet geometry."
→ Maybe split into two sentences. What is diverging?

l. 389: "ice-covered"

l. 392: " retreat**s**"

l. 406: " continental ice streams"
→ What does this mean? Is this related to the size?

Table A1 and A2: It would be nice for other modelers, if you could emphasize the options, which were added compared to the PISM base code (A1: index,precip_cutoff, lake_level (lakecc), (,sl2dcc), A2: -lakecc_dz, -lakecc_zmin , -precip_cutoff_height , -use_precip_cutoff_height).

l. 427: "Connected Components Algorithm"
→ capital or not (e.g., l., 457)? But in any case be consistent through the manuscript.

l. 442: "about **the** presence"

l. 442: I think, a common name in numerics for this approach at the boundary is "ghost point method"

l. 453: "Furthermore, two additional 2D diagnostic variables, which can be added to PISM's output, were added (see Table B1)"
→ Please be more precise. Do you mean the two variables, which are not "general" in Table B1? Maybe use the word "optional"?

l. 465-473: A simple schematic map including such cases (just some grid cells) could be helpful to understand the conditions in this paragraph.

l. 468: "ice-covered"

l. 522: "of **the** possibility "

l. 528: "model **to** identify"

l. 530: "Only patches that are connected to the margin of the computational domain are considered to be part of the ocean." and also l. 467: "but are not connected to either of the domain margins"
→ As the southern margin is land, I guess the condition would be "margin below sea level" or so?

l. 551: "usees "

**References, which are not already cited in the study:**

Albrecht, T., Winkelmann, R., & Levermann, A. (2020). Glacial-cycle simulations of the Antarctic Ice Sheet with the Parallel Ice Sheet Model (PISM)–Part 1: Boundary conditions and climatic forcing. *Cryosphere*, *14*(2).

Frieler, K., Clark, P.U., He, F., Buizert, C., Reese, R., Ligtenberg, S.R., Van Den Broeke, M.R., Winkelmann, R. and Levermann, A., 2015. Consistent evidence of increasing Antarctic accumulation with warming. *Nature Climate Change*, *5*(4), pp.348-352.

Golledge, N.R., Kowalewski, D.E., Naish, T.R., Levy, R.H., Fogwill, C.J. and Gasson, E.G., 2015. The multi-millennial Antarctic commitment to future sea-level rise. *Nature*, *526*(7573), pp.421-425.

---

## Referee Comment (RC2) · Tijn Berends (Referee) · 4 Feb 2021

The authors describe a set of simulations of the Laurentide Ice Sheet (LIS) during the last deglaciation, and the effect that proglacial lakes have on the evolution of the ice sheet. They used the well-established PISM ice-sheet model, including an additional module that dynamically tracks the extent and depth of proglacial lakes. By allowing ice shelves to form on these lakes, and by including some additional parameterisations for processes such as basal sliding, calving, and basal melt, they claim to have included all the ways a proglacial lake can affect the dynamics of the adjacent ice sheet. Their results show that the inclusion of these lakes in their model strongly accelerates the

retreat of the LIS, in a manner similar to the phenomenon of Marine Ice-Sheet Instability (MISI). While the authors do not mention this in their manuscript, this is an important conclusion, as the asymmetry in the Pleistocene glacial cycles (slow inception vs. fast deglaciation, and particularly the melt-water pulses) is something that's still not fully understood.

I think a study like this could be very interesting, and could contribute to our understanding of glacial dynamics. However, there are several issues with the methodology which I believe impact the validity of the conclusions. In particular, two important feedback processes (the lake-climate-SMB feedback and, more importantly, the geoid-MISI stabilisation) are not included, both of which would reduce the accelerated retreat the authors observe. I will detail these concerns below, after which I'll list the smaller technical questions I have.

1. The lake-climate-SMB feedback. At least two studies I know of, namely Krinner et al. (2004, Nature) and Peyaud et al. (2007, Climate of the Past), have looked at the effect proglacial lakes had on the local climate. Both find a net positive effect on the surface mass balance of the adjacent ice sheet, stabilising it against retreat. This would at least partially negate the acceleration due to grounding line dynamics described by the authors.

2. The geoid-MISI stabilisation. The authors refer to Weertman 1974 for proof of the Marine Ice-Sheet Instability (MISI). However, Weertman's proof that no stable equilibria exist for ice sheets whose margins lie on retrograde slopes did not account for GIA, nor for changes in the geoid. While the authors included a simple GIA module in their ice-sheet model, they did not account for changes in the geoid. Several different studies (the work of Natalya Gomez is probably the most important for the geoid, and that of Valentina Barletta for GIA) have shown that the fall in sea level at the ice margin, caused by the loss of ice mass in the interior, strongly reduces the retreat rate, and can even lead to stable equilibria on retrograde slopes. This has been proposed as an explanation for why the rapid West Antarctic retreat predicted by MISI is not really
visible in paleo evidence. Since the authors claim (in my view correctly) that the strongly accelerated retreat of the LIS in their model is due to the same instability, I believe it is crucial to take this feedback into account.

The fact that these two processes, particularly the geoid effect, are not included, leads me to believe that this study significantly overestimates the lake-induced acceleration of LIS retreat.

Aside from this, I also have a number of small, technical questions, which I will list here.

L29: "...ice streams, which impact the mass balance..." Don't you mean the ice dynamics? Mass balance is usually meant to include only surface and basal mass gain/loss.

L34: "...the 8.2ka event was caused by..." Too confident. While there certainly is strong evidence for this, I wouldn't say the matter is entirely settled.

L84: "The stress balance is modeled using a hybrid scheme based on the Shallow Ice (SIA) and Shallow Shelf Approximations 85 (SSA) of the full Stokes equations (Bueler and Brown, 2009)" It is well known that these hybrid models perform poorly at simulating grounding line migration. Many models now include a semi-analytical solution for the grounding-line flux as a boundary condition, but as far as I know this has not yet been implemented in PISM, and it is not discussed anywhere in the manuscript. While I can't say if this would lead to an over- or an underestimation of ice-sheet retreat in this particular study, I think it is important to discuss this, since grounding line dynamics are the root cause of all your results.

L88: "The basal resistance is determined using a model that assumes that the base of the ice sheet is underlain by deformable till." Since you explicitly state that the effect of lakes on basal sliding is important, this seems an oversimplification. The distribution of regolith in North America is far from uniform, and the interplay between (erosion and transport of) regolith, basal sliding, and glacial dynamics has been studied for over two

TCD
decades (e.g. Clark and Pollard, 1998). If you really want to present the effect of lakes on basal sliding as an important factor, then I believe a more elaborate approach is needed.

L102: "The marine boundary treatment is described in Winkelmann et al. (2011) and Martin et al. (2011). It includes a sub-shelf- melting parametrization..." If I'm not mistaken, this basal melt parameterisation was developed specifically for the Filcher-Ronne and Ross shelves in Antarctica. There, basal melt is mostly related to the intrusion of relatively warm deep water into the cavity between the ice shelves and the continental shelves, which leads to the depth-dependence in this parameterisation. I don't believe this translates well to the situation in Lake Agassiz. Since the authors show that calving plays in important role in their glacial dynamics, I suspect sub-shelf melt (which ultimately affects grounding line dynamics just as much as calving does) is equally important, and deserves a more accurate treatment than this. However, whether this oversimplification leads to an over- or underestimation of ice-sheet retreat, I cannot say.

L111: "Depending on the complexity of the lake model, computational overhead can drastically increase." I wonder if you considered using the flood-fill algorithm which I specifically developed for this kind of application (Berends and van de Wal, 2016). I've been using this for a while now, included in a 40km resolution ice-sheet model (solving for lakes at a 1 km resolution), and running full glacial cycle simulations is no problem at all (~60h computation time, including the SELEN sea-level model).

L117: "However, rapid changes in the boundary conditions resulting from this approach often cause numerical instabilities which cause the model to crash." What kind of numerical instabilities are these? I've never encountered this problem myself. It sounds like something that should be addressed in the numerical solver of the ice-sheet model itself, rather than by compromising on the lake-filling code. Regarding this compromise: exactly how fast do you move the lake level to the "target level", and how does this impact your results?

**TCD**
L155: "Therefore, the use of a more advanced sea level model is necessary. This sea level model implemented here." This is not a sea level model. Sea level is, in your setup, prescribed externally with the glacial index method. What you're describing here is a routine that determines the ocean mask.

L202: "For hydrological applications, such as lake basin reconstructions, the resolution an ice sheet model usually operates on is too coarse to resolve spillways through the terrain. Even more important than data resolution is the ice margin position and bed deformation due to GIA (Hinck et al., 2020). Considering the uncertainties of these fields retrieved from an ice sheet model, the resolution issue is regarded as a secondary issue." I'm not sure I agree here. Determining lake extent in a low-resolution DEM leads to a systematic overestimation of lake water volume (since you'll always underestimate the depth of drainage channels), which Berends and van de Wal (2016) showed to be around 10% for a resolution of 20km (this is why I developed my own algorithm!). This might not be much, and I don't it's something that should be fixed right away, but since it's an overestimation that's there throughout the simulations, it's something to keep in mind when you start looking at sea-level jumps and the likes.

L219: "Sudden jumps in water level can trigger numerical instabilities in the ice sheet model. To avoid such jumps, the water level 220 is gradually adjusted with a constant rate." Again, what do you mean by this? And why are you so sure that it is the sudden jumps in water level (which, if you look at sea-level records of the 8.2 kyr event, are definitely hinted at) that are unrealistic, rather than the behaviour of your numerical solver?

L234: "The model relates the mass flux to..." Could say anything about water temperatures in the lake? Krinner et al. (2004) find bottom water temperatures

melAlange..." It was my understanding that ice mélange buttressing is only relevant in fjords and such, where the convex coastlines can provide a backpressure to the mélange, which in turns pushes against the shelf front. Do you think this plays a significant role on the open water of Lake Agassiz?

Fig. 3: "The shaded area shows the regions attributed to the (continental) Laurentide Ice Sheet (LIS) in this study." I don't understand what you mean by this.

L269: "To prevent ice sheet growths in eastern Siberia and above 3500 m elevation, ice accumulation is prevented by setting precipitation to zero in these regions." This seems rather ad-hoc. Given that (at least in the ICE5G and ICE5G reconstructions) large parts of the LIS interior are above 3500m, how do you think this affects your results?

L272: "Transient sea level forcing is applied accordingly to the glacial index." This seems like an oversimplification, given that your main conclusion is rooted in grounding line dynamics. Even without using a geoid model, you could at least let eustatic sea level be calculated dynamically.

L274: "Before running the experiments, the model needs to be spun-up." How do you initialise englacial temperature? And why do you want the ice sheet to be in equilibrium with the prescribed climate before you start your simulations? Just as the real climate is never in a "steady state", so the real ice sheet would never have been in equilibrium with the climate, but always lagging behind it. It seems more logical to avoid these questions by starting your simulation in the Eemian interglacial.

L282: "...in order to reflect the fact that calving rates for freshwater terminating glaciers are reported to be an order of magnitude lower than rates observed for tidewater glaciers ..." I thought part of the reason why tidewater glaciers experience more calving is because of the tides after which they're named, which cause increased crevassing, as well as more wave action and other goings-on that weaken the ice. Do you think Lake Agassiz is more similar to the ocean, or to a small mountain lake, in that regard?

TCD
L285: "large ice shelves like those seen in the LCC experiment are unlikely to have existed" Why not? Do you cite any studies that support the existence of an open lake? Has IRD not been found in sediment cores, or has it never been looked for? Ice shelves are hard to track in proxy evidence, so I wouldn't be too quick to dismiss them.

L296: "Within this time, the LIS almost doubles its volume..." This sounds rather problematic. Why are your initial ice sheet and prescribed climate so far out of equilibrium? Aren't there any tuning parameters in your PDD scheme to correct for this? Also, looking at Fig. 4, your initial state has a volume of about 40 m SLE, which seems rather small for the Laurentide – I believe 60 - 80 m is a more commonly accepted number. (e.g. ICE5G, ICE6G).

Fig. 4: Where is the spin-up phase? When does the forced warming in the glacial index method start? Are lakes already included during the spin-up? And how does your "simulation time" correspond to real world time?

L314: "At around 9 kyr the water level of this lake rapidly dropped, as a lower outlet to the Atlantic became ice-free." Which outlet? I'd like to see a map showing the locations of the possible spillover (Mississipi, St. Lawrence River, MacKenzie River) and drainage (Hudson Strait, Lancaster Sound, North-West Passage) routes in relation to your ice-sheet geometry.

L321: "At around 17.9 kyr, the ice saddle over Hudson Strait breaks apart and allows the lake to drain into the Labrador Sea" The deepest part of Hudson Strait is quite narrow, so a 20km DEM might significantly underestimate the water depth, and therefore the retreat rate. How do you think this affects your results?

L322: "Due to GIA processes, the ice-free Hudson bay basin eventually rises above sea level" Is this realistic? I've never seen this happen in my own model runs (which include an actual geoid model), for me sea-level rise always outpaces isostatic rebound, but I don't know what the field data indicates.

TCD
L326: "At around 21 kyr, the saddle collapses, which drains most of the lake." Again, how does 21 kyr simulation time correspond to real-world time? Does this match the 8.2 kyr event?

Fig. 5: why does the ice margin in the LCC simulation make a  ${\sim}500$  km "jump" at  ${\sim}18$  kyr?

L336: "By linearly interpolating between the warm, humid PD and cold, dry LGM climate states, unrealistically high accumulation is produced" There is no climate feedback in your model, your entire climate is prescribed through the glacial index. With a glacial index of 1 at t=0, the prescribed climate should be exactly that of the GCM that produced it. What you mean is that your initial ice sheet is simply not in equilibrium with this steady-state climate. This is why paleo-ice-sheet models generally need some form of tuning in their SMB parameterisations, and also why it's usually better to start a simulation in an interglacial (e.g. the Eemian) and run forwards from there (since you wouldn't expect the LGM ice sheet to be in equilibrium with the LGM climate in any case).

Fig. 9: What does the thick blue line in panel d) signify?

L354: "For this reason, the lake reconstructions are not expected to match well with observations" This is a bit of a chicken-and-the-egg question; are your lakes wrong because your ice margins are wrong, or are your ice margins wrong because your lakes are wrong? Since the entire point of your paper is to show that the presence of the lakes affects the ice sheet (and the therefore the ice margin), you cannot simply ignore the feedback here.

L355: "Drainage towards the Arctic, for example, is blocked until the ice saddle connecting the LIS and the CIS collapses" This raises the question of, where would the real paleo-lakes have routed their spillover, and is that pathway indeed blocked by your modelled ice-sheet? If not, then this might be a resolution issue as I mentioned earlier.
L362: "This simple, two layered Earth model is not capable of handling the extreme deglaciation scenario of an entire continent" This is unsatisfactory. In the introduction section, you (correctly) state that the GIA depressions are the reason those lakes exist in the first place. If your GIA model isn't performing well, then this should be fixed.

L373: "We assume that this is due to the enhanced sliding at the lake boundary" This assumption can, and should, be easily verified, by turning off the "till saturation at next-to-lake ice pixels" parameterisation you described earlier.

L375: "When water depth of the proglacial lake becomes too deep, the thin advancing ice front is presumably lost due to calving." Presumably? Again, this seems like something that could (and should) be easily checked.

L390: "Lakes, however, do impact the early retreat by inducing the formation of ice streams, which drain the ice sheet interior" This conclusion is not supported by your results. You should check what happens when this lake-enhanced sliding is turned off.

L395: "Reconstruction of lakes could benefit from more realistic accounting of water fluxes" The assumption that the lakes are always filled to overflowing is probably one of the most justifiable ones you made, so I doubt that including a water transport model would significantly alter your results.

L397: "a more physically motivated calving model valid for grounded and floating ice termini or a lacustrine sub-shelf melting model, would improve the ice-dynamical response to lakes" Since you already show that the choice of calving law has a strong impact on your results, this one seems a lot more important.

L406: "The lake model promotes the formation of continental ice streams" Your figures only show ice lobes, no ice streams.

TCD

---

## Short Comment (SC1) · 5 Feb 2021

The Lingle-Clark GIA module that we use calculates the effects of the ice load on a self-gravitating, spherical Earth that responds to changes in the ice load. It is calculating the visco-elastic and elastic deformation and gravitational changes, i.e. changes in the geoid. We fully acknowledge that the Lingle-Clark model is not as sophisticated as a full sea level equation solver like SELEN, particularly because it does not include a higher viscosity lower mantle that is responsible for a substantial amount of the GIA response in North America. However, for the purposes of our experiments where we want to demonstrate that dynamically evolving lakes affect ice sheet retreat in a substantial way, the Lingle-Clark model includes the two components most important for determining lake geometry - Earth deformation and gravitational changes. The lake module responds to dynamic changes in the topography, and therefore evolves when there are changes in the geoid in the experiment.

---

## Referee Comment (RC3) · Tijn Berends (Referee) · 9 Feb 2021

I think there's some confusion about the different uses of the phrase "self-gravitating". If I have read the description of the Lingle-Clark GIA model given by Bueler et al. (2007, which is what you refer to in your manuscript) correctly, their use of the phrase "self-gravitating" mean they include a self-gravitation term in the derivation of the Green's functions, which describes the diminishing local gravity as asthenosphere mass is displaced by a surface load.

What I meant in my review is the perturbation of the geoid by the mass of the surface load itself: the changing shape of the ocean surface caused by the gravitational

attraction of an evolving ice sheet. As an ice sheet retreats, relative sea level at the margin drops due to [1] instantaneous elastic rebound of the crust, [2] delayed viscous rebound of the mantle, and [3] instantaneous lowering of the geoid, due to the diminishing gravitational attraction of the ice sheet, which causes the ocean water to "relax" back to the opposite side of the Earth. The version of the Lingle-Clark model described by Bueler et al., 2007 includes [1] and [2] (with a self-gravitation term included in the calculation), but not [3].

The work by Natalya Gomez which I referred to earlier (Gomez, N., Mitrovica, J. X., Huybers, P., and Clark, P. U.: Sea level as a stabilizing factor for marine-ice-sheet grounding lines, Nature Geoscience 3, 850-853, 2010, doi: 10.1038/ngeo1012) shows that the magnitude of [3] is similar to that of [2], but since it is instantaneous rather than delayed, the effect on ice dynamics is much stronger, such that it can significantly reduce retreat rates, or even lead to stable grounding lines on (mildly) retrograde slopes. This is what I meant with the "geoid-MISI feedback" (though indeed Gomez et al. don't use that specific phrase). I think this is very relevant for the phenomena you're investigating here, and I'm not convinced that the strongly accelerated retreat in your results would still occur if this effect would be included.

Lastly, a minor point: the phrase "dynamical topography" is typically used to describe tectonic movement, changes in elevation due to mantle convection, and other processes that act on the Myr timescale.

---

## Referee Comment (RC4) · Anonymous Referee #3 · 9 Feb 2021

This study is a great concept and shows how ice-marginal lakes can potentially change the modelled retreat of large continental ice sheets. This is mostly relevant to the glacial NH paleo-ice sheets (less so Greenland or Antarctica, or small valley glaciers), but nevertheless the study will still be interest to the many people working on N. Atlantic deglacial climates, because of the great significance of meltwater inputs to the ocean from the Laurentide ice sheet. The authors also demonstrate a process driven by ice-marginal lakes that could lead to rapid retreat of what is traditionally considered a land-terminating ice sheet margin.

There are many uncertainties, perhaps obviously - and it's good that the authors focus

mainly on the different behaviours with/without the lakes, and skip a detailed comparison with geological evidence at this stage in development.

I have listed some specific comments/questions below but my two main concerns for implementation of this new LakeCC scheme are as follows.

1) Due to numerical instabilities, the model apparently cannot cope with rapid lake drainage/filling. This is of concern because the rapid drainage of lakes into the North Atlantic is one of the main applications of this work, yet lake level changes in a 'work-around' solution appear to be limited to 1 m/yr (the gamma parameter in Eq. 1 and Table B2). This represents a maximum of only 25 cm per 3-month model time step. The cause of the instability is not discussed but if it cannot be solved without the current 'work-around' then I struggle to see how this new scheme can be implemented in a realistic scenario.

2) Use of the marine ice shelf parameterisations is really questionable in a lacustrine setting. Fast sub-shelf melting in marine settings is enabled partly by the density contrast between dense saline ocean water and light fresh melt water. This contrast drives rapid overturning in the shelf cavity, and large heat fluxes. In a lacustrine setting, the density contrast between fresh lake water and fresh melt water is much lower, so presumably the sub-shelf melt rate will also be much lower, for a given temperature forcing. In which case, perhaps a lot of the simulated shelves would really just remain as grounded ice. The authors do of course acknowledge this shortcoming, but don't then address it. Following on from this point, what is the water temperature in the lake? I can't see how this is estimated but it is crucial for calculating subshelf melting.

Since there is so little work on this subject it would be great to see this study published. In this respect, I believe that the instability issue either needs fixing or needs much more discussion, so we are convinced it isn't a symptom of some other underlying problem in the model, and so that we know how the problem of unrealistically slow lake draining/filling can be overcome. Second, because this is largely a model development
paper, we need some basic indication of importance of a few critical parameters (in this application, perhaps the till friction angle, ice shelf basal melt, grid resolution). Even some short model runs could help answer that? Finally, although not essential to the concept of the study, if you carry out some more simulations I would also revisit the climate forcing and need for the 3500m elevation limit.

Specific comments/questions:

Abstract (L7), Conclusions, & elsewhere: You don't specifically show ice streams along the continental margin, even in Fig 7. There are regions of faster flow in the LCC run but these don't look particularly stream-like. Because you are using a spatially uniform basal parameters, and a coarse grid, this aspect of ice dynamics isn't well captured by your model. Rather than saying you develop ice streams, I would recommend to simply point out that ice velocity is higher at locations X in the LCC run.

L22 "Reorganization of the lakes' drainage networks and sudden drainage events due to the opening of lower spillways may have impacted the global climate by perturbing the thermohaline circulation system of the oceans (Broecker et al., 1989; Teller et al., 2002)."

It's important here to note that drainage could have been under the ice, as well as over it. Lack of subglacial routing option means lakes can only overspill, yet subglacial drainage could lead to outburst events long before the supraglacial spillway is reached. Omitting this subglacial option perhaps helps the experimental design but it deserves at least a mention.

L34: Not an important point but the NH cooling was maybe caused by freshwater fluxes (not definitely, as implied here).

L41 "Apart from their relevance for understanding processes that lead to the demise of the late Pleistocene and early Holocene ice sheets, interest in contemporary proglacial lakes and their role in glacial retreat is growing (Carrivick and Tweed, 2013). Motiva-

TCD
tions for these studies range from predicting and managing water resources under a warming climate and recognizing possible risks due to glacial outburst floods (Carrivick et al., 2020)."

I'm very sceptical of this work being usefully down-scaled to the small present-day proglacial lakes, because the parameterisation of ice-lake interactions would be so specific to a given site. Maybe best to stick to the paleo aspect.

L79: Resolution (20 km), is this really sufficient to capture the processes you are aiming at modelling? I understand the limit on computational effort but wonder if the positive feedbacks between lake development and ice stream development could be underestimated. For example a proglacial lake extending for 50 km along the ice sheet margin would only meet the simulated ice sheet at two or three grid points, is that enough to initiate an ice stream in the model? I am not suggesting lots of runs with different configurations, but it would be good to see some sort of sensitivity. For example, run the model at 20km to a couple of interesting points and use these as initial states for some short simulations with higher (and lower) resolution.

L88: Basal boundary condition. The ice sheet sits on till, and slides when driving stress is greater than the yield stress of till. Is the substrate taken as being spatially uniform? Is there some sensitivity to this? Actually the design of this experiment is such that specifying a uniform yield stress is maybe of benefit (it is purely the addition of the LakeCC that is being studied), but perhaps the influence of LakeCC is very much dependent on the basal slipperiness, as this will dictate how far inland the "ice streams" can propagate. In which case some simple sensitivity expt as above would be very useful.

L99 "To prevent the ice sheet from expanding into regions and high elevations where a more advanced approach would limit precipitation, the second model sets precipitation to zero above a threshold height or accordingly to a given mask (see Appendix D2)"

This seems like a dubious approach to me. If the ice sheet can grow sufficiently thick
that precip cannot be parameterised, is there not something wrong with the ice sheet model configuration or climate forcing? What evidence do we have the the ice sheet elevation was less than 3500 m? This is a very low upper limit for an ice sheet of that size. On L363 you mention "the climate forcing tends to accumulate too much ice...". There are so many unknowns in the ice sheet model, could you not also say "the ice sheet tends to dissipate too little ice"? For example, why not just use a slightly more slippery bed (since this is so poorly constrained anyway) to avoid this problem of too much ice?

Sect 2.2 LakeCC

L116: Rapid filling causes the model to crash. As a modeller this is very worrying given that the maximum time step for the ice sheet model is only 0.25 yr (3 months). What is the reason for the numerical instability – e.g., is it in the ice sheet model or LakeCC? The implemented lake filling algorithm effectively dampens lake level changes, what is the time scale for this? Is the max filling/draining rate (gamma) really just 1 m/yr?

This limit on drainage rate could place quite some restriction on the usefulness of the model in a real deglacial lake drainage setting. For example, if a lake drainage is initiated by overspilling the ice sheet (or by growing a subglacial channel), then in practice the very strong positive feedback could lead to rapid drainage by incision of a supraglacial channel or growth of a subglacial conduit. But how can the model cope with this? If the lake level is reduced artificially slowly, does that mean a lot more water drains out of the lake than was actually in it? And would a drainage channel/conduit become vastly over enlarged because the lake level (and thus the hydrostatic head driving the channel/conduit development) is held artificially high?

L142 "If a basin disappears because it merged with the ocean, the lake level is gradually changed until sea level is reached, and then removed."

I don't understand here how a lake can merge with the ocean unless its lake level already matches the sea level.

TCD
L165: The general PISM marine boundary parameterisations are used. Were these not developed for saline water? I think modifying these for freshwater would be a fairly fundamental step in developing this new LakeCC component. In particular with ref to Line 237 ("Due to the difference in density between fresh and ocean water, the layer of melt water underneath the ice shelf experiences a  $\hat{a}$ Lij 200 times higher buoyancy in sea water...") this does seem like it should be addressed here rather than a future study. What if there is a 200 times less sub-shelf melt beneath a lacustrine ice shelf? Will the development of floating ice shelves then be much less likely? What is the ambient water temperature in the lake – in a marine setting this would be crucial information.

L176 "Ice temperature at the base is set to the pressure-melting point, which is a function of pressure, hence ice thickness." Presumably the PMP is also a function of salinity in your model?

L266 "When using a glacial index derived from the NGRIP  $\delta$ 18O measurements (North Greenland Ice Core Project Members, 2007), the modeled deglaciation is too rapid to identify contributions from the lake model. Instead, the deglacial climate signal was crudely approximated by a linear model (see Fig. A1a)." I agree that a linear transition is a better way of understanding how the new LakeCC model works, than using the noisy NGRIP d18O. Indeed the NGRIP record may well contain signals of the very lake drainages your model is trying to capture. But I worry from this statement (& comments above) that the model can't cope with rapid changes in climate forcing, and this would be a significant limitation given its intended application to deglacial climates.

L284 "In the LNS experiment  $\Delta hL$  was greatly increased to remove almost any floating ice, as large ice shelves like those seen in the LCC experiment are unlikely to have existed" I get the point of this sensitivity run but is there actually paleo evidence to convincingly disprove the existence of extensive ice floating ice shelves?

L315, L358 Growth of the very large proglacial lakes (current Great Lakes region &
then Agassiz). What are the lake levels and would they have been likely to drain subglacially, given their proximity to the ice margin and also considering that sea level was lower at that time? This subglacial drainage route would be one way of avoiding unreasonably large lakes – although I understand that adding that process isn't the point of the present study. Nevertheless, is surely worth a mention.

L345 "Without adding more advanced climatic feedbacks a realistic deglacial reconstruction is not expected. This, however, has not been the focus of this study, which is to test the PISM-LakeCC model and studying its impact on the ice dynamics and the glacial retreat. For these purposes the experimental setup is sufficient. Analyzing the interplay between ice sheets and proglacial lakes in more realistic setups, e.g. fully coupled to a climate model, and comparing against various geological proxy data is an interesting topic for future research"

To me this is one of the real positives of this paper – it focuses just on how the addition of lakes modifies the modelled retreat. The simple linear forcing greatly helps interpretation of the results, and similarly there is no long discussion of why the model inevitably doesn't fit geological reconstructions. This is also why I think the paper would really benefit from a little more testing of sensitivity. At the moment the paper is a model development study and in this case the sensitivity aspect is very important.

Fig 3: Last sentence of the caption – what is the continental LIS & should it really extend that far into the Atlantic?

Fig A1: would be great to have this in the main text.

Spelling/grammar: Needs a proof read to iron out several minor errors.

TCD

---

## Author Comment (AC1) · 11 May 2021

**Author response to referee comment RC1 by Torsten Albrecht (PIK) to manuscript No. tc-2020-353, "PISM-LakeCC: Implementing an adaptive proglacial lake boundary into an ice sheet model"**

**submitted to The Cryosphere by Sebastian Hinck et al.**

**General comments:**

Hinck and colleagues investigate the role of ponded proglacial lakes along the margin of the Laurentide Ice Sheet during glacial retreat. Assuming that, similar to marine-terminating glaciers (e.g. in Antarctica), the adaptive boundary conditions influence the (land-lake-terminating) ice sheet's stress balance in various ways. They can alter the ice flow (ice streams) and hence the overall ice sheet's geometry and stability. The changing ice load, in turn, results in the isostatic adjustment of the underlying lithosphere, which hence affects the formation (and demise) of lakes with up to several hundred meters depth in the vicinity of the ice sheet. By considering this feedback in a coupled model system applied to the North American ice complex the authors find in some lake regions self-amplified deglacial retreat, similar to what is commonly discussed for the Antarctic Ice Sheet in terms of the marine ice sheet instability (MISI). The significance of this "lake"-effect, comparing the four conducted experiments, is in deed surprising. Hence, the scientific insights of this study would be certainly a valuable contribution to the paleo ice sheet modelers community within the scope of "The Cryosphere".

The authors created a method based on a simple and efficient 4-neighbor "connected components (CC)" labelling algorithm that determines ocean and multiple lake basins for a given (or processed) bed topography and estimate the corresponding water levels by iterating over a set of increasing water levels, without including computational-expensive flow routing techniques. The standalone models "LakeCC" and "SL2dCC", adapted from Hinck et al., 2020, were implemented in the open-source Parallel Ice Sheet Model (PISM), which already comes with an solid-Earth deformation module. As the title of this study suggests, the ice model's marine boundary conditions have been generalized for this lake-coupling procedure. This does not mean that water density and calving rates have been simply adjusted, but that a whole PISM sub-module has been created with many functions and special considerations. For numerical stability reasons, the authors consider prescribed (ad hoc) lake filling rates that permit gradually evolving lake water levels for changing bed topography and ice margins.

The focus of this study is the description of the model implementation into PISM with a very detailed technical Appendix (which would also fit well to the Geoscientific Model Development) and to run simple deglacial simulations to test for its relevance as compared to the default case without the LakeCC method. From a modeler's point of view, this study could benefit from a few more sensitivity tests, that could help disentangling the individual contributions of the relevant processes acting at the lake-ice-bed boundary ("modification of the thermal regime at the submerged ice base, formation of ice shelves, increased ice loss due to melting and calving, and enhanced basal sliding near the grounding-line due to decreased effective pressure at the ice base... "). Hence, the reader would not only learn "that lakes matter" but also "why lakes matter". Also, the authors state, that the simulated ice sheet margins do not match well with reconstructions based on geological evidence, which is not the focus of the study. However, I encourage the authors to follow some of my suggestions in the specific comments below (mainly regarding the initialization of the LC bed deformation model and the non-linear precipitation dependence on temperature index), which likely can improve the simulation outcome.

Overall, the study is well structured and the manuscript clearly-arranged. The draft consists of 36 pages including 9 main figures, 12 Appendix pages and 57 references.

We would like to thank Torsten Albrecht for reviewing our paper. The original comments are indented, while our responses aligned to the left of the page.

For this review round several sensitivity experiments were run, which are referred to in our responses. Details about these runs and snapshots are combined in a supplementary document. This document is available online (https://doi.org/10.5281/zenodo.4746501).

**Specific comments:**

drainage events

*I. 22: "Reorganization of the lakes' drainage networks and sudden drainage events due to the opening of lower spillways may have impacted the global climate by perturbing the thermo-haline circulation system of the oceans (Broecker et al., 1989; Teller et al., 2002)."*

and I. 131: "For simplicity the fill-rates in our model are assumed to be constant."

and I. 142: "If a basin disappears because it merged with the ocean, the lake level is gradually changed until sea level is reached, and then removed."

 $\rightarrow$  This is super exciting, but is it correct, that due to numerical stability requiring gradual lake filling, such events would be prohibited (or at least smoothed over long times) in this implementation?

The implementation of proglacial lakes as described in this work only affects the direct ice - lake interaction within PISM. Modeling the impact of redistributions of freshwater onto the climate system would require modeling and coupling Earth's different subsystems. For coupling the PISM-LakeCC model to an ocean model, translation of the water levels into freshwater fluxes would be necessary. This task, however, is not trivial, as the LakeCC model does not conserve water volume. Furthermore, water fluxes would have to be calculated by comparing the water distribution between two time slices. Any changes in water level or drainage route that happened in between those steps would not be resolved.

The sentence you are referring to from the introduction (I. 22) was added to highlight the further potential impact of proglacial lakes on the climate system. However, in many ways this model is a preliminary first step in implementing dynamically evolving lakes, and the first priority was to provide a stable model that can interact directly with the ice sheet.

One way to include data from the PISM-LakeCC model in a coupled Earth System model, could be by using the information of lake basins. Water fluxes from different sources (e.g. precipitation, ice sheet run-off) could then be accumulated within these basins and extracted from the water cycle. By redistributing the collected water volumes the changing topographic setting could be attributed for. Excess water can then again be added as a freshwater flux into the climate system. However, the complexity of accounting for all of the water sources is beyond the scope of this paper.

**lake merging**

I. 127: "However, there are special cases, such as adding a lake basin to an existing lake or adding a basin that has previously been connected to the ocean, that need more advanced treatment. For more details on this, see Appendix B." I. 134: When water level is rising, this common level is chosen to be the lowest water level of that lake, hmin, while the highest level, hmax, is selected for the falling water level.

I. 139: "Only when h has exceeded the current (local) lake level, does its value get updated."

 $\rightarrow$  What exactly is the current (local) lake level? A simple sketch could help here (maybe added to Fig. 1?). An explanation as in Hinck et al., 2020 may help:

"Until a patch merges with another one that is a sink, i.e. the lake overflows, the current level h is stored for all associated cells."

We have added the lake level in the legend of Fig. 1. Probably "current (local)" lake level was confusing, as we just meant the lake level. Therefore we have shortened this description in the new manuscript.

The quote from the Hinck et al. (2020) paper that you mention describes a different context. There we describe the LakeCC algorithm, which is used here to determine the target\_level, while here we describe the gradual filling algorithm.

2d sea level

I. 157: "to determine the two-dimensional sea level field"

 $\rightarrow$  I recommend to mention here early in the manuscript that, although the sea-level in PISM is treated as a 2D variable, in this application the value in each cell in every time step equals either a global constant or NaN.

Yes, this is a good point. We have added the following to the description of the ice sheet model:

"Marine regions of the ice sheet are defined in PISM via the flotation criterion, which describes whether the ice at given sea level and thickness is grounded or floating. The sea level is defined via a 2D map, which allows it to be spatially variable. In general, however, it is set to a global mean value prescribed by a scalar time series." *I. 522: "Here, we present the implementation of a sea level modifier, which takes advantage of possibility of a spatially variable sea level..."*

 $\rightarrow$  The manuscript could benefit from some motivation, why it is not sufficient to just update the already available 2D sea-level field with the various target lake levels (as general water level) and simply add a 2D field of corresponding water densities? I guess the numerical instabilities requiring a gradual lake filling (and hence more fields as the lake level) are one important argument for it, what else?

We have added a short paragraph about it to the Appendix:

"A lake, as seen by an ice sheet model, is very similar to a locally elevated sea level position. It would be possible to implement the lake interface as a sea level modifier in PISM. However, physical differences (e.g. density, temperature distributions) require different treatment of marine and lacustrine environments. To distinguish between both cases, at least two spatial maps are required. This can either be done by providing the lake and sea level combined in one field and additionally providing a mask, or providing lake and sea level elevations as separate maps. For the implementation of the PISM-LakeCC model, we chose the latter case."

For us it seemed more clean to have these things on different fields. Gradual filling should not have been a problem, as these secondary fields are used only internally of the lake model/modifier anyway. It was just a design question and we chose the other path.

**grounding line treatment**

*I.* 92: "The till below the water level next to the ice margin and grounding-line are assumed to be saturated. " and *I.* 172 "Cells with grounded ice below water level next to a lake are assumed to have saturated till. This reduces the effective pressure at the base of the ice sheet and reduces basal resistance in this location."

 $\rightarrow$  I fully agree that this is a valid model choice. However, the underlying assumption of having saturated till within one grid cell length upstream of the grounding line (here 20km) has often been criticized within the community. Apparently, it highly increases the ice sheet's sensitivity (e.g., Golledge et al., 2015), in particular in combination with the basal melt interpolation (has this also been used)? For better comparison with other model studies it would be helpful to state the expected consequence of this model choice ("higher sensitivity"). Generally, this study could very much benefit from a few sensitivity runs in order to attribute the relative effects of some of the different processes named in Sect. 2.3 (and Fig. 2). Another (maybe more physical) way would be initializing all water-covered cells as saturated (https://github.com/pism/pism/pull/425), such that an advancing grounding line would not get temporarily stuck on initially unsaturated till (until it reaches saturation), which surprisingly seems to have a similar effect on paleo time scales, also for grounding line retreat (Albrecht et al., 2020a; Fig. A2).

Yes, this is an important point. We have conducted two more experiments targeting the sensitivity of the grounding line treatment (see experiments *TWO* and *nSG* in the appended document). These experiments confirm this high sensitivity.

In *TWO* the "till water ocean" model (PR425) was used instead of the slippery grounding line treatment. The ice sheet evolves in this scenario almost identical to the *lcc* run (which uses the slippery grounding line treatment).

In *nSG* the slippery grounding line treatment was deactivated. E.g. the till water of cells at the grounding line was not modified. This strongly impacts the glacial retreat, as the grounding line flux, and marine and lacustrine discharge are strongly reduced.

In all experiments the sub-grid grounding line treatment as activated by the -pik option are used. This also included includes the interpolation of basal melt.

We will address the issue of grounding line treatment (sensitivity of basal strength, but also sub-grid treatment) in the methods section.

**marine vs. lake boundary conditions**

*I.* 165: "Generally, the parameterizations provided by PISM for a marine boundary are applied analogously at the lake interface. This, however, may not be the optimal treatment in every case. The model might benefit from future implementations of advanced or more specialized lake boundary treatment. Such limitations are discussed in Sect. 2.4." and *I.* 175: "the same parameterization for ice base temperature and sub-shelf mass-flux are used."

 $\rightarrow$  As this marine assumption may overestimate sub-shelf melt rates in lacustrine environments, it would be helpful to state about what average melt rates we talk here, and what its relative contribution to the ice sheet mass balance is (aggregated rates) and on the enhanced deglacial retreat.

We have investigated the different contributions of mass loss for our experiments (see the figure for the LCC standard lake experiment).

Different contributions to mass loss for the lcc experiment. The top blue line shows the surface accumulation, while the lower black line shows the overall rate of change of the ice mass (The net surface mass balance is SMB = Accumulation - Runoff). The plot shows that the main process governing mass loss is surface runoff (= melt - refreeze) followed by glacial (marine) discharge (i.e. calving). Sub-shelf melting in lakes contributes only up to 6% to the total (non-surface) mass losses when large lacustrine ice shelves are present (15 kyr).

In the LCC experiment typical (depth dependent) sub-shelf melt rates in lakes are between 100 and 200 kg m-2 year-1.

For the impact on the deglacial scenario please see the next point!

**I. 234: "At the shelf base (2), mass flux is parameterized using the model proposed by Beckmann and Goosse (2003)."**

 $\rightarrow$  In this paragraph the authors mention that the melt pump is expected to be weaker in fresh water due to lower buoyancy, while it could be also stronger as the temperature at the grounding line is likely higher than in marine environments. What estimate provides the Beckmann-Goosse model when the default values (35 g/Kg and -1.7°C?) were changed (0 g/Kg and +4.0°C) accordingly? Would the effective melting be higher or lower than for the marine default values and could this still be realistic (even though it was designed for marine environments)?

To address this point we conducted an experiment (*MR*) where we tried to estimate the relative difference between melt rates in marine and lacustrine settings using the melt pump parameterization. Details can be found in the appended document describing the additional experiments. Melt rates are strongly dependent on the assumed mean temperature of the lake. Furthermore, melting depends on the pressure and thus depth dependent freezing point. For fixed lake temperature and depth we estimated the effectiveness of melting in marine relative to lacustrine environments. Using  $T=2^{\circ}C$  and d=300m marine melting is estimated to be about 40 times stronger. This factor is used to scale the melt rate tuning parameter accordingly.

The impact of sub-shelf melting on the ice sheet evolution becomes even smaller when applying this simple lake correction. Calving is still the dominant lacustrine term so that the results are very similar to the *lcc* scenario. The largest contribution on ice mass loss is from surface runoff, which is further increased due to the ice surface lowering upstream the lake boundary. This effect is also described in a recently published study by Quiquet et al. (2021). We will discuss this issue in the revised manuscript.

**calving**

*I.* 247: "Implementation of a more physically based calving model capable of accurately parameterization of mass losses in both lacustrine and marine environments, will be needed (Benn et al., 2007)."

 $\rightarrow$  The authors are also using the Eigencalving parameterization, which applies well for ice shelves in rather confined embayments, as fund around present-day Antarctica. Can you roughly estimate the relative contributions of thickness calving or Eigencalving in this study (maybe by comparing the two experiments with different thickness calving thresholds)?

For the newly conducted experiments we checked the contribution of the Eigencalving to glacial discharge and realized that for our experiments it is always zero. Ice discharge at the ice margin is thus solely due to the thickness calving parameterization. To better compare the impact of the thickness calving threshold on the glacial retreat another experiment (*redcalv*) with reduced threshold ( $\Delta$ h=20m) was conducted.

The ice loss rises roughly linearly with the thickness calving threshold  $\Delta h$ :

---

## Author Comment (AC4) · 11 May 2021

The reply to RC3 is included in the response to RC2.

―――――――――――――――――――

---

## Author Response (AR1)

**Major revision of manuscript tc-2020-353 -- PISM-LakeCC: Implementing an adaptive proglacial lake boundary into an ice sheet model**

**by Sebastian Hinck, Evan J. Gowan, Xu Zhang, and Gerrit Lohmann**

Dear Kerim Nisancioglu,

we are glad to submit our revised manuscript, based on the thoughtful comments and criticisms by the three reviewers.

With this re-submission we provide

- the revised manuscript,
- the revised Supplements,
- a document highlighting all changes made on the manuscript, and
- an updated version of our point-to-point responses to each reviewer (this document),

as required by the journal.

For the revision we conducted several new sensitivity runs, testing different aspects of the model, as demanded by the reviewers. These are documented in detail in the supplementary material and referred to in the main text. Apart from the changes suggested by the reviewers, we rephrased some sentences and checked for grammar and spelling mistakes. Furthermore, we added some relevant references, which helped to improve the manuscript and interpret the results.

In the following, we will present our responses to all comments of the reviewers and provide, where applicable, the respective changes made in the revised manuscript. We hope our revised manuscript will further be considered for publication in *The Cryosphere*.

Yours sincerely,

Sebastian Hinck on behalf of all co-authors

**Author response to referee comment RC1 by Torsten Albrecht (PIK) to manuscript No. tc-2020-353, "PISM-LakeCC: Implementing an adaptive proglacial lake boundary into an ice sheet model"**

**submitted to The Cryosphere by Sebastian Hinck et al.**

**General comments:**

> Hinck and colleagues investigate the role of ponded proglacial lakes along the margin of the Laurentide Ice Sheet during glacial retreat. Assuming that, similar to marine-terminating glaciers (e.g. in Antarctica), the adaptive boundary conditions influence the (land-lake-terminating) ice sheet's stress balance in various ways. They can alter the ice flow (ice streams) and hence the overall ice sheet's geometry and stability. The changing ice load, in turn, results in the isostatic adjustment of the underlying lithosphere, which hence affects the formation (and demise) of lakes with up to several hundred meters depth in the vicinity of the ice sheet. By considering this feedback in a coupled model system applied to the North American ice complex the authors find in some lake regions self-amplified deglacial retreat, similar to what is commonly discussed for the Antarctic Ice Sheet in terms of the marine ice sheet instability (MISI). The significance of this "lake"-effect, comparing the four conducted experiments, is in deed surprising. Hence, the scientific insights of this study would be certainly a valuable contribution to the paleo ice sheet modelers community within the scope of "The Cryosphere".

> The authors created a method based on a simple and efficient 4-neighbor "connected components (CC)" labelling algorithm that determines ocean and multiple lake basins for a given (or processed) bed topography and estimate the corresponding water levels by iterating over a set of increasing water levels, without including computational-expensive flow routing techniques. The standalone models "LakeCC" and "SL2dCC", adapted from Hinck et al., 2020, were implemented in the open-source Parallel Ice Sheet Model (PISM), which already comes with an solid-Earth deformation module. As the title of this study suggests, the ice model's marine boundary conditions have been generalized for this lake-coupling procedure. This does not mean that water density and calving rates have been simply adjusted, but that a whole PISM sub-module has been created with many functions and special considerations. For numerical stability reasons, the authors consider prescribed (ad hoc) lake filling rates that permit gradually evolving lake water levels for changing bed topography and ice margins.

> The focus of this study is the description of the model implementation into PISM with a very detailed technical Appendix (which would also fit well to the Geoscientific Model Development) and to run simple deglacial simulations to test for its relevance as compared to the default case without the LakeCC method. From a modeler's point of view, this study could benefit from a few more sensitivity tests, that could help disentangling the individual contributions of the relevant processes acting at the lake-ice-bed boundary ("modification of the thermal regime at the submerged ice base, formation of ice shelves, increased ice loss due to melting and calving, and enhanced basal sliding near the grounding-line due to decreased effective pressure at the ice base... "). Hence, the reader would not only learn "that lakes matter" but also "why lakes matter". Also, the authors state, that the simulated ice sheet margins do not match well with reconstructions based on geological evidence, which is not the focus of the study. However, I encourage the authors to follow some of my suggestions in the specific comments below (mainly regarding the initialization of the LC bed deformation model and the non-linear precipitation dependence on temperature index), which likely can improve the simulation outcome.

> Overall, the study is well structured and the manuscript clearly-arranged. The draft consists of 36 pages including 9 main figures, 12 Appendix pages and 57 references.

We would like to thank Torsten Albrecht for reviewing our paper. The original comments are indented, while our responses aligned to the left of the page.

For this review round several sensitivity experiments were run, which are referred to in the revised manuscript and supplementary materials.

**Specific comments:**

drainage events

> *l. 22:* "Reorganization of the lakes' drainage networks and sudden drainage events due to the opening of lower spillways may have impacted the global climate by perturbing the thermo-haline circulation system of the oceans (Broecker et al., 1989; Teller et al., 2002)."
> and *l. 131:* "For simplicity the fill-rates in our model are assumed to be constant."
> and *l. 142:* "If a basin disappears because it merged with the ocean, the lake level is gradually changed until sea level is reached, and then removed."

> → This is super exciting, but is it correct, that due to numerical stability requiring gradual lake filling, such events would be prohibited (or at least smoothed over long times) in this implementation?

The implementation of proglacial lakes as described in this work only affects the direct ice - lake interaction within PISM. Modeling the impact of redistributions of freshwater onto the climate system would require modeling and coupling Earth's different subsystems. For coupling the PISM-LakeCC model to an ocean model, translation of the water levels into freshwater fluxes would be necessary. This task, however, is not trivial, as the LakeCC model does not conserve water volume. Furthermore, water fluxes would have to be calculated by comparing the water distribution between two time slices. Any changes in water level or drainage route that happened in between those steps would not be resolved.

The sentence you are referring to from the introduction (l. 22) was added to highlight the further potential impact of proglacial lakes on the climate system.

One way to include data from the PISM-LakeCC model in a coupled Earth System model, could be by using the information of lake basins. Water fluxes from different sources (e.g. precipitation, ice sheet run-off) could then be accumulated within these basins and extracted from the water cycle. By redistributing the collected water volumes the changing topographic setting could be attributed for. Excess water can then again be added as a freshwater flux into the climate system. However, the complexity of accounting for all of the water sources is beyond the scope of this paper.

To mention further missing interactions of lakes with the environment, that are not modeled by the LakeCC model, we have added a section to **Sec. 2.4 Limitations (l. 259 ff)**:

**"2.4.3 Further lacustrine interactions**

**Our lake model explicitly only interacts with the ice dynamics, as described above. However, there are ways a lake impacts the various subsystems of the Earth system, which can not be solved for within an ice sheet model. Some of these mechanisms, which are ignored here are briefly mentioned in this section.**
**The presence of vast proglacial lakes impacts the local climate by reducing temperatures and increasing precipitation patterns (Krinner et al., 2004; Peyaud et al., 2007). This can locally increase the ice sheet's SMB and counteract the accelerated mass loss observed in this study. Also, the potential impact on ocean circulation and global climate due to redistribution of freshwater (Broecker et al., 1989; Teller et al., 2002; Condron and Winsor, 2012) is ignored here.**
**The location where proglacial lakes appear strongly depends on the GIA signal. The GIA signal is also partially dependent on the mass held by the lakes. This feedback, however, is not taken into account here."**
* * *
lake merging

> *l. 127:* "However, there are special cases, such as adding a lake basin to an existing lake or adding a basin that has previously been connected to the ocean, that need more advanced treatment. For more details on this, see Appendix B."
> *l. 134:* When water level is rising, this common level is chosen to be the lowest water level of that lake, hmin, while the highest level, hmax, is selected for the falling water level.
> *l. 139:* "Only when h has exceeded the current (local) lake level, does its value get updated."

> → What exactly is the current (local) lake level? A simple sketch could help here (maybe added to Fig. 1?). An explanation as in Hinck et al., 2020 may help:

> > "Until a patch merges with another one that is a sink, i.e. the lake overflows, the current level h is stored for all associated cells."

We have added the lake level in the legend of Fig. 1. Probably "current (local)" lake level was confusing, as we just meant the lake level. Therefore we have shortened this description in the new manuscript.

The quote from the Hinck et al. (2020) paper that you mention describes a different context. There we describe the LakeCC algorithm, which is used here to determine the `target_level`, while here we describe the gradual filling algorithm.
* * *
2d sea level

*l. 157: " to determine the two-dimensional sea level field"*

→ I recommend to mention here early in the manuscript that, although the sea-level in PISM is treated as a 2D variable, in this application the value in each cell in every time step equals either a global constant or NaN.

Yes, this is a good point. We have added the following to the description of the ice sheet model, *l. 104 ff*:

**"Marine regions of the ice sheet are defined in PISM via a flotation criterion, which describes whether the ice in areas below sea level is grounded or floating. The sea level is defined via a 2D map, which allows it to be spatially variable. In general, however, it is set to a global mean value prescribed by a scalar time series."**
* * *
*l. 522: "Here, we present the implementation of a sea level modifier, which takes advantage of possibility of a spatially variable sea level..."*

→ The manuscript could benefit from some motivation, why it is not sufficient to just update the already available 2D sea-level field with the various target lake levels (as general water level) and simply add a 2D field of corresponding water densities? I guess the numerical instabilities requiring a gradual lake filling (and hence more fields as the lake level) are one important argument for it, what else?

We have added a short paragraph about it to Appendix B, *l. 492 ff*:

**"A lake, as seen by an ice sheet model, is very similar to a locally elevated sea level. It would be possible to implement the lake interface as a sea level modifier in PISM. However, physical differences (e.g. density, temperature distributions) require different treatment of marine and lacustrine environments. To distinguish between both cases, at least two spatial maps are required. This can either be done by providing the lake and sea level combined in one field and additionally providing a mask, or providing lake and sea level elevations as separate maps. For the implementation of the PISM-LakeCC model, we chose the latter case."**

For us it seemed more clean to have these things on different fields. Gradual filling should not have been a problem, as these secondary fields are used only internally of the lake model/modifier anyway. It was just a design question and we chose the other path.
* * *
grounding line treatment

*l. 92: "The till below the water level next to the ice margin and grounding-line are assumed to be saturated. " and l. 172 "Cells with grounded ice below water level next to a lake are assumed to have saturated till. This reduces the effective pressure at the base of the ice sheet and reduces basal resistance in this location."*

→ I fully agree that this is a valid model choice. However, the underlying assumption of having saturated till within one grid cell length upstream of the grounding line (here 20km) has often been criticized within the community. Apparently, it highly increases the ice sheet's sensitivity (e.g., Golledge et al., 2015), in particular in combination with the basal melt interpolation (has this also been used)? For better comparison with other model studies it would be helpful to state the expected consequence of this model choice ("higher sensitivity"). Generally, this study could very much benefit from a few sensitivity runs in order to attribute the relative effects of some of the different processes named in Sect. 2.3 (and Fig. 2). Another (maybe more physical) way would be initializing all water-covered cells as saturated (https://github.com/pism/pism/pull/425), such that an advancing grounding line would not get temporarily stuck on initially unsaturated till (until it reaches saturation), which surprisingly seems to have a similar effect on paleo time scales, also for grounding line retreat (Albrecht et al., 2020a; Fig. A2).

Yes, this is an important point. We have conducted two more experiments targeting the sensitivity of the grounding line treatment (see experiments *TWO* and *nSG* in the Supplements). These experiments confirm this high sensitivity.

In *TWO* the "till water ocean" model (PR425) was used instead of the slippery grounding line treatment. The ice sheet evolves in this scenario almost identical to the *lcc* run (which uses the slippery grounding line treatment).

In *nSG* the slippery grounding line treatment was deactivated. E.g. the till water of cells at the grounding line was not modified. This strongly impacts the glacial retreat, as the grounding line flux, and marine and lacustrine discharge are strongly reduced.

In all experiments the sub-grid grounding line treatment as activated by the `-pik` option are used. This also included the interpolation of basal melt.

We added the following paragraphs to the "Limitations" section, *l. 234 ff*:

"For our experiments, the so-called slippery grounding line parameterization (1) has been used, which reduces the basal friction in the cells upstream the grounding line. This simple model therefore adds a direct dependency to the grid resolution. Furthermore, an increased sensitivity of the grounding line is reported when using this parameterization (Golledge et al., 2015). For this reason, sensitivity tests were run that confirm these findings; these results are presented in the supplementary material and discussed in Sect. 4.3. Due to lack of a more advanced implementation to model lubrication at the grounding line, we apply the slippery grounding line treatment."

and Discussion *l. 425 ff*:

"Sensitivity runs (see supplementary material) have shown that the observed response is strongly impacted by the grounding line treatment. If the basal resistance at the grounding line is not reduced, the grounding line flux is strongly decreased. This becomes aparent in reduced mass loss and slower retreat of the ice margins. Nevertheless, when compared to the CTRL experiment, the impact on ice flux and margin retreat is clear."

marine vs. lake boundary conditions

*l. 165: "Generally, the parameterizations provided by PISM for a marine boundary are applied analogously at the lake interface. This, however, may not be the optimal treatment in every case. The model might benefit from future implementations of advanced or more specialized lake boundary treatment. Such limitations are discussed in Sect. 2.4."*
and *l. 175: " the same parameterization for ice base temperature and sub-shelf mass-flux are used."*

→ As this marine assumption may overestimate sub-shelf melt rates in lacustrine environments, it would be helpful to state about what average melt rates we talk here, and what its relative contribution to the ice sheet mass balance is (aggregated rates) and on the enhanced deglacial retreat.

We have investigated the different contributions of mass loss for our experiments (see the figure for the LAKE experiment). *Note, the LAKE experiment was formerly called LCC!*

[Figure]

*Different contributions to mass loss for the LAKE experiment. The top blue line shows the surface accumulation, while the lower black line shows the overall rate of change of the ice mass (The net surface mass balance is SMB = Accumulation - Runoff). The plot shows that the main process governing mass loss is surface runoff (= melt - refreeze) followed by glacial (marine) discharge (i.e. calving). Sub-shelf melting in lakes contributes only up to 6% to the total (non-surface) mass losses when large lacustrine ice shelves are present (15 kyr).*

In the LAKE experiment typical (depth dependent) sub-shelf melt rates in lakes are between 100 and 200 kg m$^{-2}$ year$^{-1}$.

For the impact on the deglacial scenario please see the next point!

*l. 234: "At the shelf base (2), mass flux is parameterized using the model proposed by Beckmann and Goosse (2003)."*

→ In this paragraph the authors mention that the melt pump is expected to be weaker in fresh water due to lower buoyancy, while it could be also stronger as the temperature at the grounding line is likely higher than in marine environments. What estimate provides the Beckmann-Goosse model when the default values (35 g/Kg and -1.7°C?) were changed (0 g/Kg and +4.0°C) accordingly? Would the effective melting be higher or lower than for the marine default values and could this still be realistic (even though it was designed for marine environments)?

To address this point we conducted an experiment (*MR*) where we tried to estimate the relative difference between melt rates in marine and lacustrine settings using the melt pump parameterization. Details can be found in the Supplements. We added the following paragraph in the Methods section *l. 243 ff*:

**"Due to the difference in density between fresh and ocean water, the layer of melt water underneath the ice shelf experiences a ~ 200 times higher buoyancy in seawater (Funk and Röthlisberger, 1989). In saline water this increases the flux and mixing of ambient warmer water along the ice base and increases melting. Based of these considerations, the melt flux was scaled accordingly in the MR sensitivity run, which is documented in the supplementary material."**

The impact of sub-shelf melting on the ice sheet evolution becomes even smaller when applying this simple lake correction. The results are very similar to the *LAKE* scenario. The largest contribution to the ice sheets mass balance is from surface runoff, which is further increased due to the ice surface lowering upstream the lake boundary. This is discussed in the Discussion section, *l. 440 ff*:

**"Sensitivity experiments (see supplementary material) show that this rapid disintegration of the LIS is mainly driven by the strongly negative mass balance along the southern ice margin. Calving and sub-shelf melt rates play only a secondary role here. These findings are in agreement with Quiquet et al. (2021), who refer to this instability as PLISI."**

calving

*l. 247: "Implementation of a more physically based calving model capable of accurately parameterization of mass losses in both lacustrine and marine environments, will be needed (Benn et al., 2007)."*

→ The authors are also using the Eigencalving parameterization, which applies well for ice shelves in rather confined embayments, as fund around present-day Antarctica. Can you roughly estimate the relative contributions of thickness calving or Eigencalving in this study (maybe by comparing the two experiments with different thickness calving thresholds)?

For the newly conducted experiments we checked the contribution of the Eigencalving to glacial discharge and realized that for our experiments it is always zero. We added a footnote to *l. 111*:

**"Although the Eigen-calving mechnism was activated for all experiments, the model was found to not have impacted the results."**

Ice discharge at the ice margin is thus solely due to the thickness calving parameterization. To better compare the impact of the thickness calving threshold on the glacial retreat another experiment (*RedCalv*) with reduced threshold ($\Delta h=20m$) was conducted. Note, the experiment formerly called *LNS* is now labeled *IncCalv* and treated as the other sensitivity runs and described in the Supplements!

The ice loss rises roughly linearly with the thickness calving threshold $\Delta h$:

[Figure]

*Mass losses due to lacustrine calving for different experiments. Note, experiments have been renamed in the revised manuscript: lcc -> LAKE; lns -> IncCalv.*

Comparing the results of the *LAKE* (formerly *LCC*) and *RedCalv* experiments, the glacial retreat is almost unchanged. The main difference is that in the *RedCalv* experiment the lacustrine ice shelf extent is a few cells (up to 100km) wider, which exhibits a larger ice surface area to surface melting. This approximately balances the difference in ice loss due to calving. A comparison of both calving experiments (*IncCalv* and *RedCalv*) with the *LAKE* experiment is done in the Supplements.
* * *
> l. 285: "...large ice shelves like those seen in the LCC experiment are unlikely to have existed."

→ Is there some reference? Does "large" mean covering entire lakes? Or would this imply that really thin (∆hL = 50m, l. 281) ice shelves are in fact unrealistic.

By large we mean shelves that extent several grid cells from the grounding line (e.g. 100's of km). We are not aware of any reference investigating the potential size of ice shelves on large proglacial lakes. Furthermore, we are not aware of geological evidence that would support the existence of such vast ice shelves. In general, geomorphologically constrained reconstructions of glacial lakes do not depict ice shelves that extend deeply into the ice sheet, as is simulated in our experiments (e.g. Veilette, 1994; Teller and Leverington, 2004; Lemmen et al., 1994).

Our recent sensitivity experiments suggest that the formation of ice shelves is sensitive to grounding line treatment. Whether or not the currently implemented conditions are realistic for proglacial lakes should be a target for future studies. We have added the following to the Discussion *l. 443 ff*:

**"We want to note that the vast ice shelves that can be observed at this late phase of the collapse of the LIS (Fig. 5 & 8c-d) might be unrealistic. At least we are not aware of any geological evidence or reconstruction, indicating such ice shelves existed. It is possible that the simulated grounding line flux is too high or the calving rate too low. Two sensitivity runs checking these parameters (IncCalv an nSG in the supplementary material) do not exhibit such large lacustrine shelves. However, more research is necessary to decide how to best parameterize these issues."**
* * *
numerical instability and time stepping

> l. 219: "Sudden jumps in water level can trigger numerical instabilities in the ice sheet model"

→ Can you say some more words on the possible reasons for numerical instabilities? Is this due to large areas of grounded ice becoming afloat at once affecting the non-local KSP iterative solution?

We are not sure about the exact reasons for this instability. Error messages indicating that the iterative KSP solution does not converge. We think that jumps in the boundary conditions (as they are obviously introduced when the water level is immediately changed) are just incompatible when numerically solving systems of differential equations. If the relatively slow gradual filling of lake basins is considered to be a major drawback of this model, higher fill rates could be realized in future implementations by requesting smaller time steps from PISM's adaptive time stepping mechanism.

We have added a new sub-section to the Limitations section 2.4, *l. 269 ff*:

"**2.4.4 Gradual filling & numerical instability**

**When running the model, we observed that PISM occasional crashes when there is rapidly rising or dropping the water levels between time steps. Unfortunately, we can not tell the exact reason why the numerical solver failed, but we found that these crashes only appear shortly after the new lake or ocean basins, which are in contact with the ice, appear or vanish. From a numerical point of view, however, this is not surprising, as the numerical representation of a physical system requires the underlying equations to be smooth functions. These crashes, due to local unsteadiness in the fields, are known by the developers of PISM to appear (https://pism-docs.org/wiki/doku.php?id=kspdiverged, accessed: 2021/08/09).**
**With all the simplifications described above, the LakeCC model is not designed to exactly reconstruct the water level of proglacial lakes, nor to estimate freshwater fluxes. It rather gives a crude approximation of where potential lake basins are located and what their maximum water level is. Therefore, we assume the gradual filling mechanism using a (rather arbitrarily chosen) constant fill rate does not result in lake levels that are less valid.**
**If an almost instantaneous filling or draining of lake basins is desired, this could be achieved by setting a high fill rate γ. This would most likely require a reduction of the duration of the time steps, though, which increases the computational cost. In sensitivity tests, which are found in the supplementary material, several higher values were chosen. For these tests, no model crashes were observed. This might either be because we restricted the time step to 0.25yr for all experiments, or simply because no critical situations were triggered. The results of these experiments show no fundamental difference to the LAKE experiment. Future implementations could target a more realistic treatment of basin filling.**"
* * *
> *l. 83: "In this study we set an upper bound of 0.25 yr for the time step." and l.222: "Future implementations could possibly adapt a volumetric rate instead of fixing the rate of change of water level. By limiting the time step, sudden changes in water level could be performed quicker.*

> → Please better motivate this particular time step. Does this choice help with keeping numerical stability with regard to the constant lake filling rate (l. 131)? Is this a best-practice value or is there some relationship between temporal and spatial resolution, as for instance the famous CFL criterion? Is the adaptive time step of PISM's sub-modules for the used resolution usually larger?

Adaptive time stepping strongly depends on the model's state in PISM. In times when many ice shelves exist, ice flow is usually higher and therefore the adaptive time step is reduced accordingly (e.g. to ensure the CFL criterion). If the ice flows relatively slow, time step lengths can even be higher than one year. The main reasons for reducing the time step length is better comparability between different experiments:

- as we are comparing the results from different experiments we want the time step for all experiments to be as similar as possible. There is a known dependence of PISM model results dependent on when the model is evaluated.
- to estimate the efficiency of the LakeCC model (Fig. C1), we compare the efficiency of the different PISM runs. Time step length in PISM, and thus runtime of the model to finish, strongly depends on the model state (see previous point). To get the number of model evaluations for Lake and no-Lake experiments in a similar range, we manually reduced the time step.
- smaller time steps reduce the potential risk of the numerical instability of occurring (although by choosing an appropriate fill rate this might not be necessary)

We have added the following lines to the description of the experimental Setup, *l. 324 ff*:

"**For all experiments, the maximum time step of the ice sheet model was limited to 0.25 yr. We want to stress that this setting is not requisite by the LakeCC model, but we had several reasons to do so. Estimating the computational overhead of an algorithm (as done in Sec. C) requires the time steps of the compared runs to be similar. Another benefit from limiting the time step is that the occurrence of numerical instabilities due excessively large jumps in water level (see Sec. 2.4.4) is potentially reduced.**"
* * *
> solid-Earth feedback

> *l. 93: "Bed deformation due to ice load is modeled in PISM using the Lingle-Clark model ..." and*
> *l. 210: " If higher resolved input fields are available, e.g. from an external GIA model, the LakeCC model could be modified to do the calculations on that field instead and interpolate the output back onto the ice sheet model grid."*

> → PISM has been recently coupled to a global solid-Earth model (VILMA, not published yet). Therefore, PISM (https://github.com/pism/pism/pull/463) can read the history of bed topography change on the ice model grid relative to a (high resolution) reference topography. The real benefit of this external GIA over the internal LC model, however, is that it self-consistently solves for the sea-level equation, i.e. it accounts for self-gravitational effects, which can be very relevant in (deglacial) grounding line migration in marine (or lake) environments. My guess is that lakes could be quite easily included, but this may require a closed water budget between ice sheet, lakes and ocean (l. 213). In any case, this would be rather an option for a follow-on study.

Yes, indeed, this would be an interesting future study. Although, implementing a closed water budget between ice sheet and lakes would require substantial changes to our Lake model.
* * *
> > *l. 259: "The temporal evolution of the topography provided by NAICE can be used to calculate the uplift rates that are used to initialize the Lingle-Clark bed deformation model of PISM."*
>
> → The NAICE model makes use of a GIA model (which also solves the sea level equation) constrained with many different paleo data types, but it makes use of very simple assumptions on the steady ice state and boundary conditions. Please, provide some more information on how and when the NAICE uplift rates were used. It reads as you would take the LGM state and uplift rates from NAICE and run the Lingle-Clark model from there up to the present day? If this is true, I would expect for a different GIA (bed deformation) model with different mantle viscosities used, that this would imply a (almost equilibrated) bed topography at present, which may differ from what we observe today, even if the ice thickness would be perfectly reproduced. Can you quantify the misfit in bed topography in your study?

We actually do use the LGM state and uplift rates from NAICE and used this for initialization of the LC model. The misfit between the topography state at 21kyr from the *LAKE* (formerly *lcc*) experiment and the PD topography from RTopo2 is shown in the following plot.

[Figure]

*Topography anomaly between 21kyr from the lcc experiment and present day RTopo2 data.*

In the north-western part, where there is still ice in the *LAKE* experiment, topography is still deeply depressed, while south-eastern Canada and the northern US are strongly over-relaxed. The letters mark the points for which the temporal evolution is plotted in the figure below.

[Figure]

*Temporal evolution of the bed deformation relative to PD for different locations. The locations are marked by the colored letters in the above map.*

Our sensitivity runs, which are all documented in the Supplements, contain one experiment (*GIA*) where an Earth parameterization, similar to the one used by NAICE, was used. Apart from slight changes in timing, no substantial differences compared to the *LAKE* experiment are observed.
* * *
> *l. 365 "After deglaciation, the Hudson Bay region is over-relaxed and above PD sea level (see Fig. 8f)."* and
> *l. 342: " Our deglacial scenario fails at simulating realistic ice margin positions for the western LIS."*

> → A probably better way to avoid such a large misfit would be to initialize the Lingle-Clark model from present-day geometry and uplift rates and run it into the LGM state, from which you then start the experiments. Or if you want to make use of the constrained NAICE results at LGM you could make use of the simulated misfit at present an rerun each experiment with the initial bed topography adjusted according to the misfit, such that you end up with a better match at present in the second iteration.

Our initial plan was also to start from a LGM state calculated from PISM. I.e. simulating the glacial inception from a PD-like geometry (as was done in Niu et al., 2019). However, it turned out that this modeled ice sheet was unrealistically too large and caused bed depression that was much greater than reality. During the rapid retreat the modeled bed response was too slow, so that along the southern ice margin the topography was below sea level and connected to the Atlantic ocean. The presence of this ocean basin inhibited the formation of lakes along the ice margin in this region, which is the main focus of this study. Therefore we decided to start our simulations from a more geologically constrained LGM state.

We added a short paragraph to the discussion section, *l. 413 ff*:

**"Also, the initialization of the Lingle–Clark model from a glaciated state is problematic here. For the model to calculate a relief topography, to which bed deformation is applied, it should ideally be initialized from an interglacial state when the residual GIA from previous glaciations is limited. Test runs, comparable to Niu et al. (2019), however, suffered from the fact that the bed deformation along the southern ice margin was so deep, that the basin was connected to the Atlantic Ocean, which consequently inhibited the formation of lakes. We therefore chose to initiate the experiments from NAICE LGM reconstructions. The mismatch in calculated relief topography, results in ice free regions to over-relax. Hudson Bay, for example, is elevated above sea level at PD (see Fig. 8f)."**

We further followed the suggestion and did an experiment (*dtopg*) in which we subtracted the PD misfit from the initial LGM topography. The results are shown in the supplementary document (https://doi.org/10.5281/zenodo.4746502) and the following figure.

[Figure]

*Overview map of the dtopg experiment at 9kyr simulation time. The basin at the southern ice margin is connect to the ocean and thus not available for lakes.*

In this experiment the final PD topography fits relatively well with the RTopo2 data, but we have similar problems as described above. During glacial retreat the large basin at the southern ice margin is connected to the ocean and thus inhibits the formation of lake basins. We therefore did not follow this approach any further. Also, we did not include this experiment in the revised manuscript or Supplements.
* * *
> *l. 361: "Maximum water depths close to the grounding-line are up to 1000 m. The main reason for this, we assume, is the GIA response modeled by PISM's bed deformation model."*

> → What if the surface mass balance is simply overestimated (see my comments about index method), which "tends to accumulate too much ice" (l. 363), such that the Lingle-Clark model simply responds (in the correct way) to the higher load?

The reasons for the high lake depth at the grounding line are manifold. One reason is the strong bed depression, which is the reaction to the increase of ice mass due to the simple climate parameterization. Furthermore, comparison with sensitivity experiment *nSG* shows that the parameterization of basal friction at the grounding line impacts the grounding line flux. Less friction leads to dynamical thinning, which in return allows the glacial lake to further penetrate underneath the ice sheet, causing high water depths.

We have rewritten the paragraph in the revised manuscript, *l. 404 ff*:

**"Another issue, which is also partly related to drainage, is the immense size and depth of some lakes that appear along the southern ice margin. From around 6 to 9 kyr one large lake occupies the entire Great Lakes region. Later, it transitions into the basin of Lake Agassiz/Ojibway (Teller and Leverington, 2004) before expanding into Hudson Bay (~ 13 – 18 kyr) (Fig. 8a-d). Maximum water depths close to the grounding line are up to 1000 m. As a result of the increased accumulation of ice due to the simplified climate forcing, the topography is further depressed. The deeply depressed topography in combination with lowered ice thickness, due to dynamic thinning, allows the lake to further expand underneath the ice sheet."**
* * *
> l. 362: "This simple, two layered Earth model is not capable of handling the extreme deglaciation scenario of an entire continent."
>
> → I think this statement is a bit harsh, as for the assumptions made, the Lingle-Clark model in fact can handle glacial cycles over an entire continent comparably well. I would agree that it is simple as it uses only one spatially constant mantle viscosity and does not account for self-gravitational effects, which may play a large role here. But my guess would be that the proper initialization of the Lingle-Clark model may bring much improvements here (see comment above).

Yes, the formulation might sound too harsh. However, the simple two-layered Earth model neglects the large influence of the much higher viscosity lower mantle, which play an important role for determining bed response the central parts of the Laurentide Ice Sheet. The lack of a lower mantle contrast means that the depression will be overestimated, and the response to loading will be too fast. We have rephrased this paragraph and added the reference for our claim, *l. 410 ff*:

**"Furthermore, for such deglacial scenario, use of a more advanced GIA model might be needed. PISM's default model is based on a simple two-layered Earth model. Viscosity variations in the upper and lower Earth mantle are not included in this model, even though these variations significantly contribute to the GIA signal in central Canada (Wu, 2006)."**
* * *
> l. 364: "the rebound was not quick enough."
>
> → The PISM default value for the upper mantle viscosity (likely used in this study) is 10^21 Pas. NAICE, for instance, used a lower value of 4x10^20 Pas for the upper mantle, which implies a faster rebound. According to the relevance of GIA in this study, one or two sensitivity tests with varied mantle viscosity could bring some more interesting insights.

Using comparable Earth model parameters as used by NAICE (see experiment *GIA* and comment above) does not significantly change the results. Only the timing of the glacial retreat changes slightly compared to *LAKE*.
* * *
climate forcing

> l. 269: "To prevent ice sheet growths ... above 3500 m elevation, ice accumulation is prevented by setting precipitation to zero in these regions." And also
> l. 338: "Only by limiting precipitation above 3500 m elevation ... does further expansion of the ice sheets stop. "
>
> → What is the motivation for this constraint? Is this related to the findings (of maximum surface elevation) in the NAICE model? Or is it just gained experience with the model setup?

At high elevations very little precipitation is expected. In the simple glacial index climate model the decrease of precipitation with temperature, which decays with elevation based on a simple lapse rate approach, is calculated. However, with our input data this still resulted in accumulation of mass even at high elevations. This simple cut-off model is an easy workaround to reduce the ice sheet growth. The choice of the cutoff height (3500m) was rather ad-hoc. However, several LGM ice sheet reconstructions (e.g. NAICE, ICE-6G) indicate that most of the LIS was below this threshold. Only in few high mountainous areas the ice surface is slightly higher. Note, that PISM and we refer to surface elevations relative to PD sea level/geoid.

To clarify, the following paragraph was added to the revised manuscript, *l. 303 ff*:

**"To prevent ice sheet growths in eastern Siberia and above 3500 m elevation (relative to PD sea level), the precipitation to zero in these regions. The maximum elevation threshold was chosen in an ad-hoc fashion. However, current LGM reconstructions (e.g. NAICE (Gowan et al., 2016), and ICE-6G (Peltier et al., 2015)) exhibit higher surface elevations only in few mountainous regions."**
* * *
> l.96: "The surface mass balance (SMB) is estimated from monthly means of precipitation and surface air temperature fields... Precipitation and temperature fields are interpolated between two distinct climatic states, which are weighted according to a glacial index (see Appendix D1). "
>
> → Does this mean that the interpolation is between each month of the two climate states?

In principle yes. If the input fields are time dependent, they are assumed to be periodic over one year and each field is treated as piecewise linear in time. When accessing the temperature and precipitation fields for time t from the glacial index model, the respective LGM and PD fields are evaluated at t (one year periodicity) and used for interpolation. Therefore, the calculated climate fields do have a seasonal cycle.
* * *
> l. 295: "Starting from the LGM initial state after the spin-up, the ice sheets laterally expand and gain mass for about 6 kyr. Within this time, the LIS almost doubles its volume, before it retreats during the rest of the simulation." and also
> l. 332: "...dynamical equilibrium, as the initial ice sheet was reconstructed using geological evidence of ice margin history and GIA observations, and no ice dynamics were included (Gowan et al., 2016)."
>
> → A model drift after initialization is typical. A dynamic equilibrium simulation (with constant dry LGM climate conditions) prior to actual forcing experiments could help here identifying relevant parameter settings that counteract the lateral expansion.

The growth of the ice sheet could also be related to a discrepancy between the modeled climate from equilibrium simulation using NAICE, and how a dynamic ice sheet reacts to the modeled climate, so it is not surprising that the ice sheet could grow after starting the simulation (e.g. the climate forcing might be too cold). However, as the goal of our experiments is simply to test how proglacial lakes affect the ice sheet, and we successfully simulate deglaciation, further investigation of the cause of this growth is not relevant to this study.
* * *
> l.335: "The reason for the ice accumulation is assumed to be due to the climate forcing. By linearly interpolating between the warm, humid PD and the cold, dry LGM climate states, unrealistically high accumulation is produced, especially in cold regions and on top of the ice sheet."
>
> → This seems to be quite some considerable imbalance for applied linearly changing climate forcing. I agree that the interpolation is the most likely candidate here, as precipitation is rather non-linearly related to temperature change (Frieler et al., 2015). This means that precipitation after 6kyr would be much overestimated by the arithmetic mean (index 0.5) between LGM and PD state.

In future studies a more appropriate climate forcing should be applied. For our model test case, however, this simple forcing is acceptable.
* * *
SLE unit

> l. 295: " sea level equivalent (SLE) ice volume" and
> Fig. 4 caption: "...and also includes ice shelves"
>
> → I assume that the authors use SLE as a converted unit of grounded ice volume. If so, please specify the used conversion factor and also mention, that you do not make use of a 'volume above flotation' definition (for good reasons), to avoid misunderstandings.

We have added a paragraph to the beginning of the results section, **l. 336 ff**:
**"In the following, all expressions of time are given in model years, relative to the start of the experiments, and ice sheet volume is quantified as sea level equivalent (SLE). SLE denotes the rise of the water level, if the ice volume $V\_IS$ would melt and the equivalent freshwater volume $V\_fw$ is added into a basin of the mean ocean area $A\_O = 3.625 \cdot 10^8$ km^2 :**

**$\Delta$ SLE = $V\_fw$ /$A\_O$ = $V\_IS$ * $\rho\_i$/$\rho\_fw$ * $A\_O$,**

**where $\rho\_{i,fw}$ are the respective densities of ice and freshwater. Note, this formulation neglects any changes in the ocean surface area and the geoid due to mass redistribution. Furthermore, the ice sheet volume accounts for both grounded and floating ice."**

The caption of Figure 4 was adapted accordingly:
**"[...] Ice volume is given in sea level equivalent (SLE) units; see main text for definition. [...]"**
* * *
ice lobes

> l. 300: "...ice margin between about 1 and 3.5 kyr is due to formation of small ice lobes that advance into small lakes (compare also with Fig. 6). "
>
> → Is the ice margin at those ice lobes grounded or floating? Maybe provide some definition here? Is there a difference to an "ice shelf tongue"? There is a reference quite later in the text (l. 375).

The ice margin here is grounded. Thin floating ice tongues would immediately be calved off by the thickness calving parameterization.

Ice lobes (i.e. thin, broad glaciers that extend from the main core region of the ice sheet) are a common occurrence along the terrestrially terminating margin of the Laurentide Ice Sheet (Margold et al 2015). These lobes are grounded and are often associated with ice marginal lakes. So these features are not analogous to marine terminating glaciers.

> ice streams
>
> > l. 360: "The formation of ice streams .."
>
> → Does a speed-up of ice flow imply that there wasn't an ice stream before, i.e. in terms of confined ice flow with speeds above (let's say) 100 m/yr?

As was also mentioned by the other reviewers, the speedup of the ice does not imply the formation of a confined ice stream. In this context it might be more correct to call it an increase of ice flow/velocity. We have rephrased this in the revised manuscript.

> channel filter
>
> > l. 504: "Finally, to get rid of narrow lakes, which are often caused by the under-resolved topography, the target level is filtered by applying the FilterLakesCC method using a filter size of Nfilter."
>
> → Why would this be an issue? Does it mean, that a drainage event can only occur for a channel opening of width larger than Nfilter?

This filtering scheme helps removing few-cell-wide lake basins, which often appear due to resolution issues in rough topography (i.e. river valleys that are under-resolved and incorrectly are filled as a lake). It is not a requirement to apply this method. As we are mainly interested in the interactions between ice sheet and the major lakes, we decided to remove them. Wet boundary grid cells potentially trigger faster ice flow, which require smaller time steps. Therefore, limiting the number of lake cells can be beneficial in terms of time step length.

The filtering method does not modify the underlying topography and thus does not impact the drainage routes of other lakes. It only labels and removes narrow basins in an extra step, after the lake mask was determined.

**Technical corrections:**

Thank you for all the suggestions and corrections. If not stated otherwise, these were corrected in the revised manuscript.

> > l. 11 and l. 70 "...at the end of the simulation"
>
> → "... at present-day"

> > l. 28: "These processes can lead to the formation of ice streams..."
>
> → or: "...speed-up of ice streams..."

> > l. 30 "Due to various differences between the freshwater and ocean water, the interactions might be different than at a marine boundary (Benn et al., 2007)."
>
> → "Due to various differences between the freshwater and ocean water **regime** , the interactions **at the lake-ice boundary** might be different than at a marine boundary (Benn et al., 2007)."

> > l. 32 "In times when there was the production of large amounts of meltwater that caused the formation of proglacial lakes..."
>
> → "In **periods with** large amounts of meltwater that caused the formation of proglacial lakes..."

> > l. 34: " caused by rapid demise"
>
> → " caused by **a/the** rapid demise"

> > l. 36: "governed by the negative surface mass balance due to a warming climate"

→ As there are counteracting effects on the surface mass balance (also more precipitation for higher temperatures), it would be good to be more precise here, i.e. "due to **enhanced melting in** a warming climate"
* * *
*l. 40: "Lake reconstructions of this time suggest water depths up to several hundred meters (Teller et al., 2002; Leverington et al., 2002)."*

→ This has been already stated in l. 20, but that's ok.
* * *
*l. 46: global **mean** sea level*
* * *
*l. 56: "The latter reference discusses concerns when implementing such a lake-ice boundary condition for ice sheet models."*

→ "The latter referenced article discusses **challenges for the implementation of** such a lake-ice boundary condition for ice sheet models."
* * *
*l. 68: "In places where there is ice-inward sloping topography, the lake can rapidly expand, as the water is deep enough that the ice begins to float."*

→ "In places where the water is deep enough, the ice begins to float. In regions of ice-inward sloping topography, the grounding line can retreat in a self-amplified manner, which corresponds to a rapid expansion of the lake."
* * *
*l. 109: "...quite dynamic" → "...particularly dynamic"*
* * *
*l. 110: "Therefore, when dynamically coupling ice sheets and proglacial lakes, steady updating of the geometry is necessary. "*

→ " **For the dynamical** coupling **of** ice sheets and proglacial lakes **the continuous update** of **their geometries** is necessary. "
* * *
*l. 112: "continental **-** sized "*
* * *
*l. 113: " trade-offs have to be **made** "*
* * *
*Fig. 1 caption: "before it overflows into the ocean **or a neighboring lake basin** ."*

The target level is the maximum water level before a lake basin overflows into the ocean. This often also means that drainage is routed via other lake basins into the ocean, but this is not what is meant here. Overflowing into another lake basin can also mean that two lakes merge, but in that case the filling algorithm is not complete. We think that it is less confusing to just say that it is overflowing. Where the water is routed to is irrelevant here.
* * *
*l . 183: "The pressure difference against the submerged ice margin is done analogously to the marine boundary"*

→ but evaluated for the fresh water density
* * *
*l. 207: "a after applying a secondary field."*

→ As far as I understand this means "applying a correction" or "adjustment" of the low resolution data. Is this just an "anomaly field" added"?

Yes, basically we determine the anomaly between the PD smoothed and "minimum - filtered" topography, which is then added as a correction. We rephrased this.

> *l. 259 "that are used  to initialize..."*
* * *
> *l. 286: "... and Ctrl, without lakes. The PISM default run Def produces..."*
>
> → Would it be possible to use some different font for the experiment names?

Yes, all experiment names are written in italics in the revised manuscript.
* * *
> *l. 293: "topographic setting"*
>
> → maybe "configuration" would also fit here
* * *
> *l. 299: "Fig. 5 "*
>
> → Figure 5
* * *
> *l. 300: "southward - shifted"*
* * *
> *l. 311: "Where ice streams diverge, the surface velocity anomaly partially shows a slowdown of ice flow."*
>
> → Please be more precise, where is this and why?

We removed the sentence. In the discussion we mention these features again.
* * *
> *l. 320: "... in **the** ice volume evolution of the LIS"*
* * *
> *l. 325: " CIS" probably means Cordilleran Ice Sheet, please define.*

That's correct. We have defined it in the revised manuscript!
* * *
> *l. 350: " **The r** econstruction"*
* * *
> *l. 368: " large response"*
>
> → "high sensitivity"
* * *
> *l. 369: " upstream **of** the lake boundary"*
* * *
> *l. 371: " due to the changes in ice dynamics, the different experiments constantly diverge and features visible in these plots might therefore only indirectly be triggered by the lake boundary, but rather be a result of differences in ice sheet geometry."*
>
> → Maybe split into two sentences. What is diverging?

The sentence has been simplified. ***L. 424 f***:

**"It should be noted that some features in these plots also arise from the fact that the ice geometries of the experiments diverge."**
* * *
> *l. 389: "ice - covered"*

> *l. 392: " retreat **s** "*

> > *l. 406: " continental ice streams"*
>
> → What does this mean? Is this related to the size?

We mean, non-marine-terminating ice streams. We rephrased this in the revised manuscript, ***l. 458 f***:

**"The lake model promotes the acceleration of ice flow where the ice margin is bounded by a lake, which leads to a lowering of the upstream ice surface."**

> *Table A1 and A2:* It would be nice for other modelers, if you could emphasize the options, which were added compared to the PISM base code (A1: index,precip_cutoff, lake_level (lakecc), (,sl2dcc), A2: - lakecc_dz, -lakecc_zmin , -precip_cutoff_height , -use_precip_cutoff_height).

The table entries have been marked by a footnote.

> > *l. 427: "Connected Components Algorithm"*
>
> → capital or not (e.g., l., 457)? But in any case be consistent through the manuscript.

> *l. 442: "about **the** presence"*

> l. 442: I think, a common name in numerics for this approach at the boundary is "ghost point method"

Ok, thanks for the hint. We added it to the revised manuscript.

> > *l. 453: "Furthermore, two additional 2D diagnostic variables, which can be added to PISM's output, were added (see Table B1)"*
>
> → Please be more precise. Do you mean the two variables, which are not "general" in Table B1? Maybe use the word "optional"?

We have rephrased this. ***L. 514 f***:

**"Optionally, the two 2D diagnostic variables lake_depth and lake_level_real can be added to PISM's output (see Table B1)."**

> l. 465-473: A simple schematic map including such cases (just some grid cells) could be helpful to understand the conditions in this paragraph.

We have added such a figure to the manuscript.

cc_sketches **"Figure B1. Sketches depicting the different algorithms implemented. a) SeaLevelCC – only basins below sea level and connected to either of the domain margins is considered ocean. b) IsolationCC – Lake basins fully enclosed by the ice are considered invalid. c) FilterLakesCC – Lakes are only retained if they contain at least one cell with a certain amount of lake neighbors. In the example N = 4 was chosen. d) LakePropertiesCC – This algorithm collects the minimum and maximum water levels for each lake basin. e) FilterExpansionCC – This algorithm compares the current lake mask with the previous mask and labels newly appeared and vanished lake cells. It distinguishes between narrow strips (purple) and wider basins (dark blue)."**

> *l. 468: "ice - covered"*

> *l. 522: "of **the** possibility "*

> *l. 528: "model **to** identify"*
* * *
> *l. 530: "Only patches that are connected to the margin of the computational domain are considered to be part of the ocean."* and also
> *l. 467: "but are not connected to either of the domain margins"*

→ As the southern margin is land, I guess the condition would be "margin below sea level" or so?

Yes, correct. However, if the margin is above sea level, it would not be part of the patch.
* * *
> *l. 551: "usees "*

**References**

- Albrecht, T., Winkelmann, R., Levermann, A., 2020. Glacial-cycle simulations of the Antarctic Ice Sheet with the Parallel Ice Sheet Model (PISM) – Part 1: Boundary conditions and climatic forcing. The Cryosphere 14, 599–632. https://doi.org/10.5194/tc-14-599-2020

- Gowan, E.J., Tregoning, P., Purcell, A., Montillet, J.-P., McClusky, S., 2016. A model of the western Laurentide Ice Sheet, using observations of glacial isostatic adjustment. Quaternary Science Reviews 139, 1–16. https://doi.org/10.1016/j.quascirev.2016.03.003

- Hinck, S., Gowan, E.J., Lohmann, G., 2020. LakeCC: a tool for efficiently identifying lake basins with application to palaeogeographic reconstructions of North America. Journal of Quaternary Science 35, 422–432. https://doi.org/10.1002/jqs.3182

- Lemmen, D.S., Duk-Rodkin, A. and Bednarski, J.M., 1994. Late glacial drainage systems along the northwestern margin of the Laurentide Ice Sheet. Quaternary Science Reviews, 13(9-10), pp.805-828. https://doi.org/10.1016/0277-3791(94)90003-5

- Margold, M., Stokes, C.R., Clark, C.D., 2015. Ice streams in the Laurentide Ice Sheet: Identification, characteristics and comparison to modern ice sheets. Earth-Science Reviews 143, 117–146. https://doi.org/10.1016/j.earscirev.2015.01.011

- Niu, L., Lohmann, G., Hinck, S., Gowan, E.J., Krebs-Kanzow, U., 2019. The sensitivity of Northern Hemisphere ice sheets to atmospheric forcing during the last glacial cycle using PMIP3 models. Journal of Glaciology 1–17. https://doi.org/10.1017/jog.2019.42

- Peltier, W.R., Argus, D.F., Drummond, R., 2015. Space geodesy constrains ice age terminal deglaciation: The global ICE-6G_C (VM5a) model. Journal of Geophysical Research: Solid Earth 120, 450–487. https://doi.org/10.1002/2014JB011176

- Quiquet, A., Dumas, C., Paillard, D., Ramstein, G., Ritz, C., Roche, D.M., 2021. Deglacial Ice Sheet Instabilities Induced by Proglacial Lakes. Geophysical Research Letters 48, e2020GL092141. https://doi.org/10.1029/2020GL092141

- Teller, J.T. and Leverington, D.W., 2004. Glacial Lake Agassiz: A 5000 yr history of change and its relationship to the δ18O record of Greenland. Geological Society of America Bulletin, 116(5-6), pp.729-742. https://doi.org/10.1130/B25316.1

- Veillette, J.J., 1994. Evolution and paleohydrology of glacial lakes Barlow and Ojibway. Quaternary Science Reviews, 13(9-10), pp.945-971. https://doi.org/10.1016/0277-3791(94)90010-8

- Wu, P., 2006. Sensitivity of relative sea levels and crustal velocities in Laurentide to radial and lateral viscosity variations in the mantle. Geophysical Journal International 165, 401–413. https://doi.org/10.1111/j.1365-246X.2006.02960.x

📖 Reviewer2.md

**Author response to referee comments RC2 & RC3 by Tijn Berends to manuscript No. tc-2020-353, "PISM-LakeCC: Implementing an adaptive proglacial lake boundary into an ice sheet model"**

**submitted to The Cryosphere by Sebastian Hinck et al.**

> The authors describe a set of simulations of the Laurentide Ice Sheet (LIS) during the last deglaciation, and the effect that proglacial lakes have on the evolution of the ice sheet. They used the well-established PISM ice-sheet model, including an additional module that dynamically tracks the extent and depth of proglacial lakes. By allowing ice shelves to form on these lakes, and by including some additional parameterisations for processes such as basal sliding, calving, and basal melt, they claim to have included all the ways a proglacial lake can affect the dynamics of the adjacent ice sheet. Their results show that the inclusion of these lakes in their model strongly accelerates the retreat of the LIS, in a manner similar to the phenomenon of Marine Ice-Sheet Instability (MISI). While the authors do not mention this in their manuscript, this is an important conclusion, as the asymmetry in the Pleistocene glacial cycles (slow inception vs. fast deglaciation, and particularly the melt-water pulses) is something that's still not fully understood.
>
> I think a study like this could be very interesting, and could contribute to our understanding of glacial dynamics. However, there are several issues with the methodology which I believe impact the validity of the conclusions. In particular, two important feedback processes (the lake-climate-SMB feedback and, more importantly, the geoid-MISI stabilisation) are not included, both of which would reduce the accelerated retreat the authors observe. I will detail these concerns below, after which I'll list the smaller technical questions I have.

We would like to thank Tijn Berends for thoughtful comments on our manuscript.

We would like to point out that we do not claim to "include all of the ways a proglacial can affect the dynamics of the adjacent ice sheet", and we explicitly state the shortcomings of our model (e.g. the lack of freshwater-ice interactions, the gradual filling and emptying of the lakes for model stability purposes, and the lack of a lake-specific calving law). We also would like to emphasize that the experiment we present is purely designed to test how lakes interact with the ice sheet during deglaciation. We acknowledge that in many ways it is not a realistic deglacial scenario, and it was never designed to be.

For this review round several sensitivity experiments were run, which are referred to in the revised manuscript and supplementary materials.
* * *
> 1. The lake-climate-SMB feedback. At least two studies I know of, namely Krinner et al. (2004, Nature) and Peyaud et al. (2007, Climate of the Past), have looked at the effect proglacial lakes had on the local climate. Both find a net positive effect on the surface mass balance of the adjacent ice sheet, stabilising it against retreat. This would at least partially negate the acceleration due to grounding line dynamics described by the authors.

This is indeed a very interesting topic! We have neglected this issue so far, because the impact of the lake on the climate, only has an indirect impact on the ice sheet. However, even if this feedback is important, we consider it as beyond the scope of our study, since our experimental setup is not coupled to a climate model.

Adding such feature would require development and testing of a coupled model setup, which does not yet exist. Furthermore, regular coupling of the ice sheet model to a climate model would add significant computational overhead.

In Earth System models, and especially ice sheet models, there are lots of feedbacks that are not included or crudely approximated. Most previous studies have entirely neglected the impact of proglacial lakes (on the ice dynamics and on the climate). The purpose of our paper is to introduce a lake model that changes this, and provides a further improvement to the way we model ice sheets.

We use a simple index forcing method in order to test our lake model, and therefore there is no feedbacks between the lakes and the climate. We added a sub-section, mentioning some processes which are not represented in our setup, under Limitations, *l. 259 ff*:

**"2.4.3 Further lacustrine interactions

**Our lake model explicitly only interacts with the ice dynamics, as described above. However, there are ways a lake impacts the various subsystems of the Earth system, which can not be solved for within an ice sheet model. Some of these mechanisms, which are ignored here are briefly mentioned in this section.**

**The presence of vast proglacial lakes impacts the local climate by reducing temperatures and increasing precipitation patterns (Krinner et al., 2004; Peyaud et al., 2007). This can locally increase the ice sheet's SMB and counteract the accelerated mass loss observed in this study. Also, the potential impact on ocean circulation and global climate due to redistribution of freshwater (Broecker et al., 1989; Teller et al., 2002; Condron and Winsor, 2012) is ignored here.**

**The location where proglacial lakes appear strongly depends on the GIA signal. The GIA signal is also partially dependent on the mass held by the lakes. This feedback, however, is not taken into account here."**
* * *
> 2. The geoid-MISI stabilisation. The authors refer to Weertman 1974 for proof of the Marine Ice-Sheet Instability (MISI). However, Weertman's proof that no stable equilibria exist for ice sheets whose margins lie on retrograde slopes did not account for GIA, nor for changes in the geoid. While the authors included a simple GIA module in their ice-sheet model, they did not account for changes in the geoid. Several different studies (the work of Natalya Gomez is probably the most important for the geoid, and that of Valentina Barletta for GIA) have shown that the fall in sea level at the ice margin, caused by the loss of ice mass in the interior, strongly reduces the retreat rate, and can even lead to stable equilibria on retrograde slopes. This has been proposed as an explanation for why the rapid West Antarctic retreat predicted by MISI is not really visible in paleo evidence. Since the authors claim (in my view correctly) that the strongly accelerated retreat of the LIS in their model is due to the same instability, I believe it is crucial to take this feedback into account.

Also this issue is very interesting! However, we are bound to the model implementations currently available in PISM. There are efforts to include a more realistic mode of GIA into PISM (e.g. computed by an external solid-Earth model, VILMA), but it is not yet publicly available. Coupling PISM to an external GIA model requires lots of work and testing.

We agree, that the results could benefit from a more advanced GIA model (and we state this in the original and new manuscript), but we are not sure about the relevance of gravitational effects at the lacustrine boundary. Since our current model does not include the gravitational effect of the ice sheet, we would assume the water depth to increase at the lake ice boundary, which would further accelerate the instability. Regardless, the Earth deformation will be by far the largest part of the GIA signal for the deglaciating Laurentide Ice Sheet, and this component is included in the Lingle-Clark model.
* * *
> The fact that these two processes, particularly the geoid effect, are not included, leads me to believe that this study significantly overestimates the lake-induced acceleration of LIS retreat.

Gravitational attraction would increase the pull of water towards the ice sheet. When the ice volume decreases, it would decrease that attraction, but the magnitude of this would undoubtly be smaller in magnitude than the residual rebound (which is included in the LC model), and geometry changes in the lake due to margin fluctuations. The ice volume changes that contribute to changes in gravitation are relatively slow compared to these factors. Therefore we do not believe that our model will "significantly" overestimate the lake-induced acceleration. Considering all of the other limitations of our current implementation, the lack of gravitational attraction is not likely to be largest source of error in our simulations.

> Aside from this, I also have a number of small, technical questions, which I will list here.
* * *
> L29: "...ice streams, which impact the mass balance..."

> Don't you mean the ice dynamics? Mass balance is usually meant to include only surface and basal mass gain/loss.

The mass balance (not surface mass balance!) of an ice sheet is the net balance of mass accumulation and losses. Ice streams transport mass from the inner ice sheets towards the ice margins, where ice losses are highest. Therefore, we consider our statement as correct.
* * *
> L34: "...the 8.2ka event was caused by..."

> Too confident. While there certainly is strong evidence for this, I wouldn't say the matter is entirely settled.

Yes, the formulation was too confident. We have reformulated this sentence, *l. 32 ff*:

**"In periods with large amounts of meltwater that caused the formation of proglacial lakes the lake-ice interactions might have been a key factor to explain rapid glacial retreat. As an example, one hypothesis for the cause of the 8.2ka event, a period characterized by a sudden drop in Northern Hemispheric mean temperatures, is the rapid demise of the central LIS (Carlson et al., 2008; Gregoire et al., 2012; Matero et al., 2017; Lochte et al., 2019) leading to a large freshwater input into the Labrador Sea."**
* * *
> *L84: "The stress balance is modeled using a hybrid scheme based on the Shallow Ice (SIA) and Shallow Shelf Approximations 85 (SSA) of the full Stokes equations (Bueler and Brown, 2009)"*

It is well known that these hybrid models perform poorly at simulating grounding line migration. Many models now include a semi-analytical solution for the grounding-line flux as a boundary condition, but as far as I know this has not yet been implemented in PISM, and it is not discussed anywhere in the manuscript. While I can't say if this would lead to an over- or an underestimation of ice-sheet retreat in this particular study, I think it is important to discuss this, since grounding line dynamics are the root cause of all your results.

In PISM the position of the grounding line is determined by applying the flotation criterion. The exact position is further refined using an sub-grid interpolation scheme. Using these schemes, even at relatively coarse resolution grounding line position is reasonably well represented (Feldmann et al., 2014). PISM does not include a model to prescribe the grounding line flux, and its implementation is beyond of the scope of this study.

Another related issue is the basal friction at the grounding line. In our standard setup we set the till in cells next to the grounding line as saturated, which decreases the overburden pressure and thus reduces the basal friction. In sensitivity tests without this parameterization (see experiment *nSG* in the Supplements) we could confirm the sensitivity of the grounding line position as it was also reported by Golledge et al. (2015).

We have added a paragraph about the grounding line treatment to the Limitations section, *l. 234 ff*:

**"For our experiments, the so-called slippery grounding line parameterization (1) has been used, which reduces the basal friction in the cells upstream the grounding line. This simple model therefore adds a direct dependency to the grid resolution. Furthermore, an increased sensitivity of the grounding line is reported when using this parameterization (Golledge et al., 2015). For this reason, sensitivity tests were run that confirm these findings; these results are presented in the supplementary material and discussed in Sect. 4.3. Due to lack of a more advanced implementation to model lubrication at the grounding line, we apply the slippery grounding line treatment."**

and to the Discussion, *l. 425 ff*:

**"Sensitivity runs (see supplementary material) have shown that the observed response is strongly impacted by the grounding line treatment. If the basal resistance at the grounding line is not reduced, the grounding line flux is strongly decreased. This becomes aparent in reduced mass loss and slower retreat of the ice margins. Nevertheless, when compared to the CTRL experiment, the impact on ice flux and margin retreat is clear."**
* * *
> *L88: "The basal resistance is determined using a model that assumes that the base of the ice sheet is underlain by deformable till."*

Since you explicitly state that the effect of lakes on basal sliding is important, this seems an oversimplification. The distribution of regolith in North America is far from uniform, and the interplay between (erosion and transport of) regolith, basal sliding, and glacial dynamics has been studied for over two decades (e.g. Clark and Pollard, 1998). If you really want to present the effect of lakes on basal sliding as an important factor, then I believe a more elaborate approach is needed.

We are in the process of implementing a more elaborate basal conditions model in PISM that takes into account changes in sediment distribution and grain size (e.g. Gowan et al., 2019). While we agree that the differences in basal conditions could change the outcome of the simulations, we do not believe it would fundamentally alter our conclusion that proglacial lakes affect ice sheet dynamics. Once this new basal conditions model is complete, we will attempt this test.
* * *
> *L102: "The marine boundary treatment is described in Winkelmann et al. (2011) and Martin et al. (2011). It includes a sub-shelf-melting parametrization..."*

If I'm not mistaken, this basal melt parameterisation was developed specifically for the Filcher-Ronne and Ross shelves in Antarctica. There, basal melt is mostly related to the intrusion of relatively warm deep water into the cavity between the ice shelves and the continental shelves, which leads to the depth-dependence in this parameterisation. I don't believe this translates well to the situation in Lake Agassiz. Since the authors show that calving plays in important role in their glacial dynamics, I suspect sub-shelf melt (which ultimately affects grounding line dynamics just as much as calving does) is equally important, and deserves a more accurate treatment than this. However, whether this oversimplification leads to an over- or underestimation of ice-sheet retreat, I cannot say.

To address this point we conducted an experiment (*MR*) where we tried to estimate the relative difference between melt rates in marine and lacustrine settings using the melt pump parameterization (Beckmann and Goosse, 2003). Details can be found in the Supplements.

Melt rates are strongly dependent on the assumed mean temperature of the lake. Furthermore, melting depends on the pressure and thus depth dependent freezing point. For fixed lake temperature and depth we estimated the effectiveness of melting in marine relative to lacustrine environments. Using T=2°C and d=300m marine melting is estimated to be about 40 times stronger. This factor is used to scale the melt rate tuning parameter accordingly. Compared to the *LAKE* experiment, the impact of sub-shelf melting on the ice sheet evolution is minor when applying this simple correction for lake melt rates.

We have added the following paragraph to the Methods section, *l. 243 ff*

**"Due to the difference in density between fresh and ocean water, the layer of melt water underneath the ice shelf experiences a ~ 200 times higher buoyancy in seawater (Funk and Röthlisberger, 1989). In saline water this increases the flux and mixing of ambient warmer water along the ice base and increases melting. Based of these considerations, the melt flux was scaled accordingly in the MR sensitivity run, which is documented in the supplementary material."**

Since you mention the importance of calving on the ice sheet evolution, we shortly discuss the results of sensitivity run *RedCalv*, with reduced lacustrine calving with Δh=20m. For this scenario we expected the glacial retreat to be slower that in the *LAKE* run, but found that this changed thickness calving threshold hardly impacts the retreat pattern. Only in the extreme scenario (*IncCalv* (formerly *Ins*) with Δh=500m) does lacustrine calving become the dominant process.

To get an overview about the different contributions to the ice sheet's mass balance we have prepared a plot (here for the *LAKE* (formerly *lcc*) experiment).

[Figure]

*Different contributions to mass loss for the LAKE experiment. The top blue line shows the surface accumulation, while the lower black line shows the overall rate of change of the ice mass (The net surface mass balance is SMB = Accumulation - Runoff).*

The plot shows that the main process governing mass loss is surface runoff (= melt - refreeze) followed by glacial (marine) discharge (i.e. calving). Sub-shelf melting in lakes contributes only up to 6% to the total (non-surface) mass losses when large lacustrine ice shelves are present (15 kyr).

The experiments with reduced calving rate *RedCalv* and adapted sub-shelf melt rate *MR* show that the driving process of the ice sheet instability is neither calving nor sub-shelf melting. The driving process is rather surface melting due to the surface elevation feedback. The same was recently also found by Quiquet et al. (2021). We added the following paragraph to the revised manuscript, *l. 440 ff*:

**"Sensitivity experiments (see supplementary material) show that this rapid disintegration of the LIS is mainly driven by the strongly negative mass balance along the southern ice margin. Calving and sub-shelf melt rates play only a secondary role here. These findings are in agreement with Quiquet et al. (2021), who refer to this instability as PLISI."**

> *L111: "Depending on the complexity of the lake model, computational overhead can drastically increase."*

> I wonder if you considered using the flood-fill algorithm which I specifically developed for this kind of application (Berends and van de Wal, 2016). I've been using this for a while now, included in a 40km resolution ice-sheet model (solving for lakes at a 1 km resolution), and running full glacial cycle simulations is no problem at all (~60h computation time, including the SELEN sea-level model).

Yes, we actually took your algorithm into consideration when developing our lake module. However, if I understood correctly, each lake requires a seed element, which we wanted to omit in our algorithm, since through a full glacial cycle, the locations that might need a seed could change. Our lake filling algorithm takes only a trivial amount of computation time, and provides a reasonable lake configuration even at lower resolutions (Hinck et al 2020). We found the main increase in computation was due to lakes causing the ice velocity to increase, necessitating the increase use of the SSA model.

> *L117: "However, rapid changes in the boundary conditions resulting from this approach often cause numerical instabilities which cause the model to crash."*

> What kind of numerical instabilities are these? I've never encountered this problem myself. It sounds like something that should be addressed in the numerical solver of the ice-sheet model itself, rather than by compromising on the lake-filling code. Regarding this compromise: exactly how fast do you move the lake level to the "target level", and how does this impact your results?

I have to admit, that I am not an expert on the numerical solver of PISM (it is based on PetSc). The model crashed with a message that the stress balance solver failed (KSPSolve). This problem is, however, not unique to using the LakeCC model, but also appear under other circumstances (see https://pism-docs.org/wiki/doku.php?id=kspdiverged). In our case, we could narrow the cause down to either abruptly forming or draining lake or ocean basins. This causes a rapid change in ice geometry and stress boundary condition. It is not surprising that a numerical model, which is based on the assumption of steadily evolving physical system, can not cope with such sudden jumps at the boundary. In my opinion this issue can not be solved within the numerical solver , but could be tackled by accordingly reducing the numerical time step to maximize the speed in which the water level is gradually adapted. The default value, which was used in the experiments is 1m year^-1 (this parameter choice was rather ad-hoc). We conducted some further test runs (*FR5*, *FR10* and *FR50*, see Supplements), where this value was increased to 5, 10 and 50 m year^-1, respectively. Surprisingly, all runs finished without any problem and the results do not show any major differences to the *lcc* run. Experiment *FR50* already comes pretty close to an immediate response of the lake level to a change in the target level. We have to state that the appearance of the instability is strongly dependent on the configuration of the ice sheet when the model is evaluated. That is to say, for these experiments we might simply have been lucky that no critical situation was triggered and that slight changes in the ice sheet configuration might crash the numerical solver even at a lower fill rate.

We do not see this gradual adaption of the water level as a big problem, since uncertainties in water level, ice geometry and missing closed hydrological cycle are at least of the same order. Consequently, lake reconstruction from this model can not be used for direct use in a coupled Earth model. This, however, has never been the intention of our modeling approach. We only focus attention to direct ice-lake interactions, which were missing in most ice sheet modeling studies.

We have added a sub-section about the numerical instability and gradual filling to the Limitations section, ***l. 269 ff***

**"2.4.4 Gradual filling & numerical instability**

When running the model, we observed that PISM occasional crashes when there is rapidly rising or dropping the water levels between time steps. Unfortunately, we can not tell the exact reason why the numerical solver failed, but we found that these crashes only appear shortly after the new lake or ocean basins, which are in contact with the ice, appear or vanish. From a numerical point of view, however, this is not surprising, as the numerical representation of a physical system requires the underlying equations to be smooth functions. These crashes, due to local unsteadiness in the fields, are known by the developers of PISM to appear (https://pism-docs.org/wiki/doku.php?id=kspdiverged, accessed: 2021/08/09).

With all the simplifications described above, the LakeCC model is not designed to exactly reconstruct the water level of proglacial lakes, nor to estimate freshwater fluxes. It rather gives a crude approximation of where potential lake basins are located and what their maximum water level is. Therefore, we assume the gradual filling mechanism using a (rather arbitrarily chosen) constant fill rate does not result in lake levels that are less valid.

If an almost instantaneous filling or draining of lake basins is desired, this could be achieved by setting a high fill rate γ. This would most likely require a reduction of the duration of the time steps, though, which increases the computational cost. In sensitivity tests, which are found in the supplementary material, several higher values were chosen. For these tests, no model crashes were observed. This might either be because we restricted the time step to 0.25yr for all experiments, or simply because no critical situations were triggered. The results of these experiments show no fundamental difference to the LAKE experiment. Future implementations could target a more realistic treatment of basin filling."
* * *
> L155: "Therefore, the use of a more advanced sea level model is necessary. This sea level model implemented here."

This is not a sea level model. Sea level is, in your setup, prescribed externally with the glacial index method. What you're describing here is a routine that determines the ocean mask.

This is correct. We were rather referring to the naming of this sub-system within PISM. We have rephrased the respective paragraph as follows (*I. 158 ff*):

**"As described in Hinck et al. (2020), the LakeCC algorithm relies on a proper land-sea mask to properly determine lake basins. This mask can be computed by the SL2DCC model [...]"**
* * *
> L202: "For hydrological applications, such as lake basin reconstructions, the resolution an ice sheet model usually operates on is too coarse to resolve spillways through the terrain. Even more important than data resolution is the ice margin position and bed deformation due to GIA (Hinck et al., 2020). Considering the uncertainties of these fields retrieved from an ice sheet model, the resolution issue is regarded as a secondary issue."

I'm not sure I agree here. Determining lake extent in a low-resolution DEM leads to a systematic overestimation of lake water volume (since you'll always underestimate the depth of drainage channels), which Berends and van de Wal (2016) showed to be around 10% for a resolution of 20km (this is why I developed my own algorithm!). This might not be much, and I don't it's something that should be fixed right away, but since it's an overestimation that's there throughout the simulations, it's something to keep in mind when you start looking at sea-level jumps and the likes.

This is indeed a valid point, we added a sentence acknowledging the systematic overestimation of lake volume to the revised manuscript. However, given the large uncertainties in the modeled glacial topography, we don not see this as the main problem. Focus of this study is not to perfectly model the water volumes, but to determine potential positions where proglacial lakes might have existed and impact the ice dynamics.

*I. 206 ff*:

**"At low resolutions, lake volumes are potentially overestimated (Berends and van de Wal, 2016). Since we are not so much interested in lake volume, the ice margin position and bed deformation due to GIA are, in our case, expected to be more important than resolution (Hinck et al., 2020)."**
* * *
> L219: "Sudden jumps in water level can trigger numerical instabilities in the ice sheet model. To avoid such jumps, the water level 220 is gradually adjusted with a constant rate."

Again, what do you mean by this? And why are you so sure that it is the sudden jumps in water level (which, if you look at sea-level records of the 8.2 kyr event, are definitely hinted at) that are unrealistic, rather than the behaviour of your numerical solver?

The basic LakeCC model (as it was described in Hinck et al.(2020)) does calculate potential lake basins for a given ice and topography setting. This means, changes in these fields cause the sudden appearance or disappearance of lake basins. If these changes happen at ice covered grid cells, the ice sheet at these spots could suddenly become afloat or ground. This marks sudden and extreme changes for the stress boundary condition. The sea level record you are referring to does show rapid jumps, but these are never 10's of meters in one time step. We are not saying that sudden jumps are unrealistic, but even in reality they are steady and not immediate. Even the final drainage of Lake Agassiz at the 8.2 ka event likely happened over the course of centuries (Gautier et al 2020), so gradual filling and emptying might not be so unrealistic in reality. There might be better ways to handle this, but all would certainly include gradually adapting the water level. For more details see our previous replies.

> *L234: "The model relates the mass flux to..."*
>
> Could say anything about water temperatures in the lake? Krinner et al. (2004) find bottom water temperatures < 4∘C in a proglacial lake that's frozen over 7 – 11 months per year. How does this compare to your parameterisation?

We refer here to our sensitivity experiment *MR* (see Supplements), where we tried to find parameters for the melt pump model (Beckmann and Goosse, 2003) more suitable for lacustrine settings. However, the model assumes constant mean water temperatures. A simple seasonal ice cover (and corresponding buttressing) could be added by making use of the ice melange parameterization of PISM. This has already been mentioned in the manuscript. This would, however, require parameterization of the lake temperatures.

> *L249: "Another important issue that is ignored in our model is the effect of ice mélange..."*
>
> It was my understanding that ice mélange buttressing is only relevant in fjords and such, where the convex coastlines can provide a backpressure to the mélange, which in turns pushes against the shelf front. Do you think this plays a significant role on the open water of Lake Agassiz?

We agree that buttressing might not be relevant for large open lakes as Lake Agassiz. But it might be a missing effect to reduce mass flux into smaller lakes. As also mentioned in the previous reply, this parameterization could also be used to mimic the presence of a seasonal ice cover.

> *Fig. 3: "The shaded area shows the regions attributed to the (continental) Laurentide Ice Sheet (LIS) in this study."*
>
> I don't understand what you mean by this.

For discussing the temporal evolution of a spatially bound ice sheet, we have to define a region within which ice masses are attributed to it. By "continental" we emphasize that we have excluded the parts of the LIS in the Canadian Arctic Archipelago from our discussion. We have added the two following sentences to the Results section, *l. 332 f*:

**"The highlighted area in Fig. 3 marks the region of interest for this study. In the following, when referring to the LIS, we will consider only the parts of the ice sheet that are within this region."**

> *L269: "To prevent ice sheet growths in eastern Siberia and above 3500 m elevation, ice accumulation is prevented by setting precipitation to zero in these regions."*
>
> This seems rather ad-hoc. Given that (at least in the ICE5G and ICE5G reconstructions) large parts of the LIS interior are above 3500m, how do you think this affects your results?

We want to state that we are not talking about ice thickness, but rather surface elevation relative to the present day geoid, here. Ice sheet reconstructions Ice6G and NAICE both exhibit LGM surface elevations above 3500m only in few high mountainous grid cells.

We added the following paragraph. *L. 303 ff*:

**"To prevent ice sheet growths in eastern Siberia and above 3500 m elevation (relative to PD sea level), the precipitation to zero in these regions. The maximum elevation threshold was chosen in an ad-hoc fashion. However, current LGM reconstructions (e.g. NAICE (Gowan et al., 2016), and ICE-6G (Peltier et al., 2015)) exhibit higher surface elevations only in few mountainous regions."**

> *L272: "Transient sea level forcing is applied accordingly to the glacial index."*

> This seems like an oversimplification, given that your main conclusion is rooted in grounding line dynamics. Even without using a geoid model, you could at least let eustatic sea level be calculated dynamically.

Our simulation is only meant as an idealized experiment. We have not included the other ice sheets, so calculating sea level change in this way is not possible. Regardless, the configuration of the lakes in continental North America is not dependent on sea level (since the lakes form above sea level), and therefore prescribing sea level in this way does not affect our results. As mentioned by Reviewer #1, coupling PISM with a proper GIA model is underway, so such an experimental setup will be available in the future. At that point, we can test this.
* * *
> L274: "Before running the experiments, the model needs to be spun-up."

> How do you initialise englacial temperature? And why do you want the ice sheet to be in equilibrium with the prescribed climate before you start your simulations? Just as the real climate is never in a "steady state", so the real ice sheet would never have been in equilibrium with the climate, but always lagging behind it. It seems more logical to avoid these questions by starting your simulation in the Eemian interglacial.

Yes, we agree that the ice sheet can not be expected to have been in equilibrium with the climate at LGM. Our spin-up does only aim to thermo-dynamically equilibrate the ice sheet. Therefore, PISM is run in a fixed geometry state until the internal enthalpy field are in better agreement with the energy fluxes prescribed by the boundary conditions.

Initially we planned to start from an interglacial state and run the full cycle (similarly as it was done in Niu et al. (2019)), but the resulting LGM ice sheet and bed deformation were too large. The post-glacial topography was depressed so deeply that large parts of Canada were below sea level and connected to the Atlantic Ocean, so that lakes could not form. Since studying the impact of proglacial lakes on the ice sheet dynamics in the main focus of this study, this setup was inappropriate, and we decided to initialize the experiments from a more geologically constrained LGM state.

We added the following paragraph to the Discussion section, *l. 413 ff*:

**"Also, the initialization of the Lingle–Clark model from a glaciated state is problematic here. For the model to calculate a relief topography, to which bed deformation is applied, it should ideally be initialized from an interglacial state when the residual GIA from previous glaciations is limited. Test runs, comparable to Niu et al. (2019), however, suffered from the fact that the bed deformation along the southern ice margin was so deep, that the basin was connected to the Atlantic Ocean, which consequently inhibited the formation of lakes. We therefore chose to initiate the experiments from NAICE LGM reconstructions. The mismatch in calculated relief topography, results in ice free regions to over-relax. Hudson Bay, for example, is elevated above sea level at PD (see Fig. 8f)."**
* * *
> L282: "...in order to reflect the fact that calving rates for freshwater terminating glaciers are reported to be an order of magnitude lower than rates observed for tidewater glaciers..."

> I thought part of the reason why tidewater glaciers experience more calving is because of the tides after which they're named, which cause increased crevassing, as well as more wave action and other goings-on that weaken the ice. Do you think Lake Agassiz is more similar to the ocean, or to a small mountain lake, in that regard?

I guess a large lake as Lake Agassiz can neither directly be compared to the ocean nor to a small mountain lake. I would expect higher waves than in a small lake, but smaller waves compared to the ocean. Tides will not play a major role in an enclosed water body (e.g. the tides in the modern Great Lakes are less than 5 cm https://oceanservice.noaa.gov/facts/gltides.html).
* * *
> L285: "large ice shelves like those seen in the LCC experiment are unlikely to have existed"

> Why not? Do you cite any studies that support the existence of an open lake? Has IRD not been found in sediment cores, or has it never been looked for? Ice shelves are hard to track in proxy evidence, so I wouldn't be too quick to dismiss them.

We are not aware of any reference investigating the potential size of ice shelves on large proglacial lakes. Furthermore, we are not aware of geological evidence that would support the existence of such vast ice shelves. In general, geomorphologically constrained reconstructions of glacial lakes do not depict ice shelves that extend deeply into the ice sheet, as is simulated in our experiments (e.g. Veilette, 1994; Teller and Leverington, 2004; Lemmen et al., 1994).

Our recent sensitivity experiments suggest that the formation of ice shelves is sensitive to grounding line treatment and calving condition. Whether or not the currently implemented conditions are realistic for proglacial lakes should be a target for future studies. We added the following paragraph to the Discussion section, *l. 443 ff*:

"We want to note that the vast ice shelves that can be observed at this late phase of the collapse of the LIS (Fig. 5 & 8c-d) might be unrealistic. At least we are not aware of any geological evidence or reconstruction, indicating such ice shelves existed. It is possible that the simulated grounding line flux is too high or the calving rate too low. Two sensitivity runs checking these parameters (IncCalv an nSG in the supplementary material) do not exhibit such large lacustrine shelves. However, more research is necessary to decide how to best parameterize these issues."
* * *
> L296: "Within this time, the LIS almost doubles its volume..."
>
> This sounds rather problematic. Why are your initial ice sheet and prescribed climate so far out of equilibrium? Aren't there any tuning parameters in your PDD scheme to correct for this? Also, looking at Fig. 4, your initial state has a volume of about 40 m SLE, which seems rather small for the Laurentide – I believe 60 – 80 m is a more commonly accepted number. (e.g. ICE5G, ICE6G).

Our initial ice sheet is taken from the NAICE reconstruction. It is not in equilibrium state calculated by PISM, we therefore expect a model drift after initialization. Directly after initialization, the climate is shifted from the cold and dry LGM state towards a warm and humid PD state. On top of the ice sheet temperatures are still cold due to the temperature-elevation drop, but precipitation increases as the PD climate becomes more dominant. This leads to a strong increase of ice mass.

The NAICE LGM reconstruction features less ice volume in the LIS, compared to the IceXG models. The advantage of using NAICE is that it includes some (minimal) ice sheet physics in its construction, and therefore works as a more realistic starting point for an ice sheet model than the ICE-xG models. Furthermore, we did not take into account the entire LIS, as it was stated in the text. By LIS we denote only the "continental" LIS (which does to take the Canadian Arctic Archipelago into account). See also the reply above.
* * *
> Fig. 4:
>
> Where is the spin-up phase? When does the forced warming in the glacial index method start? Are lakes already included during the spin-up? And how does your "simulation time" correspond to real world time?

As already stated in a previous reply and in the manuscript, the spin-up phase only equilibrates the thermodynamic fields of the ice sheet model. The model is run in a fixed geometry state, with boundary conditions fixed at LGM values.

I am not sure if I understand your question about the simulation time. Simulation time starts at year 0, which corresponds to the LGM state (~21ka BP). Simulation year 21000 would correspond to a present day state (i.e. the climate conditions according to the glacial index matches the pre-industrial state). However, if you compare the ice sheet states to a geologically constrained glacial history, the timing does not match. The idealized experiments we present are meant to simulate a deglaciation, and it is not meant to represent the specific last deglaciation.

We mention this in the Results section, *l. 336*:

**"In the following, all expressions of time are given in model years, relative to the start of the experiments, [...]"**
* * *
> L314: "At around 9 kyr the water level of this lake rapidly dropped, as a lower outlet to the Atlantic became ice-free."
>
> Which outlet? I'd like to see a map showing the locations of the possible spillover (Mississipi, St. Lawrence River, MacKenzie River) and drainage (Hudson Strait, Lancaster Sound, North-West Passage) routes in relation to your ice-sheet geometry.

For the major lakes shown in Fig. 8 we have determined these points and corresponding spillways and marked them on the map.

**Figure 8. [...] Modeled spill point and drainage routes for Lake Agassiz are shown in green in panels a-d.**
* * *
> L321: "At around 17.9 kyr, the ice saddle over Hudson Strait breaks apart and allows the lake to drain into the Labrador Sea"
>
> The deepest part of Hudson Strait is quite narrow, so a 20km DEM might significantly underestimate the water depth, and therefore the retreat rate. How do you think this affects your results?

The retreat of the Hudson Strait does not happen until near the end of the simulation in our idealized experiment. In reality, the Hudson Strait deglaciation happens much earlier, and the final ice dam is located in the southern part of Hudson Bay (Gautier et al., 2020). It might be possible that the depth is underestimated and preventing a more realistic retreat of the Hudson Strait. An earlier retreat of the Hudson Strait would cause an earlier drainage of the lakes south of the ice sheet. This might allow for the preservation of some of the ice that is located in the Labrador sector of the ice sheet compared to our simulation, but Hudson Bay would likely still become completely ice free due to ocean interaction.
* * *
> L322: "Due to GIA processes, the ice-free Hudson bay basin eventually rises above sea level"
>
> Is this realistic? I've never seen this happen in my own model runs (which include an actual geoid model), for me sea-level rise always outpaces isostatic rebound, but I don't know what the field data indicates.

No, this is not realistic. We believe that it is related to the initialization of the Lingle Clark GIA model of PISM. As already discussed before, we were not able to initialize the experiment from an interglacial state and run it into the glacial. Starting from a glaciarized continent, the model overestimates the relief topography because the initial LGM state was calculated using a different GIA model.

We added a short paragraph to the discussion section (*l. 413 ff*, see above).
* * *
> L326: "At around 21 kyr, the saddle collapses, which drains most of the lake."
>
> Again, how does 21 kyr simulation time correspond to real-world time? Does this match the 8.2 kyr event?

The timing would match with PD (21kyr after LGM). However, as we also write, the timing of the glacial retreat does not match with geologically inferred retreat pattern. Our idealized model setup is too simple to create a realistic retreat. Geologically constrained ice margin reconstructions (e.g. NAICE - Gowan et al., 2016) indicate separation of the Cordilarian and Laurentide ice sheets by ~14ka BP (which would be around 7kyr in model time).
* * *
> Fig. 5:
>
> why does the ice margin in the LCC simulation make a~500 km "jump" at~ 18 kyr?

The jump that can be seen in Fig. 5 happens between 17.4 and 17.5 kyr and is because the ice shelf section shown in Fig. 5 is close to the western margin of the ice shelf. For this time the displayed cross-section (northwards, along 90°W) is disadvantageous, because the predominant retreat direction of the ice margin here is eastwards. This can be seen in Fig. 8d.

Even though this "jump" does not directly display the rapid disintegration of the large ice shelf, the breakdown happens within ~500 yrs (compare with Fig. 8d-e). At 17.5 kyr the drainage through Hudson Strait is only blocked by last rest of grounded ice. As the ice becomes thinner, the high water level lifts this ice barrier and drains into the Atlantic - the lake level drops, until the ice barrier is grounded again. The rapid drop in water level caused the grounding line advances again a bit southwards (see Fig. 5) and the mass flux into the shelf drastically drops, which stops balancing the mass loss.

We have added a sentence about this in the Results section, *l. 367 f*:

**"We want to note that the sudden jump of the ice margin (~ 500 km), seen at this time in Fig. 5, stems from the ice margin vanishing laterally out of the 90°W profile."**
* * *
> L336: "By linearly interpolating between the warm, humid PD and cold, dry LGM climate states, unrealistically high accumulation is produced"
>
> There is no climate feedback in your model, your entire climate is prescribed through the glacial index. With a glacial index of 1 at t=0, the prescribed climate should be exactly that of the GCM that produced it. What you mean is that your initial ice sheet is simply not in equilibrium with this steady-state climate. This is why paleo-ice-sheet models generally need some form of tuning in their SMB parameterisations, and also why it's usually better to start a simulation in an interglacial (e.g. the Eemian) and run forwards from there (since you would not expect the LGM ice sheet to be in equilibrium with the LGM climate in any case).

I would call the lapse rate corrections of the temperature and precipitation fields a feedback. The other criticisms were already addressed above.
* * *
> *Fig. 9:*
>
> What does the thick blue line in panel d) signify?

We have added the following description to the figure caption:

**"Figure 9. [...] The blue line in panel (d) marks the lake level elevation in the LAKE experiment, while the gray line marks the sea level elevation of the DEF experiment."**
* * *
> *L354: "For this reason, the lake reconstructions are not expected to match well with observations"*
>
> This is a bit of a chicken-and-the-egg question; are your lakes wrong because your ice margins are wrong, or are your ice margins wrong because your lakes are wrong? Since the entire point of your paper is to show that the presence of the lakes affects the ice sheet (and the therefore the ice margin), you cannot simply ignore the feedback here.

To discuss the impact of the lake model we compare the different experimental setups ("lakes vs. no lakes"). From the no lake scenarios, we do not see a better match with geological reconstructions. We do, however, see that adding the simple lake boundary condition does have a massive impact on the resulting ice sheet configuration (i.e. Hudson Bay becomes ice free when lakes are included).

This is an idealized experiment, and obviously the margins would not be able to be simulated exactly the same as the geological observations. For that, we would need to implement the lakes in a coupled climate model (as an example), which is beyond the scope of this study. However, our idealized experiments do demonstrate that the presence of lakes will affect ice sheet dynamics and played a role in the deglaciation of the Laurentide Ice Sheet.
* * *
> *L355: "Drainage towards the Arctic, for example, is blocked until the ice saddle connecting the LIS and the CIS collapses"*
>
> This raises the question of, where would the real paleo-lakes have routed their spillover, and is that pathway indeed blocked by your modelled ice-sheet? If not, then this might be a resolution issue as I mentioned earlier.

Geological records show that the Cordilarean and Larentide ice sheets separated quite early, as noted already earlier. Since in our experiments the ice barrier collapses quite late in the simulation, drainage, which in reality was towards the Arctic, is thus not possible in our experiment. It is possible that this is related to the resolution, but more likely this an issue with the climate forcing, which has an obvious cold bias in the Arctic (as the ice sheet never retreats there). In our LakeCC standalone paper (Hinck et al., 2020) we discuss different lake stages and compare these to geologically inferred data.
* * *
> *L362: "This simple, two layered Earth model is not capable of handling the extreme deglaciation scenario of an entire continent"*
>
> This is unsatisfactory. In the introduction section, you (correctly) state that the GIA depressions are the reason those lakes exist in the first place. If your GIA model isn't performing well, then this should be fixed.

For further studies aiming to reproduce the deglaciation this might be necessary. Here we studied the impact of the presence of lakes on glacial retreat. A perfect model setup is not required to demonstrate this.
* * *
> *L373: "We assume that this is due to the enhanced sliding at the lake boundary"*
>
> This assumption can, and should, be easily verified, by turning off the "till saturation at next-to-lake ice pixels" parameterisation you described earlier.

We have run a sensitivity experiment (*nSG* - see Supplements) where we turned off the "slippery grounding line" parameterization. In this experiment no such ice lobes are observed. We rephrased the sentence in the revised manuscript (*l. 429 f*):

**"Due to the enhanced sliding at the lake boundary an advance is locally promoted."**
* * *
> *L375: "When water depth of the proglacial lake becomes too deep, the thin advancing ice front is presumably lost due to calving."*
>
> Presumably? Again, this seems like something that could (and should) be easily checked.

We have checked the accumulated mass loss within each of these lakes that border the ice sheet, and they all have a calving rate greater zero. This means that, even though no ice shelves are obvious, ice is lost due to calving as the ice sheet promotes into the lake. In the revised manuscript we have deleted the 'presubably'.

Using the thickness threshold calving model, the thin ice front of an advancing ice shelf is calved off when it is below the threshold thickness. Only if the ice thickness is is either above this value or above the flotation thickness at a given water depth (i.e. it is grounded), is the ice not removed. The lakes shown in Fig. 6 all have water depths below 40m, and ~20m at the grounding line. The advancing ice therefore needs to be greater than ~22m to be grounded. If the lake is deeper, the ice would need to be even thicker, which we do not observe in our experiments.
* * *
> L390: "Lakes, however, do impact the early retreat by inducing the formation of ice streams, which drain the ice sheet interior"

> This conclusion is not supported by your results. You should check what happens when this lake-enhanced sliding is turned off.

In the sensitivity experiment *nSG* (see supplementary document), which was also mentioned earlier, the "slippery grounding line" parameterization was turned off. This does indeed drastically change the results, as the mass flux into the lakes decreases. The sensitivity to the grounding line treatment is discussed in the revised manuscript (*l. 234 ff* and *l. 425 ff*, see above).
* * *
> L395: "Reconstruction of lakes could benefit from more realistic accounting of water fluxes"

> The assumption that the lakes are always filled to overflowing is probably one of the most justifiable ones you made, so I doubt that including a water transport model would significantly alter your results.

This might not be true though. For instance, there is a hypothesis that the Moorhead low stage of Lake Agassiz was caused when there was insufficient meltwater and water from other sources in the catchment to exceed evaporation (Lowell et al., 2013). Therefore the lake were not necessarily filled to the top.
* * *
> L397: "a more physically motivated calving model valid for grounded and floating ice termini or a lacustrine sub-shelf melting model, would improve the ice-dynamical response to lakes"

> Since you already show that the choice of calving law has a strong impact on your results, this one seems a lot more important.

We agree.
* * *
> L406: "The lake model promotes the formation of continental ice streams"

> Your figures only show ice lobes, no ice streams.

Our results show the increase of ice flow, which is triggered by interactions with proglacial lakes. The increase of ice flow, which we called ice streams, might not fulfill the characteristics of locally confined ice streams. We will therefore reformulate these text passages and speak of increased ice velocity/flow.

**From RC3**

> Lastly, a minor point: the phrase "dynamical topography" is typically used to describe tectonic movement, changes in elevation due to mantle convection, and other processes that act on the Myr timescale.

You are referring to line 22, where we write "[...] and topography was dynamic", don't you? We have rephrased this sentence in the revised manuscript, *l. 22*:

**"[...] and topography was not static."**

**References**

- Beckmann, A., Goosse, H., 2003. A parameterization of ice shelf–ocean interaction for climate models. Ocean Modelling 5, 157–170. https://doi.org/10.1016/S1463-5003(02)00019-7

-

Feldmann, J., Albrecht, T., Khroulev, C., Pattyn, F., Levermann, A., 2014. Resolution-dependent performance of grounding line motion in a shallow model compared with a full-Stokes model according to the MISMIP3d intercomparison. Journal of Glaciology 60, 353–360. https://doi.org/10.3189/2014JoG13J093

- Gauthier, M.S., Kelley, S.E. and Hodder, T.J., 2020. Lake Agassiz drainage bracketed Holocene Hudson Bay ice saddle collapse. Earth and Planetary Science Letters, 544, p.116372. https://doi.org/10.1016/j.epsl.2020.116372

- Golledge, N.R., Kowalewski, D.E., Naish, T.R., Levy, R.H., Fogwill, C.J., Gasson, E.G.W., 2015. The multi-millennial Antarctic commitment to future sea-level rise. Nature 526, 421–425. https://doi.org/10.1038/nature15706

- Gowan, E.J., Tregoning, P., Purcell, A., Montillet, J.-P., McClusky, S., 2016. A model of the western Laurentide Ice Sheet, using observations of glacial isostatic adjustment. Quaternary Science Reviews 139, 1–16. https://doi.org/10.1016/j.quascirev.2016.03.003

- Gowan, E.J., Niu, L., Knorr, G., Lohmann, G., 2019. Geology datasets in North America, Greenland and surrounding areas for use with ice sheet models. Earth System Science Data 11, 375–391. https://doi.org/10.5194/essd-11-375-2019

- Hinck, S., Gowan, E.J., Lohmann, G., 2020. LakeCC: a tool for efficiently identifying lake basins with application to palaeogeographic reconstructions of North America. Journal of Quaternary Science 35, 422–432. https://doi.org/10.1002/jqs.3182

- Lemmen, D.S., Duk-Rodkin, A. and Bednarski, J.M., 1994. Late glacial drainage systems along the northwestern margin of the Laurentide Ice Sheet. Quaternary Science Reviews, 13(9-10), pp.805-828. https://doi.org/10.1016/0277-3791(94)90003-5

- Lowell, T.V., Applegate, P.J., Fisher, T.G. and Lepper, K., 2013. What caused the low-water phase of glacial Lake Agassiz?. Quaternary Research, 80(3), pp.370-382. https://doi.org/10.1016/j.yqres.2013.06.002

- Niu, L., Lohmann, G., Hinck, S., Gowan, E.J., Krebs-Kanzow, U., 2019. The sensitivity of Northern Hemisphere ice sheets to atmospheric forcing during the last glacial cycle using PMIP3 models. Journal of Glaciology 1–17. https://doi.org/10.1017/jog.2019.42

- Quiquet, A., Dumas, C., Paillard, D., Ramstein, G., Ritz, C., Roche, D.M., 2021. Deglacial Ice Sheet Instabilities Induced by Proglacial Lakes. Geophysical Research Letters 48, e2020GL092141. https://doi.org/10.1029/2020GL092141

- Teller, J.T. and Leverington, D.W., 2004. Glacial Lake Agassiz: A 5000 yr history of change and its relationship to the δ18O record of Greenland. Geological Society of America Bulletin, 116(5-6), pp.729-742. https://doi.org/10.1130/B25316.1

- Veillette, J.J., 1994. Evolution and paleohydrology of glacial lakes Barlow and Ojibway. Quaternary Science Reviews, 13(9-10), pp.945-971. https://doi.org/10.1016/0277-3791(94)90010-8

**Author response to referee comment RC4 to manuscript No. tc-2020-353, "PISM-LakeCC: Implementing an adaptive proglacial lake boundary into an ice sheet model"**

**submitted to The Cryosphere by Sebastian Hinck et al.**

> This study is a great concept and shows how ice-marginal lakes can potentially change the modelled retreat of large continental ice sheets. This is mostly relevant to the glacial NH paleo-ice sheets (less so Greenland or Antarctica, or small valley glaciers), but nevertheless the study will still be interest to the many people working on N. Atlantic deglacial climates, because of the great significance of meltwater inputs to the ocean from the Laurentide ice sheet. The authors also demonstrate a process driven by ice-marginal lakes that could lead to rapid retreat of what is raditionally considered a land-terminating ice sheet margin.

> There are many uncertainties, perhaps obviously – and it's good that the authors focus mainly on the different behaviours with/without the lakes, and skip a detailed comparison with geological evidence at this stage in development.

> I have listed some specific comments/questions below but my two main concerns for implementation of this new LakeCC scheme are as follows.

> 1. Due to numerical instabilities, the model apparently cannot cope with rapid lake drainage/filling. This is of concern because the rapid drainage of lakes into the North Atlantic is one of the main applications of this work, yet lake level changes in a 'workaround' solution appear to be limited to 1 m/yr (the gamma parameter in Eq. 1 and Table B2). This represents a maximum of only 25 cm per 3-month model time step. The cause of the instability is not discussed but if it cannot be solved without the current 'work-around' then I struggle to see how this new scheme can be implemented in a realistic scenario.

> 2. Use of the marine ice shelf parameterisations is really questionable in a lacustrine setting. Fast sub-shelf melting in marine settings is enabled partly by the density contrast between dense saline ocean water and light fresh melt water. This contrast drives rapid overturning in the shelf cavity, and large heat fluxes. In a lacustrine setting, the density contrast between fresh lake water and fresh melt water is much lower, so presumably the sub-shelf melt rate will also be much lower, for a given temperature forcing. In which case, perhaps a lot of the simulated shelves would really just remain as grounded ice. The authors do of course acknowledge this shortcoming, but don't then address it. Following on from this point, what is the water temperature in the lake? I can't see how this is estimated but it is crucial for calculating subshelf melting.

> Since there is so little work on this subject it would be great to see this study published. In this respect, I believe that the instability issue either needs fixing or needs much more discussion, so we are convinced it isn't a symptom of some other underlying problem in the model, and so that we know how the problem of unrealistically slow lake draining/filling can be overcome. Second, because this is largely a model development paper, we need some basic indication of importance of a few critical parameters (in this application, perhaps the till friction angle, ice shelf basal melt, grid resolution). Even some short model runs could help answer that? Finally, although not essential to the concept of the study, if you carry out some more simulations I would also revisit the climate forcing and need for the 3500m elevation limit.

We would like to thank the anonymous referee for reviewing our paper. The original comments are indented, while our responses aligned to the left of the page.

For this review round several sensitivity experiments were run, which are referred to in the revised manuscript and the supplementary materials.

We will elaborate a bit more on the instability in the revised manuscript and mention the sensitivity test for higher fill rates. These issues are related to the numeric solver (see responses to reviewer #1 on this matter). When doing additional tests, the stability issues we encountered before were not happening. Furthermore, we will clarify that the model output can not be used for direct freshwater forcing of an ocean model.

In the following, we will respond to all comments and questions.

**Specific comments/questions:**

*Abstract (L7), Conclusions, & elsewhere:*
You don't specifically show ice streams along the continental margin, even in Fig 7. There are regions of faster flow in the LCC run but these don't look particularly stream-like. Because you are using a spatially uniform basal parameters, and a coarse grid, this aspect of ice dynamics isn't well captured by your model. Rather than saying you develop ice streams, I would recommend to simply point out that ice velocity is higher at locations X in the LCC run.

Yes, this was also criticized by the other reviewers. We rephrased the relevant sentences and rather speak of increased ice flow/velocity.
* * *
*L22: "Reorganization of the lakes' drainage networks and sudden drainage events due to the opening of lower spillways may have impacted the global climate by perturbing the thermohaline circulation system of the oceans (Broecker et al., 1989; Teller et al., 2002)."*

It's important here to note that drainage could have been under the ice, as well as over it. Lack of subglacial routing option means lakes can only overspill, yet subglacial drainage could lead to outburst events long before the supraglacial spillway is reached. Omitting this subglacial option perhaps helps the experimental design but it deserves at least a mention.

Subglacial drainage is indeed very likely to have happened, for instance during the final drainage of Lake Agassiz during the 8.2 ka event (Gautier et al., 2020). For several reasons, discussed in the manuscript, our model does not conserve water and thus can not be used to calculate water fluxes or drainage events. Developing a model to simulate subglacial drainage (and subglacial water storage) would be very complicated and likely would require a very sophisticated model that is beyond the scope of our current study.
* * *
*L34:* Not an important point but the NH cooling was maybe caused by freshwater fluxes (not definitely, as implied here).

Yes, that is true. We rephrased the paragraph in the revised manuscript, **l. 32 ff**:

**"In periods with large amounts of meltwater that caused the formation of proglacial lakes the lake-ice interactions might have been a key factor to explain rapid glacial retreat. As an example, one hypothesis for the cause of the 8.2ka event, a period characterized by a sudden drop in Northern Hemispheric mean temperatures, is the rapid demise of the central LIS (Carlson et al., 2008; Gregoire et al., 2012; Matero et al., 2017; Lochte et al., 2019) leading to a large freshwater input into the Labrador Sea."**
* * *
*L41 "Apart from their relevance for understanding processes that lead to the demise of the late Pleistocene and early Holocene ice sheets, interest in contemporary proglacial lakes and their role in glacial retreat is growing (Carrivick and Tweed, 2013). Motivations for these studies range from predicting and managing water resources under a warming climate and recognizing possible risks due to glacial outburst floods (Carrivick et al., 2020)."*

I'm very sceptical of this work being usefully down-scaled to the small present-day proglacial lakes, because the parameterisation of ice-lake interactions would be so specific to a given site. Maybe best to stick to the paleo aspect.

That is correct. We have removed this point from the introduction.
* * *
*L79:* Resolution (20 km), is this really sufficient to capture the processes you are aiming at modelling? I understand the limit on computational effort but wonder if the positive feedbacks between lake development and ice stream development could be underestimated. For example a proglacial lake extending for 50 km along the ice sheet margin would only meet the simulated ice sheet at two or three grid points, is that enough to initiate an ice stream in the model? I am not suggesting lots of runs with different configurations, but it would be good to see some sort of sensitivity. For example, run the model at 20km to a couple of interesting points and use these as initial states for some short simulations with higher (and lower) resolution.

Our results show that lakes that are only a few cells wide impact the ice dynamics (see for example Fig. 4). Resolution is, however, an issue in ice sheet models in general, especially when it comes to determining the grounding line position. In PISM it is determined using a sub-grid interpolation scheme. Using this scheme, even at relatively coarse resolution, grounding line position is reasonably well represented (Feldmann et al., 2014).

One issue though, related to resolution is the "slippery grounding line" parameterization, which reduces the basal friction in the cells next to the grounding line. At a lower resolution the lubricated strip underneath the ice sheet would be more narrow, which would reduce the grounding line flux.

We further want to highlight that, since the few-cell-wide lakes are potentially the result of under-resolved topography, the focus of this model are the major lakes, which are well above the size of the grid resolution.

> *L88:* Basal boundary condition. The ice sheet sits on till, and slides when driving stress is greater than the yield stress of till. Is the substrate taken as being spatially uniform? Is there some sensitivity to this? Actually the design of this experiment is such that specifying a uniform yield stress is maybe of benefit (it is purely the addition of the LakeCC that is being studied), but perhaps the influence of LakeCC is very much dependent on the basal slipperiness, as this will dictate how far inland the "ice streams" can propagate. In which case some simple sensitivity expt as above would be very useful.

Yes, the basal properties are assumed to be spatially uniform in our experiments, which is not realistic but helps isolate the effect of the lake boundary.

We are in the process of implementing a more elaborate basal conditions model in PISM that takes into account changes in sediment distribution and grain size (e.g. Gowan et al., 2019). Once this new basal conditions model is complete, we will attempt this test.

> *L99 "To prevent the ice sheet from expanding into regions and high elevations where a more advanced approach would limit precipitation, the second model sets precipitation to zero above a threshold height or accordingly to a given mask (see Appendix D2)"*
>
> This seems like a dubious approach to me. If the ice sheet can grow sufficiently thick that precip cannot be parameterised, is there not something wrong with the ice sheet model configuration or climate forcing? What evidence do we have the the ice sheet elevation was less than 3500 m? This is a very low upper limit for an ice sheet of that size. On L363 you mention "the climate forcing tends to accumulate too much ice...".
> There are so many unknowns in the ice sheet model, could you not also say "the ice sheet tends to dissipate too little ice"? For example, why not just use a slightly more slippery bed (since this is so poorly constrained anyway) to avoid this problem of too much ice?

Ice sheet reconstructions of the Laurentide Ice Sheet (e.g. NAICE and ICE-6G) do not have much area that exceeds 3500 m, so we regard this as a crude way to ensure the ice sheet is not growing to an unrealisticly large size. We agreee that this is likely a problem related to how the climate forcing is parameterized, but this approach was necessary to ensure that the deformation of the Earth did not become so large that the area covered by the ice sheet doesn't become a large sea during deglaciation (and preventing us from testing the lake model). Using a different slippery bed condition might also allow us to avoid this issue, but that is not the approach we took in our idealized experiments.

We added the following paragraph to the Methods section, *l. 303 ff*:

**"To prevent ice sheet growths in eastern Siberia and above 3500 m elevation (relative to PD sea level), the precipitation to zero in these regions. The maximum elevation threshold was chosen in an ad-hoc fashion. However, current LGM reconstructions (e.g. NAICE (Gowan et al., 2016), and ICE-6G (Peltier et al., 2015)) exhibit higher surface elevations only in few mountainous regions."**

**Sect 2.2 LakeCC**

> *L116:* Rapid filling causes the model to crash. As a modeller this is very worrying given that the maximum time step for the ice sheet model is only 0.25 yr (3 months). What is the reason for the numerical instability – e.g., is it in the ice sheet model or LakeCC? The implemented lake filling algorithm effectively dampens lake level changes, what is the time scale for this? Is the max filling/draining rate (gamma) really just 1 m/yr?
> This limit on drainage rate could place quite some restriction on the usefulness of the model in a real deglacial lake drainage setting. For example, if a lake drainage is initiated by overspilling the ice sheet (or by growing a subglacial channel), then in practice the very strong positive feedback could lead to rapid drainage by incision of a supraglacial channel or growth of a subglacial conduit. But how can the model cope with this? If the lake level is reduced artificially slowly, does that mean a lot more water drains out of the lake than was actually in it? And would a drainage channel/conduit become vastly over enlarged because the lake level (and thus the hydrostatic head driving the channel/conduit development) is held artificially high?

As elaborated in the response to reviewer #1, the stability issue is related to how PISM numerically solves the ice sheet flow. In our original tests, if the lake filled or drained too quickly, it would cause the model to crash. We have further run some sensitivity tests with higher rates (*FR5*, *FR10*, *FR50*, with 5, 10 and 50m/year, respectively). For details on these, see the Supplements. Surprisingly, all runs finished without any problem and the results do not show any big difference to the *LAKE* run. Experiment *FR50* already comes pretty close to an immediate response of the lake level to a change in the target level. We have to state that the appearance of the instability is strongly dependent on the configuration of the ice sheet when the model is evaluated. That is to say, for these new experiments, we might simply have been lucky that no critical situation was triggered and that slight changes in the ice sheet configuration might crash the numerical solver even at a lower fill rate. If still a higher fill rate is desired but the model becomes instable, reduction of the time step would likely help.

Concerning the instability and gradual filling, we added a new sub-section to our Limitations section, **l. 269 ff**:

"**2.4.4 Gradual filling & numerical instability**

**When running the model, we observed that PISM occasional crashes when there is rapidly rising or dropping the water levels between time steps. Unfortunately, we can not tell the exact reason why the numerical solver failed, but we found that these crashes only appear shortly after the new lake or ocean basins, which are in contact with the ice, appear or vanish. From a numerical point of view, however, this is not surprising, as the numerical representation of a physical system requires the underlying equations to be smooth functions. These crashes, due to local unsteadiness in the fields, are known by the developers of PISM to appear (https://pism-docs.org/wiki/doku.php?id=kspdiverged, accessed: 2021/08/09).**
**With all the simplifications described above, the LakeCC model is not designed to exactly reconstruct the water level of proglacial lakes, nor to estimate freshwater fluxes. It rather gives a crude approximation of where potential lake basins are located and what their maximum water level is. Therefore, we assume the gradual filling mechanism using a (rather arbitrarily chosen) constant fill rate does not result in lake levels that are less valid.**
**If an almost instantaneous filling or draining of lake basins is desired, this could be achieved by setting a high fill rate γ. This would most likely require a reduction of the duration of the time steps, though, which increases the computational cost. In sensitivity tests, which are found in the supplementary material, several higher values were chosen. For these tests, no model crashes were observed. This might either be because we restricted the time step to 0.25yr for all experiments, or simply because no critical situations were triggered. The results of these experiments show no fundamental difference to the LAKE experiment. Future implementations could target a more realistic treatment of basin filling.**"

Furthermore, we want to note that advanced processes, such as incision of drainage channels, are not included in the model. The model only adds an dynamical "wet" boundary to the margin of the ice sheet.
* * *
> L142 *"If a basin disappears because it merged with the ocean, the lake level is gradually changed until sea level is reached, and then removed."*

> I don't understand here how a lake can merge with the ocean unless its lake level already matches the sea level.

This kind of situation could happen if, for instance, the lake is formed purely from ice damming, but is otherwise the basin is below sea level. Once the ice margin retreats, the lake becomes part of the ocean. This happened with the final drainage of Lake Agassiz (Gautier et al., 2020), for instance. It transitioned into the Tyrrell Sea (the proto-Hudson Bay) after the ice saddle collapsed.
* * *
> *L165:* The general PISM marine boundary parameterisations are used. Were these not developed for saline water? I think modifying these for freshwater would be a fairly fundamental step in developing this new LakeCC component. In particular with ref to Line 237 ("Due to the difference in density between fresh and ocean water, the layer of melt water underneath the ice shelf experiences a ~200 times higher buoyancy in sea water...") this does seem like it should be addressed here rather than a future study. What if there is a 200 times less sub-shelf melt beneath a lacustrine ice shelf? Will the development of floating ice shelves then be much less likely? What is the ambient water temperature in the lake – in a marine setting this would be crucial information.

We revisited the sub-shelf melting parameterization used in our experiments and tried to estimate values more suitable for the lacustrine setting. For details see sensitivity experiment *MR* in the Supplements.

Even though we estimated a ~40x lower melt rate in a freshwater lake (water temperature of 2°C at a depth of 300m) the results did not significantly change compared to the *LAKE* experiment. We rather found that the sub-shelf melting and calving are secondary, and that mass loss in our experiment is dominated by surface melt due to the warming climate and lowered ice margin.

We added the following paragraph to the Discussion, **l. 440 ff**:

"**Sensitivity experiments (see supplementary material) show that this rapid disintegration of the LIS is mainly driven by the strongly negative mass balance along the southern ice margin. Calving and sub-shelf melt rates play only a secondary role here. These findings are in agreement with Quiquet et al. (2021), who refer to this instability as PLISI.**"

Concerning the lake temperatures:
The main problem is, that it is not possible to find values valid for all proglacial lakes. Also seasonal variability might change the properties of a lake greatly.
* * *
> L176 *"Ice temperature at the base is set to the pressure-melting point, which is a function of pressure, hence ice thickness."*

> Presumably the PMP is also a function of salinity in your model?

The pressure at the ice base is only a function of the ice thickness (given constant ice density). Therefore, the pressure does not depend on the salinity of the ambient water. Salinity would only change the submerged height of the ice shelf.

The temperature at the ice base is different to the temperature used for calculating the melt rate. In this model the temperature difference for calculating the energy flux, is calculated between the mean ambient water temperature and the freezing point of water (which depends on pressure and salinity). For details see the model description in Beckmann & Goosse (2003).
* * *
> L266 "When using a glacial index derived from the NGRIP $\delta18O$ measurements (North Greenland Ice Core Project Members, 2007), the modeled deglaciation is too rapid to identify contributions from the lake model. Instead, the deglacial climate signal was crudely approximated by a linear model (see Fig. A1a)."

I agree that a linear transition is a better way of understanding how the new LakeCC model works, than using the noisy NGRIP d18O. Indeed the NGRIP record may well contain signals of the very lake drainages your model is trying to capture. But I worry from this statement (& comments above) that the model can't cope with rapid changes in climate forcing, and this would be a significant limitation given its intended application to deglacial climates.

The sensitivity experiments discussed above (*FR..*) suggest that even higher fill rates are possible without causing problems. If this value should be much higher, this would have to come at the cost of smaller time steps, but it appears to be more stable than we initially thought.
* * *
> L284 "In the LNS experiment $\Delta hL$ was greatly increased to remove almost any floating ice, as large ice shelves like those seen in the LCC experiment are unlikely to have existed"

I get the point of this sensitivity run but is there actually paleo evidence to convincingly disprove the existence of extensive ice floating ice shelves?

We are not aware of any reference investigating the potential size of ice shelves on large proglacial lakes. Furthermore, we are not aware of geological evidence that would support the existence of such vast ice shelves. In general, geomorphologically constrained reconstructions of glacial lakes do not depict ice shelves that extend deeply into the ice sheet, as is simulated in our experiments (e.g. Veilette, 1994; Teller and Leverington, 2004; Lemmen et al., 1994).

We added the following paragraph to the Discussion, *l. 443 ff*:

**"We want to note that the vast ice shelves that can be observed at this late phase of the collapse of the LIS (Fig. 5 & 8c-d) might be unrealistic. At least we are not aware of any geological evidence or reconstruction, indicating such ice shelves existed. It is possible that the simulated grounding line flux is too high or the calving rate too low. Two sensitivity runs checking these parameters (IncCalv an nSG in the supplementary material) do not exhibit such large lacustrine shelves. However, more research is necessary to decide how to best parameterize these issues."**
* * *
> L315, L358 Growth of the very large proglacial lakes (current Great Lakes region & then Agassiz). What are the lake levels and would they have been likely to drain subglacially, given their proximity to the ice margin and also considering that sea level was lower at that time? This subglacial drainage route would be one way of avoiding unreasonably large lakes – although I understand that adding that process isn't the point of the present study. Nevertheless, is surely worth a mention.

The lake levels of the major lakes shown in Fig. 8 are: a) 160m, b) 205m, c)435m, d) 421.85m (falling towards target level: 410m), e) 410m. Note the water level is relative to the PD geoid. We have checked the spillpoint and drainage route for every lake in Lake Agassiz basin and marked these in Fig. 8:

**Figure 8. [...] Modeled spill point and drainage routes for Lake Agassiz are shown in green in panels a-d.**

In panel d only a shallow plug of ice preventing the lake from draining into the Atlantic. However, as discussed above, the model does not allow for subglacial drainage, and thus a lake is dammed by ice as long as the ice stays grounded.

The following figure is a plot of this time slice (17.5kyr). The panels left and right show transects along the horizontal and vertical red lines.

*Map depicting the glacial configuration of the LAKE experiment at 17.5kyr over Hudson Bay. The panels left and right show cross-sections along the red lines.*

The sea level elevation does only have minor impact on the drainage route.
* * *
> *L345 "Without adding more advanced climatic feedbacks a realistic deglacial reconstruction is not expected. This, however, has not been the focus of this study, which is to test the PISM-LakeCC model and studying its impact on the ice dynamics and the glacial retreat. For these purposes the experimental setup is sufficient. Analyzing the interplay between ice sheets and proglacial lakes in more realistic setups, e.g. fully coupled to a climate model, and comparing against various geological proxy data is an interesting topic for future research"*

To me this is one of the real positives of this paper – it focuses just on how the addition of lakes modifies the modelled retreat. The simple linear forcing greatly helps interpretation of the results, and similarly there is no long discussion of why the model inevitably doesn't fit geological reconstructions. This is also why I think the paper would really benefit from a little more testing of sensitivity. At the moment the paper is a model development study and in this case the sensitivity aspect is very important.

Thank you. As mentioned earlier, we have performed a number of additional sensitivity experiments.
* * *
> *Fig 3:* Last sentence of the caption – what is the continental LIS & should it really extend that far into the Atlantic?

To split the contributions from the different ice sheets, we defined regions via a mask, in which ice is attributed to the respective ice sheets. With the name "continental" LIS, we want to emphasize, that parts of the LIS (namely the parts on the Canadian Arctic Archipelago) were not taken into account. The shaded area in Fig. 3 depicts the region chosen focused on in the study. The selected region extends so far into the Atlantic, to also capture ice shelves that (potentially) extent far into the ocean. This does, however, not happen. In the Atlantic, it is usually ice free and thus does not contribute to the LIS.

We have rephrased the explanation of the LIS in the Results section, *l. 332 f*: **"The highlighted area in Fig. 3 marks the region of interest for this study. In the following, when referring to the LIS, we will consider only the parts of the ice sheet that are within this region."**
* * *
> *Fig A1:* would be great to have this in the main text.

Since the linear transition of the glacial index and mean sea level is also described in the text, we think that the figure is not necessarily required for understanding. We therefore decided against moving the figure into the main text.
* * *
> *Spelling/grammar:* Needs a proof read to iron out several minor errors.

We did more proof reading to fix grammar and spelling mistakes.

**References**

- Beckmann, A., Goosse, H., 2003. A parameterization of ice shelf–ocean interaction for climate models. Ocean Modelling 5, 157–170. https://doi.org/10.1016/S1463-5003(02)00019-7

- Feldmann, J., Albrecht, T., Khroulev, C., Pattyn, F., Levermann, A., 2014. Resolution-dependent performance of grounding line motion in a shallow model compared with a full-Stokes model according to the MISMIP3d intercomparison. Journal of Glaciology 60, 353–360. https://doi.org/10.3189/2014JoG13J093

- Gauthier, M.S., Kelley, S.E. and Hodder, T.J., 2020. Lake Agassiz drainage bracketed Holocene Hudson Bay ice saddle collapse. Earth and Planetary Science Letters, 544, p.116372. https://doi.org/10.1016/j.epsl.2020.116372

- Gowan, E.J., Niu, L., Knorr, G., Lohmann, G., 2019. Geology datasets in North America, Greenland and surrounding areas for use with ice sheet models. Earth System Science Data 11, 375–391. https://doi.org/10.5194/essd-11-375-2019

- Lemmen, D.S., Duk-Rodkin, A. and Bednarski, J.M., 1994. Late glacial drainage systems along the northwestern margin of the Laurentide Ice Sheet. Quaternary Science Reviews, 13(9-10), pp.805-828. https://doi.org/10.1016/0277-3791(94)90003-5

- Teller, J.T. and Leverington, D.W., 2004. Glacial Lake Agassiz: A 5000 yr history of change and its relationship to the δ18O record of Greenland. Geological Society of America Bulletin, 116(5-6), pp.729-742. https://doi.org/10.1130/B25316.1

- Veillette, J.J., 1994. Evolution and paleohydrology of glacial lakes Barlow and Ojibway. Quaternary Science Reviews, 13(9-10), pp.945-971. https://doi.org/10.1016/0277-3791(94)90010-8

---

## Referee Report (RR1)

Review of Hinkc et al. 2021, revised version

The authors present a set of simulations of the deglacial retreat of the Laurentide Ice Sheet (LIS), where they investigate the effect of proglacial lakes on this retreat. By considering the effect of proglacial lakes on grounding-line dynamics, sub-shelf melt, and calving, they show that the presence of these lakes significantly accelerates the LIS retreat. When no lakes are present in their model at all, a sizeable ice-sheet remains when their model reaches the present day, indicating that it is important to consider proglacial lakes when studying the dynamics of glacial cycles. Determining the processes behind the extremely rapid retreat of the LIS during several phases of the last deglaciation, such as the meltwater pulses, is becoming more and more important as the implications of ice-dynamical instabilities for projections of future sea-level rise are becoming more apparent. I therefore believe that studies such as this one could be very interesting, as they demonstrate that explanations for such rapid retreat can potentially be found without invoking some strong atmospheric forcing.

However, I have a few concerns about both the methodology and the framing of the results, which I believe should be addressed before the manuscript can be published.

Firstly, regarding the nature of the "proglacial lake ice-sheet instability" (PLISI). While the authors compare this to the better-known phenomenon of marine ice-sheet instability (MISI), they ascribe this instability to the elevation-temperature feedback (e.g. the first paragraph of section 5 Conclusions). However, MISI has nothing to do with mass balance processes, but is a purely ice-dynamical process, which is why all the different MISMIP experiments assume a uniform, unchanging, elevation-independent surface mass balance. In their rebuttal to my previous review, the authors show a timeseries of the different components of the total mass balance, showing that the retreat of the LIS is dominated by runoff and oceanic (not lacustrine) calving. This seems to be at odds with the findings of the different studies investigating MISI, and it also raises the question of how the presence of the lake can lead to such a strong lowering of the land-based ice dome, if the lake itself hardly removes any mass. Indeed, the authors report (in their rebuttal and in the supplementary material) that neither changing the sub-shelf melt rate nor the calving threshold thickness over the lakes significantly affects the results. If neither of these processes is significant, then what causes the difference in retreat rate between the lakes and the non-lakes simulations? The ice-dynamical processes governing MISI affect mass transport from the sheet to the shelf, but the shelf mass still has to go somewhere. If it is not removed by either sub-shelf melt or calving, then the shelf will grow thicker over time, the grounding-line will advance, and the basin will fill with grounded ice. The modelled grounding-line retreat must be caused by mass loss either on the shelf or on the sheet; the former does not happen, so the authors claim, but the latter should not be so different between the lake and no-lake experiment. This issue should be investigated further.

Regarding grounding-line retreat: in my previous review, I referred to the work of Natalya Gomez, who showed that gravitational effects can significantly reduce grounding-line retreat, and can even lead to stable configurations on (mildly) retrograde slopes even in the absence of buttressing. This was followed by a response from one of the authors, who claimed that the Lingle&Clark GIA model used by PISM is "self-gravitating". After consulting

with a colleague who specialises in GIA, I found that this is only partially true; the Green's functions in the Lingle&Clark model include a self-gravitating term that is appropriate for a solid Earth that is in equilibrium with the surface load. The added ice mass on the surface is then balanced by the displaced mantle mass, so that the resulting gravitational perturbation is very small (deviating from zero because of the tensile strength of the Earth's crust, so that the locally displaced mantle mass does not necessarily equal the local ice load). However, this assumption is not appropriate for a retreating ice sheet; as phrased in the original article by Lingle and Clark: "Additional changes in depth caused by perturbation of the gravitational potential field are not included." The very existence of the vast proglacial lakes studied here is owed to the delayed rebound of the Earth's surface. At such moments, the gravitational signal can be significant, and the effect on water depth at the grounding line, and therefore on grounding-line retreat, should not be neglected. Based on the different studies by Natalya Gomez, I expect that this could significantly reduce the accelerated retreat reported by the authors. While I acknowledge that it might be too much work to include an appropriate GIA model in PISM for this study, the drawbacks of not doing so should be discussed in the manuscript.

Then, regarding the experimental set-up. The authors explain that their model is initialised with ice thickness and bed topography from the NAICE model, and thermodynamics are spun up to achieve a stable englacial temperature. However, when the simulation starts, ice volume rapidly increases to ~50% more than the initial value in all experiments (both with and without lakes), which suggests that the surface mass balance parameterisation is not properly tuned. The resulting over-sized ice-sheet (exactly how over-sized is difficult to quantify, as the authors rather confusingly chose to exclude ice in the Cordillera and the Canadian Arctic from the volume calculation) causes an unrealistically deep GIA depression, which leads to modelled lakes that are probably significantly larger than they would have been in reality. This likely means that the accelerated retreat reported by the authors is overestimated.

The authors mention a "problem" with the initialisation of the Lingle&Clark GIA model, which causes the Hudson Bay to become subaerial when the simulation reaches the present day. They ascribe this to the difficulty of differentiating in their code between the initial state and the equilibrium reference state. They also state that circumventing this problem by starting the simulation during the previous interglacial was not feasible, as this "suffered from the fact that the bed deformation along the southern ice margin was so deep, that the basin was connected to the Atlantic Ocean, which consequently inhibited the formation of lakes". I find this unsatisfying; as with the "numerical instabilities" they report elsewhere (which they circumvent by creating a rather convoluted scheme of different lake water levels, masks, and filling rates), these kinds of coding problems should really be solved before using a model for research applications.

Regarding the surface mass balance: in my previous review I referred to a few studies that showed how the presence of large proglacial lakes could positively affect the surface mass balance over the adjacent ice sheet, thereby potentially reducing retreat rates. The authors responded to this very briefly in their rebuttal, stating that including such SMB effects was beyond the scope of their study. However, they also claim that the accelerated retreat observed in their simulations is caused by surface mass balance processes (via the elevationtemperature feedback), which are triggered by the presence of the lake. I'd like to see some more discussion about why they think the latter process is so much stronger than the former.

Lastly, regarding the framing of the results: as I mentioned at the start, studies such as this one are important not only from a purely palaeoclimatological / palaeoglaciological perspective, but also for the way we think about near-future retreat of the Greenland and Antarctic ice sheets. The idea that ice-dynamical processes such as MISI, and more recently the ice-cliff instability caused by brittle fracture, can be as or even more important than atmospheric processes has only relatively recently become commonly accepted; the uncertainty in sea-level projections beyond 2100 is dominated by ice-dynamical terms, and a lot of effort is being dedicated to improving our understanding of these processes and reducing those uncertainties. Understanding the interplay between atmospheric and ice-dynamical processes in the geological past is an important part of this effort. I feel that the authors here could improve the readability of their manuscript by more clearly framing their study in this context; they could choose to present it as (A) a schematic study that investigates a particular process (e.g. PLISI), (B) a reconstruction of ice-sheet / lake / GIA evolution during the last deglaciation, or (C) a system-based study that looks at the role of lakes in the Earth system. Right now, I feel the manuscript does not really fall in any of these three categories, which makes it difficult to decide which drawbacks are acceptable and which are not. If the only aim is to quantify the ice-dynamical processes, then the lack of atmospheric processes is not problematic. If the authors want to go for a realistic reconstruction, then the choice of climate forcing is probably the largest source of errors. If they want to take a comprehensive approach to the Earth system, then the forcing, timing, and geometry are probably of lesser concern than the lack of atmospheric / geoid / GIA feedbacks. I suggest that the authors make a conscious choice about which direction they want to move in with this study, and frame the drawbacks and uncertainties of their findings accordingly.

---

## Author Response (AR2)

**Major revision of manuscript tc-2020-353 -- PISM-LakeCC: Implementing an adaptive proglacial lake boundary into an ice sheet model**

**by Sebastian Hinck, Evan J. Gowan, Xu Zhang, and Gerrit Lohmann**

Dear Kerim Nisancioglu,

we are glad to submit our revised manuscript, based on the thoughtful comments and criticisms by the two remaining reviewers.

With this resubmission we provide

- the revised manuscript,
- a document highlighting all changes made on the manuscript, and
- our point-to-point responses to each reviewer (this document),

as required by the journal.

In the following, we will present our responses to all comments of the reviewers and provide, where applicable, the respective changes made in the revised manuscript. We hope our revised manuscript will further be considered for publication in *The Cryosphere*.

Yours sincerely,

Sebastian Hinck on behalf of all co-authors

**Author response to Report#2 by Tijn Berends to the revised version of manuscript No. tc-2020-353, "PISM-LakeCC: Implementing an adaptive proglacial lake boundary into an ice sheet model"**

**submitted to The Cryosphere by Sebastian Hinck et al.**

> Review of Hinkc et al. 2021, revised version
>
> The authors present a set of simulations of the deglacial retreat of the Laurentide Ice Sheet (LIS), where they investigate the effect of proglacial lakes on this retreat. By considering the effect of proglacial lakes on grounding-line dynamics, sub-shelf melt, and calving, they show that the presence of these lakes significantly accelerates the LIS retreat. When no lakes are present in their model at all, a sizeable ice-sheet remains when their model reaches the present day, indicating that it is important to consider proglacial lakes when studying the dynamics of glacial cycles. Determining the processes behind the extremely rapid retreat of the LIS during several phases of the last deglaciation, such as the meltwater pulses, is becoming more and more important as the implications of ice-dynamical instabilities for projections of future sea-level rise are becoming more apparent. I therefore believe that studies such as this one could be very interesting, as they demonstrate that explanations for such rapid retreat can potentially be found without invoking some strong atmospheric forcing.

We would like to thank Tijn Berends for reviewing our paper. The original comments are indented, while our responses aligned to the left of the page.

> However, I have a few concerns about both the methodology and the framing of the results, which I believe should be addressed before the manuscript can be published.

Before replying to all comments individually in detail, we want to make a general statement about the intent of our study.

In our study, we are showing the impact of our adaptive lake boundary condition on the ice dynamics, and thus on the glacial retreat. We show with our novel approach that the impacts of a lacustrine boundary condition on the glacial retreat is important and should be taken into account when studying the glacial retreat of land-terminating ice sheets. We are not trying to demonstrate a perfect reconstruction of the North American ice sheets. We are aware that for such an endeavor requires more advanced models for sub-components, and tuning of these would be necessary.

To test our model, we completed a side-by-side comparison of model runs that only differ in the absence or presence of a lake boundary. In this (preliminary) study, the focus with our experimental design was to simulate a glacial retreat scenario, in which lake basins could freely evolve along the retreating ice margin. As long as the model setup stays the same, it is sufficient to use a simplified model setup for other components.

In future studies, when aiming for more realistically reconstructions of the glacial retreat of ice sheets, more attention must be put into identifying and tuning the other feedbacks that are relevant to ice sheet evolution. These include, for example, the feedbacks between lakes and climate, more advanced ablation parameterizations at the ice-lake interface or more advanced GIA models.

In the following, we reply to all of your comments and highlight, where applicable, our changes made on the manuscript.
* * *
> Firstly, regarding the nature of the "proglacial lake ice-sheet instability" (PLISI). While the authors compare this to the better-known phenomenon of marine ice-sheet instability (MISI), they ascribe this instability to the elevation-temperature feedback (e.g. the first paragraph of section 5 Conclusions). However, MISI has nothing to do with mass balance processes, but is a purely ice-dynamical process, which is why all the different MISMIP experiments assume a uniform, unchanging, elevation-independent surface mass balance.

Yes, we agree. The original formulation was a bit misleading. We have rephrased the respective paragraphs in the Introduction, Discussion and Conclusion:

*l.74 ff*:

"In regions where there is ice-inward sloping topography, the grounding line retreats in a self-amplified manner, which corresponds to a rapid expansion of the lake. The PLISI, in combination with the increased runoff at the lowered ice surface in the warming climate, results in an accelerated retreat of the ice sheet. In our study, this mechanism leads to the demise of the LIS and finally to the drainage of the lake through Hudson Strait. In the control experiments, Hudson Bay is still glaciated when the present day conditions are achieved."

*l.392 ff*:

"The most drastic impact on the glacial retreat can be observed at a later stage, when the grounding line enters the deeply depressed basin of Hudson Bay (Fig. 8c - e). Since there is a reverse sloping bed, no stable grounding line position can be achieved (see Fig. 5), leading to an increase of ice flux into the expanding lake basin. This process is similar to the marine ice sheet instability (MISI, Weertman, 1974; Thomas and Bentley, 1978; Schoof, 2007). Contrary to the MISI, where the ice loss is generally driven by calving and sub-shelf melt processes at the ice shelf, we observe a dominance in ice loss via surface runoff over the latter processes. The strong increase of surface runoff is not surprising at the strongly lowered ice surface in the warming climate. However, we want to stress here that changes in local climate due to the presence of the lake might weaken this process (Krinner et al., 2004; Peyaud et al., 2007). This proglacial lake ice sheet instability (PLISI) described here, was also reported by Quiquet et al. (2021). Finally, the PLISI results in the disintegration of the ice saddle blocking drainage through Hudson Strait (Fig. 8d - e)."

*l.425 ff*:

"Once the lacustrine grounding line enters a deep basin on a retrograde bed, the ice sheet exhibits an instability similar to the MISI, which is called accordingly PLISI. As the ice thickness over the new grounding line position is higher, this increases the grounding line flux, which forces the grounding line to retreat even further. This positive feedback loop is ongoing until a stable grounding line position is reached. Due to increased melting at the lowered ice surface, the feedback cycle is further accelerated."

> In their rebuttal to my previous review, the authors show a timeseries of the different components of the total mass balance, showing that the retreat of the LIS is dominated by runoff and oceanic (not lacustrine) calving. This seems to be at odds with the findings of the different studies investigating MISI, and it also raises the question of how the presence of the lake can lead to such a strong lowering of the land-based ice dome, if the lake itself hardly removes any mass.

I am not sure here if I can follow you. As you said before, the MISI is a purely ice-dynamical process, which is caused by the unstable grounding line position. This increases the flow of ice over the grounding line. The MISI itself does not extract ice from the ice sheet. Ice from the shelf is then removed by different processes such as surface melt, calving or sub-shelf melt. I don't understand why it is so surprising that surface runoff is the dominating factor for ice loss in a transient experiment. Compared to other experiments that study MISI at Antarctica, air temperatures in our mid-latitude experiment are much higher and calving rates and sub-shelf melting are set to lower values in the lacustrine setting.

Furthermore, the figure that you refer to shows higher oceanic calving, because the LIS has longer marine than lacustrine ice margins, which have higher calving rates.

> Indeed, the authors report (in their rebuttal and in the supplementary material) that neither changing the sub-shelf melt rate nor the calving threshold thickness over the lakes significantly affects the results. If neither of these processes is significant, then what causes the difference in retreat rate between the lakes and the non-lakes simulations? The ice-dynamical processes governing MISI affect mass transport from the sheet to the shelf, but the shelf mass still has to go somewhere. If it is not removed by either sub-shelf melt or calving, then the shelf will grow thicker over time, the grounding-line will advance, and the basin will fill with grounded ice. The modelled grounding-line retreat must be caused by mass loss either on the shelf or on the sheet; the former does not happen, so the authors claim, but the latter should not be so different between the lake and no-lake experiment. This issue should be investigated further.

As described above, the shelf mass is mainly lost due to surface runoff.

> Regarding grounding-line retreat: in my previous review, I referred to the work of Natalya Gomez, who showed that gravitational effects can significantly reduce grounding-line retreat, and can even lead to stable configurations on (mildly) retrograde slopes even in the absence of buttressing. This was followed by a response from one of the authors, who claimed that the Lingle&Clark GIA model used by PISM is "self-gravitating". After consulting with a colleague who specialises in GIA, I found that this is only partially true; the Green's functions in the Lingle&Clark model include a self-gravitating term that is appropriate for a solid Earth that is in equilibrium with the surface load. The added ice mass on the surface is then balanced by the displaced mantle mass, so that the resulting gravitational perturbation is very small (deviating from zero because of the tensile strength of the Earth's crust, so that the locally displaced mantle mass does not necessarily equal the local ice load). However, this assumption is not appropriate for a retreating ice sheet; as phrased in the original article by Lingle and Clark: "Additional changes in depth caused by perturbation of the gravitational potential field are not included." The very existence of the vast proglacial lakes studied here is owed to the delayed rebound of the Earth's surface. At such moments, the gravitational signal can be significant, and the effect on water depth at the grounding line, and therefore on grounding-line retreat, should not be neglected. Based on the different studies by Natalya Gomez, I expect that this could significantly reduce the accelerated retreat reported by the authors. While I acknowledge that it might be too much work to include an appropriate GIA model in PISM for this study, the drawbacks of not doing so should be discussed in the manuscript.

Studying the self-gravitational effect of ice sheets on proglacial lakes (and thus on the glacial retreat) is undoubtedly a topic that will need to be investigated in the future. However, we assume that its impact on the lake geometry is minor compared to other factors as ice margin location, GIA signal and topography. See i.e. James et al. (2000): "*Changes to a level surface are affected by gravitational potential changes, although the effect is much smaller than crustal displacement changes.*".

Gomez et. al (2015), for example study the impact of the Laurentide-Cordilleran ice saddle collapse on the sea level, and they provide an estimate of ~160m local SL change for the combined effect of elastic GIA and gravitational changes. Although the Lingle-Clark model does not perfectly capture the details of the GIA signal, it produces Earth deformation that is in the same range as this value, and therefore provides a suitable way to test our lake model. However, we can not estimate the magnitude of the gravitational effect on the tilting of proglacial lakes.

Adding gravitational effects of the ice sheets would, compared to our current lake reconstructions, lead to a tilt of the lake surface normal towards the ice sheet. This would lead to an increase in water depth at the ice margin, and not, as you propose, to a drop in water depth. The magnitude would depend on the ice mass and distribution, but also on the lake spillway location. The difference here to the studies of Gomez et al., where they investigate the impact of loss of parts of Antarctica on the SL, is that their initial state (e.g. present day) has the gravitational anomaly of the ice mass already included. Therefore, when studying the impact of ice loss on the SL, this fact leads to a relative drop in SL. For this reason, we would argue that adding this effect would accelerate the glacial retreat rather than dampen it.

Nevertheless, we think that the gravitational effect would be smaller than the uncertainties in our lake model, or the modeled ice sheet itself. The right place to add such feature would be the GIA model, that provides the topography to the LakeCC model. We therefore added one sentence to the description of the ice sheet model:

*l.100 f*:

**"Self-gravitational effects of the ice sheet onto the sea level are not taken into account by this model. Compared to crustal deformation, this effect is only secondary (James et al., 2000)."**

We further want to point to your last issue here. With regard to this point we have reframed our results. Our intention with this study is not to study the glacial retreat of the LIS, but study the impact of the LakeCC model on the ice sheet retreat, by comparing it to the no-lake experiments. Therefore, discussing every limitation of each of the ice sheet model's submodules, is out of scope of this study. We do, however, acknowledge that this issue could indeed be relevant for simulating more realistic glacial retreat.
* * *
> Then, regarding the experimental set-up. The authors explain that their model is initialised with ice thickness and bed topography from the NAICE model, and thermodynamics are spun up to achieve a stable englacial temperature. However, when the simulation starts, ice volume rapidly increases to ~50% more than the initial value in all experiments (both with and without lakes), which suggests that the surface mass balance parameterisation is not properly tuned. The resulting over-sized ice-sheet (exactly how over-sized is difficult to quantify, as the authors rather confusingly chose to exclude ice in the Cordillera and the Canadian Arctic from the volume calculation) causes an unrealistically deep GIA depression, which leads to modelled lakes that are probably significantly larger than they would have been in reality. This likely means that the accelerated retreat reported by the authors is overestimated.

This is correct, we also state this in the manuscript. However we added the following paragraph to the Discussion:

*l.413 ff*:

**"We further want to note that the glacial retreat seen in our results might be strongly accelerated due to our simplified experiment setup. The large ice sheet growth, caused by the simple climate forcing, leads to deeply depressed topography and thus deeper lake basins."**

> The authors mention a "problem" with the initialisation of the Lingle&Clark GIA model, which causes the Hudson Bay to become subaerial when the simulation reaches the present day. They ascribe this to the difficulty of differentiating in their code between the initial state and the equilibrium reference state. They also state that circumventing this problem by starting the simulation during the previous interglacial was not feasible, as this "suffered from the fact that the bed deformation along the southern ice margin was so deep, that the basin was connected to the Atlantic Ocean, which consequently inhibited the formation of lakes". I find this unsatisfying; as with the "numerical instabilities" they report elsewhere (which they circumvent by creating a rather convoluted scheme of different lake water levels, masks, and filling rates), these kinds of coding problems should really be solved before using a model for research applications.

We want to clarify, that this problem is not a "coding" problem, that can fixed. It is rather a problem of the simple experimental setup. Ideally, a GIA model is initialized from an ice-free state, so that when the ice retreats again, the topography tends towards this initial topography. This, however did not result in a "LGM" state, from which we could start our experiments to test the lake model (Due to the excessively large ice sheet, the topography was unrealistically depressed). Therefore we chose an initial state from a prescribed LGM condition, which does allow for lakes to evolve, but that results in an unrealistic PD state. We agree that more effort need to be spend on this when aiming for a more realistic deglacial scenario. Since the goal of our study was more generally to look at the impacts of having a lacustrine boundary condition, this is out of the scope of the present study.

> Regarding the surface mass balance: in my previous review I referred to a few studies that showed how the presence of large proglacial lakes could positively affect the surface mass balance over the adjacent ice sheet, thereby potentially reducing retreat rates. The authors responded to this very briefly in their rebuttal, stating that including such SMB effects was beyond the scope of their study. However, they also claim that the accelerated retreat observed in their simulations is caused by surface mass balance processes (via the elevation-temperature feedback), which are triggered by the presence of the lake. I'd like to see some more discussion about why they think the latter process is so much stronger than the former.

We are not claiming, that the lakes' impact on local climate is generally negligible. In the Limitations section, we do acknowledge the lack of this feedback, stating that adding this feedback could potentially counteract the observed processes. In the revised manuscript, we have added a sentence about the potential stabilizing effect of the local climate effect of lakes on the ice sheet:

*l.398 f*:

**"However, we want to stress here that changes in local climate due to the presence of the lake might weaken this process (Krinner et al., 2004; Peyaud et al., 2007)."**

When aiming for a more realistic deglacial scenario, adding or further discussing this feature is needed. This will require a dedicated modelling setup, which is currently not implemented in PISM.

> Lastly, regarding the framing of the results: as I mentioned at the start, studies such as this one are important not only from a purely palaeoclimatological / palaeoglaciological perspective, but also for the way we think about near-future retreat of the Greenland and Antarctic ice sheets. The idea that ice-dynamical processes such as MISI, and more recently the ice-cliff instability caused by brittle fracture, can be as or even more important than atmospheric processes has only relatively recently become commonly accepted; the uncertainty in sea-level projections beyond 2100 is dominated by ice-dynamical terms, and a lot of effort is being dedicated to improving our understanding of these processes and reducing those uncertainties. Understanding the interplay between atmospheric and ice-dynamical processes in the geological past is an important part of this effort. I feel that the authors here could improve the readability of their manuscript by more clearly framing their study in this context; they could choose to present it as (A) a schematic study that investigates a particular process (e.g. PLISI), (B) a reconstruction of ice-sheet / lake / GIA evolution during the last deglaciation, or (C) a system-based study that looks at the role of lakes in the Earth system. Right now, I feel the manuscript does not really fall in any of these three categories, which makes it difficult to decide which drawbacks are acceptable and which are not. If the only aim is to quantify the ice-dynamical processes, then the lack of atmospheric processes is not problematic. If the authors want to go for a realistic reconstruction, then the choice of climate forcing is probably the largest source of errors. If they want to take a comprehensive approach to the Earth system, then the forcing, timing, and geometry are probably of lesser concern than the lack of atmospheric / geoid / GIA feedbacks. I suggest that the authors make a conscious choice about which direction they want to move in with this study, and frame the drawbacks and uncertainties of their findings accordingly.

As we have also written above, the intention of this study is to describe and test our lake model and demonstrate that proglacial lakes likely play a significant role in ice sheet retreat. It is neither *(B) a reconstruction of ice-sheet / lake / GIA evolution during the last deglaciation*, nor *(C) a system-based study that looks at the role of lakes in the Earth system*. But we think that it is also not *(A) a schematic study that investigates a particular process (e.g. PLISI)*. The PLISI is only one process being triggered by the presence of the lakes.

To be more clear, we highlight in several places that the glacial retreat setup is simplified and meant only as a testcase for the LakeCC model. E.g. in the Abstract:

*l.7 ff*:

**"As a test scenario, a simplified glacial retreat setup of the Laurentide Ice Sheet (LIS) is used. By comparing the lake experiments with no-lake control runs, we show that the presence of proglacial lakes [...]"**

As we have written before, we are testing the impact of our model on the ice dynamics, by side-by-side comparing the lake vs. no-lake experiments. For this purpose, the lack of more advanced feedbacks is not problematic.

In the previous manuscript we discussed problems of the glacial retreat that were due to the simple model configuration. Since these were not related to glacio-lacustrine interactions, we chose to change this: The "Experimental setup" section was renamed "Experiments" and all details about initialization and parameterization of the different submodels were moved into the Appendix. In the Experiments section, we refer to the respective section in the Appendix and briefly mention that realistic glacial retreat patterns can not be expected from our simplified model setup:

*l.295 ff*:

**"Further details on the used parameterizations are given in Sect. A.
The experiments for this study are all based on the same simplified model setup. To properly simulate a realistic glacial retreat, more advanced models and setups would be needed. For example, the simple climate forcing using a glacial index leads to increased mass accumulation in cold regions and on top of the ice sheet. The experiments therefore suffer from excessive ice sheet growth after model initialization. Also, PISM's default GIA model is based on a simple two-layered Earth model and therefore lacks viscosity variations in the upper and lower Earth mantle. These variations significantly contribute to the GIA signal in central Canada (Wu, 2006). However, these shortcomings are the same for all of our experiments."**

Furthermore, we removed the sub-section "Transient experiments" from the Discussion, because the differences to a realistic glacial retreat and their causes, are not relevant in this context.

**Literature**

- James, T. S., Clague, J. J., Wang, K. & Hutchinson, I. Postglacial rebound at the northern Cascadia subduction zone. Quaternary Science Reviews 19, 1527–1541 (2000).

- Gomez, N., Gregoire, L. J., Mitrovica, J. X. & Payne, A. J. Laurentide-Cordilleran Ice Sheet saddle collapse as a contribution to meltwater pulse 1A. Geophysical Research Letters 42, 3954–3962 (2015).

**Author response to Report#1 by Referee#3 to the revised version of manuscript No. tc-2020-353, "PISM-LakeCC: Implementing an adaptive proglacial lake boundary into an ice sheet model"**

**submitted to The Cryosphere by Sebastian Hinck et al.**

> First I would like to thank the authors for their efforts in providing detailed and thoughtful responses to the reviews. From my point of view I still really like this study, and the questions I raised in the first review have been addressed - but with just one exception which I still believe is important.

We would like to thank the anonymous referee #3 for reviewing our paper. The original comments are indented, while our responses aligned to the left of the page.

In the following, we will respond to all comments and questions.

> It is clear that the lake levels are determined by the elevation of spillways, either over land or over ice - as in the snapshots in Fig 8 where the changes in spillways are very helpfully highlighted. Although I agree that modelling subglacial drainage of proglacial lakes is far beyond the scope of this paper, neglecting this process does represent a potentially important limitation of LakeCC. I really urge the authors to clearly acknowledge this limitation, in both the discussion and abstract, particularly as the paper is in TC not GMD.
>
> Perhaps I could suggest two possibilities in support of my opinion.
>
> Firstly, subglacial drainage events (if they do occur) will prevent the lake from reaching the level/extent predicted by LakeCC. This is particularly likely once the ice saddle becomes narrow (see the 17.5 kyr snapshot). The lake level must of course be at least as high as the lowest bedrock saddle, or sea level, whichever is higher. However, the potential for triggering either PLISI or MISI could be considerably moderated if subglacial drainage prevents lakes from filling completely.
>
> Secondly, overspilling over an ice saddle with LakeCC is associated with rapid disintegration of the ice saddle due to the MISI (as reported in Section 4.3). Ice from the saddle is presumably lost completely by this process. Even if the MISI is initiated, the ice saddle can remain intact once rapid subglacial drainage triggered by dynamic thinning empties the lake before the MISI has completely removed the ice saddle. Once the lake has drained, remaining ice will close the subglacial drainage channels and allow the lake to refill. Amongst several obvious differences in simulated deglaciation under this scenario is the potential for cyclic drainage/filling events, or for a smaller but more persistent lake than that predicted by overspill alone.
>
> Adding some discussion of this limitation (and ideally a sentence in the abstract) would not detract from what is a useful and well-presented paper.

We agree that subglacial drainage is an important process that can regulate the evolution of proglacial lakes. However, mentioning it directly in the abstract is not appropriate in this context, we think. We rather highlight in the abstract the simplified assumptions made for water level calculation, which the lack of sub-glacial drainage is somewhat related to:

*l.6 f*:

**"For simplicity, the PISM-LakeCC model assumes lake basins to always tend to be filled to the brim."**

However, subglacial drainage could also be integrated into such model keeping the simplifications and assumptions unchanged, as it can be regarded simply as a lower spillway, that limits the maximum water level of a basin. Implementation of such process is not straightforward, as the dynamics is complex. We mention the lack of such a parameterization in the Limitations section:

*l.226 ff*:

**"Another process which is missing in this model is sub-glacial drainage through channels. This process would potentially limit the maximum water level of basins dammed by a narrow ice saddle. However, parameterization of channel formation is not trivial."**

> Two minor points... Line numbers refer to the version with track changes.

>> L205 "The model does not allow for refreezing at the ice shelf base".

> Why is this not the case? Also it's inconsistent with Fig 2, Process #2 "Sub-shelf melting/refreezing".

The way this melting/refreezing scheme was implemented into PISM, the model uses prescribed salinity and ambient water temperatures (35PSU & -1.7°C), as also mentioned in the manuscript. Using these fixed parameters the calculated (salinity and depth- dependent) freezing point of water is always below ~ -1.9°C. This is colder than the prescribed ambient water temperature and consequently refreezing is not permitted. We have rephrased this sentence more precisely:

*l.185 ff*:

**"The model's choice of parameters is such that the temperature of the ambient water is always above the calculated freezing temperature and consequently the model does not allow for refreezing at the shelf base."**
* * *
>> L302 "the numerical representation of a physical system requires the underlying equations to be smooth functions".

> If 'smooth' means differentiable then I'm not sure that restriction applies to all numerical representations of all physical systems. The difficulty with finding the reason for the numerical instability is clear and I suggest simply stating that the solver issues are associated with the appearance/disappearance of lake and ocean basins.

You were right, the original formulation was written too generally. We kept the sentence, but weakened its statement.

*l.278 ff*:

**"From a numerical point of view, however, this is not surprising, as commonly the numerical representation of a physical system requires the underlying equations to be sufficiently smooth."**
* * *
> Please check again for typos/grammar. For example:
> L18. Eurasian and North American continent. Change "continent" to "continents".
> L62. Recent work of Sutherland (2020) present... Change "present" to "presents".
> L113. The till below the water level next to the grounding line are... Change "are" to "is".
> L138. Details on how this lake boundary condition affects the ice dynamics is... Change "is" to "are".
> L266. ...implementation to model lubrication... Change "to" to "of".
> L333 "by setting" has been mistakenly deleted in "by setting precipitation to zero".

Thank you for the suggested edits. We have corrected the respective passages in the revised manuscript and also checked for further typos!

---

## Author Response (AR3)

**Author response to comments from the Editor to the revised version of manuscript No. tc-2020-353, "PISM-LakeCC: Implementing an adaptive proglacial lake boundary into an ice sheet model"**

**submitted to The Cryosphere by Sebastian Hinck et al.**

As requested by the editorial office, we provide

- the updated manuscript,
- a marked-up manuscript version showing the changes made, and
- the point-to-point response to the comments (this document).

The original comments are indented, while our responses aligned to the left of the page.

> Dear Hinck et all,
> thanks for your detailed reply to the comments by the two reviewers on your revisions of the manuscript submitted to the Cryosphere. The revisions where major and you have answered most of these.
>
> However, the reviewers had a number of concerns which you should answer in more detail in the final revised manuscript. In particular, this relates to the second review by Tijn Berends.
>
> You have written out answers to each comment in the reply to the reviewers. However, I expect that you also make these points in the reply letter clear to the readers, as it is likely they will have similar questions when reading your paper.
>
> In particular, please add details to the manuscript making the following key points from the reviewers more clear:

We want to thank the Editor for his thoughtful comments and for giving us the opportunity to add further details that help improving our manuscript. In the revised manuscript, we tried to respond to all of the concerns.

In the following, we reply to all of your comments and highlight, where applicable, the changes made in the manuscript.

> 1. the relative impact of surface runoff on the shelf mass (as opposed to sub-shelf melt or calving);

This is a good point, that was really not made clear enough in the previous manuscript. To show the relative impact of surface runoff, we added an additional figure, showing the runoff side-by-side with sub-shelf melt and calving rates. This plot is discussed in the Results section:

**l. 342ff: "At the southern ice margin and especially at the ice shelves, where the surface elevation is low, the surface runoff is greatly increased (see Fig. 10a). Figure 10b shows the modeled mass flux due to sub-shelf melting and calving. In this region, surface runoff contributes the most to mass loss."**

[Figure]

**Figure 10.** Comparison of the modelled mass loss due to a) surface runoff and b) sub-shelf melting and calving, shown here at 16 kyr. At the shelf regions, the surface runoff is about an order of magnitude larger than sub-shelf melting. Locally, at the shelf margin, the mass loss may be greatly increased due to calving events. The fields shown here are averaged over the ice model's reporting interval (here: 5 years).

We further expand on this in the Discussion section:

**l. 387ff: "Contrary to MISI, where the ice loss is generally driven by calving and sub-shelf melt processes at the ice shelf, we observe a dominance in ice loss via surface runoff (see Fig. 10), due to the strong surface-elevation feedback in the warm climate. At the ice-shelves, the surface runoff is so large that when comparing with an experiment that has a reduced calving threshold, the shelf geometry hardly differs. The ice that is not calved off is subject to strong melting (see sensitivity run *RedCalv* in the supplementary material)."**

> 2. expand on the impact of crustal deformation relative to gravitational effects;

As previously stated, due to lack of an appropriate model that includes gravitational effects we can not give an estimate of the relative strength of this process compared to crustal deformation. We do, however, expand on this by describing the potential impact on the lake surface.

**l. 484ff: "Another feature missing in the GIA model is self-gravitational effects of the ice sheet. The ice sheet's mass impacts the geoid, along which the free water surfaces align. As a result, the lake water is attracted towards the ice sheet and the water depth at the grounding line would potentially increase. Due to lack of an appropriate model of gravitational change, we can not estimate the potential magnitude of this effect. However, according to James et al. (2000) the effect is secondary compared to the crustal deformation."**

> 3. clarify the main focus of the paper (see note from Berents);

In the Introduction we do describe the potential importance of the proglacial lake boundary for model simulations of palaeo ice sheets, such as the Laurentide or the Fennoscandian ice sheet. In previous modelling studies this aspect was mostly ignored. Our main intention is to fill this gap by proposing a new type of model, which, as we demonstrate using a simple test scenario, drastically impacts the glacial retreat. We think that this sufficiently sets the context for our study.

To highlight the study's focus, we reworded parts of the Abstract:

**During the Late Pleistocene and Holocene retreat of palaeo ice sheets in North America and Europe, vast proglacial lakes existed along the land terminating margins. These proglacial lakes impacted ice sheet dynamics by imposing boundary conditions analogous to a marine terminating margin. These lacustrine boundary conditions cause changes in the ice sheet's geometry, stress balance and frontal ablation and therefore affect the entire ice sheet's mass balance. Despite this, dynamically evolving proglacial lakes have rarely been considered in detail in ice sheet modelling endeavors. In this study, we describe the implementation of an adaptive lake boundary into the Parallel Ice Sheet Model (PISM), which we call the PISM-LakeCC model. We test our model with a simplified glacial retreat setup of the Laurentide Ice Sheet (LIS). By comparing the experiments with lakes with control runs with no lakes, we show that the presence of proglacial lakes locally enhances the ice flow, which leads to a lowering of the ice sheet surface. In some cases, this also results in an advance of the ice margin that causes the emergence of ice lobes. In the warming climate, increased melting on the lowered ice surface drives the glacial retreat. For the LIS, the presence of lakes triggers a process similar to the marine ice sheet instability, which causes the collapse of the ice saddle over Hudson Bay. In the control experiments without lakes, Hudson Bay is still glaciated when the climate reaches present day conditions. The results of our study demonstrate that glacio-lacustrine interactions play a significant role of the retreat of land terminating ice sheet margins.**

In the revised manuscript, we expanded the discussion on the results: We now provide an extended table giving an overview about all experiments (including the sensitivity runs).

**Table 1. Overview of all experiments done for this study. The first three experiments are discussed in the text, details about the other experiments can be found in the supplementary material.**

| Name | Description | Results |
|---|---|---|
| *LAKE*[*] | standard lake experiment, as described in the text | accelerated glacial retreat; occurrence of PLISI; Hudson Bay fully ice free at the end of the experiment |
| *CTRL*[*] | standard no-lake experiment, land-sea mask corrected by the SL2DCC model | Hudson Bay remains mostly glaciated at the end of the run |
| *DEF*[*] | PISM default no-lake setup, occurrence of inner-continental ocean basins | slightly faster glacial retreat than *CTRL*; Hudson Bay glaciated north of $\sim 58°$ N at the end of the run |
| *IncCalv* | increased calving; lacustrine thickness calving threshold set to 500m | almost immediate removal of shelf ice, which leads to a more rapid glacial retreat than in *LAKE* |
| *RedCalv* | reduced calving; lacustrine thickness calving threshold set to 20m | apart from slightly larger ice shelves, similar to *LAKE* |
| *MR* | tuning parameter for sub-shelf melting adapted to account for differences between marine and lacustrine environment | as above |
| *nSG* | slippery grounding line model disabled | strongly reduced grounding line flux leads to smaller ice shelves; no PLISI; Hudson Bay still glaciated north of $\sim 59°$ N |
| *TWO* | use of grounding line treatment proposed in Albrecht et al. (2020) (tillwater ocean) instead of slippery grounding line model | no qualitative difference to *LAKE* |
| *GIA* | adapted Earth model parameters for the Lingle-Clark bed deformation model | no qualitative difference to *LAKE*; only the timing is slightly different |
| *FR5* | lake fill rate set to 5m year$^{-1}$ | no qualitative difference to *LAKE* |
| *FR10* | lake fill rate set to 10m year$^{-1}$ | as above |
| *FR50* | lake fill rate set to 50m year$^{-1}$ | as above |

[*] default experiments

Furthermore, we have improved the structure of the manuscript by collecting and expanding on the shortcomings of the experimental setup into an own section. This section also includes the content from former section "2.4.3 - Further lacustrine interactions". As these issues all deal with issues of other sub-models of the ice sheet model, and thus only indirectly concern the LakeCC model, this section was moved into the **Appendix A - Experiments**:

**I. 469ff:**
**"A2 Limitations**
**In the following, we will mention and shortly discuss some limitations of the experimental setup.**

**A2.1 Climate model**

The climate forcing is relatively simple in our setup. Using a glacial index leads to increased mass accumulation in cold regions and on top of the ice sheet. The experiments therefore suffer from excessive ice sheet growth after model initialization. Furthermore, the presence of vast proglacial lakes would impact the local climate by reducing temperatures and increasing precipitation patterns (Krinner et al., 2004; Peyaud et al., 2007). This could locally increase the ice sheet's SMB and counteract the accelerated mass loss observed in this study. Also, the potential impact on ocean circulation and global climate due to redistribution of freshwater (Broecker et al., 1989; Teller et al., 2002; Condron and Winsor, 2012) is ignored here.

**A2.2 GIA model**

PISM's default GIA model (Lingle–Clark; Bueler et al., 2007) is based on a simple two-layered Earth model and therefore lacks viscosity variations between the upper and lower mantle. This variation significantly contributes to the GIA signal in central Canada (Wu, 2006). In combination with the excessive mass accumulation due to the simple climate forcing, our results show a strongly depressed topography, with deep lake basins that only slowly relax. For a realistic simulation of the LIS deglaciation, with a proper representation of proglacial lakes, a more advanced model to calculate GIA signal would be needed.

Another feature missing in the GIA model is self-gravitational effects of the ice sheet. The ice sheet's mass impacts the geoid, along which the free water surfaces align. As a result, the lake water is attracted towards the ice sheet and the water depth at the grounding line would potentially increase. Due to lack of an appropriate model of gravitational change, we can not estimate the potential magnitude of this effect. However, according to James et al. (2000) the effect is secondary compared to the crustal deformation.

Furthermore, in the calculation of the GIA signal we do not include the mass held by the lakes. We would expect the water mass to have a significant impact on the GIA and thus also on the lake basins.

**A2.3 Model initialization**

Initialization of the Lingle–Clark model from a glaciated state is problematic here. For the model to calculate a relief topography, to which bed deformation is applied, it should ideally be initialized from an interglacial state when the residual GIA from previous glaciations is limited. Test runs, comparable to those in Niu et al. (2019), however, suffered from the fact that the bed deformation along the southern ice margin was so deep, that the basin was connected to the Atlantic Ocean. This consequently inhibited the formation of lakes. We therefore chose to initiate the experiments from NAICE LGM reconstructions. The mismatch in calculated relief topography, results in ice free regions to over-relax compared to the modern topography. Hudson Bay, for example, is elevated above sea level at PD (see Fig. 8f)."
* * *
> 4. expand on the potential impact of sub-glacial drainage.

In the revised manuscript, we expand on this topic.

l. 226ff: "One process that can limit the maximum fill height of a lake is sub-glacial drainage. In the LakeCC model it is only crudely included via the flotation criterion: when an ice dam becomes buoyant and opens a new drainage route. In reality, however, sub-glacial drainage also can happen on much smaller scales through channels underneath the ice. This process could also lead to repeated lake drainage and refilling events. Even though sub-glacial drainage through channels might be an important aspect of glacio-lacustrine interactions, its parameterization is not trivial and is not included in our model."
* * *
> For most of these issues you have answered the reviewers concerns in the reply letter, but not added sufficient detail in the revised manuscript. When resubmitting your final paper, keep in mind that you are writing for a reader in TC and not GMD - as such, the article should focus on the scientific results, their importance and context (in addition to the detailed description of the proglacial lake module).

> These additional revisions should not take too long. Please include a version with tracked changes when submitting the final version.

We hope that our revised manuscript could resolve all issues! The manuscript and a version with all marked changes is uploaded.

---

## Author Response (AR4)

**Author response to comments from the Editor to the revised version of manuscript No. tc-2020-353, "PISM-LakeCC: Implementing an adaptive proglacial lake boundary into an ice sheet model"**

**submitted to The Cryosphere by Sebastian Hinck et al.**

> Comments to the author:
> Dear Sebastian Hinck et al. thanks again for revising your manuscript to TCD. I am happy to accept the current version for publication in TC.

Dear Kerim Nisancioglu,

we are glad to have finally finished the review process. Thank you for your thoughtful comments that helped improve our manuscript.

> Non-public comments to the Author:
> Please find a few very minor technical corrections with suggested edits to the language:
>
> In some cases, this also results in an advance of the ice margin that causes the emergence of ice lobes.
> -> In some cases, this also results in an advance of the ice margin AND the emergence of ice lobes.
>
> The results of our study demonstrate that glacio-lacustrine interactions play a significant role of the retreat of land terminating ice sheet margins.
> -> The results of our study demonstrate that glacio-lacustrine interactions play a significant role IN the retreat of land terminating ice sheet margins.
>
> In reality, however, sub-glacial drainage also can happen on much smaller scales through channels underneath the ice.
> -> In reality, however, sub-glacial drainage can ALSO happen on much smaller scales through channels underneath the ice.
>
> This process could also lead to repeated lake drainage and refilling events.
> -> This process could lead to repeated lake drainage and refilling events.

We have applied the requested changes to the final manuscript. Furthermore, minor changes (rephrasing and spelling) were applied to the supplementary material.

Best regards,
Sebastian Hinck